# Targeting CXCL16 and STAT1 augments immune checkpoint blockade therapy in triple-negative breast cancer

Bhavana Palakurthi[1,2], Shaneann R. Fross[1,2], Ian H. Guldner[1,2],
Emilija Aleksandrovic [1,2], Xiyu Liu[1,2], Anna K. Martino [1], Qingfei Wang [1,2],
Ryan A. Neff[1], Samantha M. Golomb[1,2], Cheryl Lewis [3], Yan Peng[3],
Erin N. Howe[1,2] & Siyuan Zhang [1,2,3,4] ✉

Chemotherapy prior to immune checkpoint blockade (ICB) treatment appears to improve ICB efficacy but resistance to ICB remains a clinical challenge and is attributed to highly plastic myeloid cells associating with the tumor immune microenvironment (TIME). Here we show by CITE-seq single-cell transcriptomic and trajectory analyses that neoadjuvant low-dose metronomic chemotherapy (MCT) leads to a characteristic co-evolution of divergent myeloid cell subsets in female triple-negative breast cancer (TNBC). Specifically, we identify that the proportion of CXCL16 + myeloid cells increase and a high STAT1 regulon activity distinguishes Programmed Death Ligand 1 (PD-L1) expressing immature myeloid cells. Chemical inhibition of STAT1 signaling in MCT-primed breast cancer sensitizes TNBC to ICB treatment, which underscores the STAT1's role in modulating TIME. In summary, we leverage single-cell analyses to dissect the cellular dynamics in the tumor microenvironment (TME) following neoadjuvant chemotherapy and provide a pre-clinical rationale for modulating STAT1 in combination with anti-PD-1 for TNBC patients.

Cancer is a heterogeneous disease. As such, a systemic view of cancer as an evolving ecosystem, rather than a singular view of specific cell type or gene, is essential to guide a more effective and sustained anti-cancer treatment[1]. The divergent response to anti-cancer immunotherapy among patients highlighted an urgent need for systemic interrogation of the dynamic tumor ecosystem due to the nature of immune regulation – a dynamic equilibrium, balancing the immunogenic and immunosuppressive/tolerogenic signals[1–3]. Myeloid cells are prominent regulators of immune equilibrium vital to normal tissue homeostasis and tumor progression[2]. Their diverse roles in tuning tumor immune microenvironment (TIME) have been increasingly recognized as critical, sometimes decisive factors, for an optimal anti-tumor immunotherapy response[4].

Growing literature suggests a perplexing antagonistic/compensatory/redundant phenotypic plasticity among different myeloid cell subsets under various cancer contexts - from the binary M1/M2 paradigm and functionally-defined myeloid-derived suppressor cells (MDSC) to the ever growing characterization of the sub-types of myeloid cells[2,5]. Myeloid cells are extremely heterogeneous and plastic in response to the cues imposed from TIME and systemic anti-cancer treatment[6]. For instance, PARP inhibitors (in triple-negative breast cancer (TNBC)) mediate glucose and lipid reprogramming and enhance both anti- and pro-tumorigenic features of macrophages through sterol regulatory element-binding protein-1 pathway[7]. Similarly, expansion of immunosuppressive M2-like macrophages and contraction of inflammatory-macrophages during advanced renal cell

[1]Department of Biological Sciences, College of Science, University of Notre Dame, Notre Dame, IN 46556, USA. [2]Mike and Josie Harper Cancer Research Institute, University of Notre Dame, 1234N. Notre Dame Avenue, South Bend, IN 46617, USA. [3]Department of Pathology and Simmons Comprehensive Cancer Center, University of Texas Southwestern Medical Center, Dallas, TX 75235, USA. [4]Indiana University Melvin and Bren Simon Cancer Center, Indianapolis, IN 46202, USA. ✉e-mail: Siyuan.Zhang@UTSouthwestern.edu

carcinoma leads to decreased T-cell receptor clonality[8]. It is clear that the coordinated interplay between heterogeneous TIME cells dictates the anti-tumor immune responses. A systematic understanding of this coordination is critical for designing effective novel immunotherapy regimens.

In this study, beyond the isolated view of a given myeloid cell subset function, we use breast cancer as a model to take a systemic view of tumor-associated myeloid cells in response to a commonly used chemotherapy regimen. Through single-cell trajectory analysis, we observe a dynamic homeostasis between sub-clusters of myeloid cells in response to chemotherapy priming. Modulating the STAT1 signaling pathway in tumor-infiltrating myeloid cells fine-tunes the anti-cancer immunity of TIME to maximize the therapeutic efficacy of anti-PD1 for TNBC.

## Results

### MCT improves T-myeloid spatial proximity

We compared the impact of different chemotherapy dosing schemes (maximum tolerated dose (MTD) or MCT) on breast tumor growth in vivo using three syngeneic breast tumor mouse models (Supplementary Fig. 1a). In the Human Epidermal Growth Factor Receptor 2 positive (HER2$^+$) tumor model (MMTV-neu), MTD and MCT achieved similar efficacy by reducing the tumor growth to ~50% (mean relative volumes being 2 and 1.8 respectively) of the Vehicle-treated tumors at the end of treatment on Day 28 (mean relative volume 3.7) (Fig. 1a, left). In the TNBC mouse model (C3-1-TAg), MTD did not, but MCT significantly reduced the tumor growth to 46% of Vehicle-treated tumors (Fig. 1a, center). In the adenocarcinoma tumor model (MMTV-PyMT), MCT significantly reduced the tumor growth to 50% of Vehicle-treated tumors (Fig. 1a, right). In all three models, MCT treatment significantly extended progression-free survival (PFS) (Fig. 1b) and reduced tumor proliferation, as depicted by H-Score of Ki-67 staining (Supplementary Fig. 1b). Above results derived from three breast cancer models are consistent with observed clinical benefit of MCT[9,10].

Given the perceived myeloid toxicity and potential benefit of MCT in modulating TIME, next, we sought to understand the immune cell composition and spatial proximity within the tumor under different treatment conditions. The count of CD45$^+$ cells in peripheral blood decreased from $5 \times 10^3$ in Vehicle treated to $3 \times 10^3$ per µL blood upon MTD treatment while the count increased to $14 \times 10^3$ per µL of blood in MCT-treated mice (Supplementary Fig. 1c). However, in the tumor-bearing mice, all treatments significantly reduced the CD45$^+$ immune cell percent in the peripheral blood (Supplementary Fig. 1c), suggesting tumor formation profoundly changed systemic immune regulation[6,11]. Interestingly, MCT-treated tumors showed a significant increase in tumor-infiltrating CD3$^+$ lymphocytes (TIL) (Supplementary Fig. 1d) with evident "hotspot" areas with spatially proximal Iba1$^+$ myeloid cells and CD3$^+$ TIL (Fig. 1c). The percent of Iba1$^+$ myeloid cells spatially proximal to CD3$^+$ TIL significantly increased in MCT-treated tumors upon MCT treatment (Fig. 1c). Similarly, in the MMTV-PyMT model, CD8a$^+$ T cells that are spatially proximal to Iba1$^+$ myeloid cells increased in MCT-treated MMTV-PyMT tumors (Fig. 1d).

To examine the MCT-mediated tumor-associated immune landscape modulation, we immunophenotyped primary breast tumors under Vehicle, MTD, and MCT strategies using a 19-antibody Cytometry Time of Flight (CyTOF) panel (Supplementary Table 1). We downsampled CyTOF detected immune cells in each treatment group to the equal total number per treatment group and clustered them using the viSNE algorithm (Supplementary Fig. 1e). After pooling the data from two independent CyTOF experiments, we quantified the changes of tumor-associated immune cell subtypes based on traditional manual gating (Supplementary Fig. 1f). Interestingly, MCT treatment preserves a significant amount of CD11c$^+$ dendritic cells (DC) of all myeloid cells (Fig. 1e, f).

To investigate potential antigen-specific CD8$^+$ T-cell priming and proliferation induced by DCs, we performed co-culture experiments using a chicken ovalbumin (Ova 257-264)-OT-I system. We first established E0771.ova tumor cells with the membrane presentation of SIINFEKL peptide as validated by flow cytometer (Supplementary Fig. 1g). We also validated Ova 257-264 specific T cells cells from OT-I transgenic mice using Flex-T™ H-2K(b) OVA (SIINFEKL) Tetramer (Supplementary Fig. 1h). Next, four weeks after MCT treatment, we isolated CD11c$^+$ tumor-infiltrating DCs (TIL DCs) or E0771-Ova tumor-draining lymph node derived DCs (LN-DCs) and co-cultured them with total CD3$^+$ OT-I T cells. We examined T-cell expansion/proliferation using CFSE dye dilution assay after co-culture with TIL-DCs or LN-DCs. When naïve T cells from OT-I mice were stimulated with anti-CD3-CD28, 60.5% of CD8 T cells divided more than 3 times (Fig. 1g, h). When co-cultured with MCT-treated TIL-DCs, 28.3% of naive CD8$^+$ T cells have proliferated more than 3 times compared to 7.6% in the Vehicle-treated TIL-DCs group (Fig. 1g, h), indicating an increase of ova antigen-specific activation of T cells under the MCT treatment. On the other hand, when OT-I T cells were co-cultured with LN-DCs, T-cell proliferation was similar in both the treatment groups (Supplementary Fig. 1i). Furthermore, we observed that MCT derived DC co-cultured with T cells led to more CD44$^+$ T cells and ~15% CD44$^+$ proliferating CD8$^+$ T cells showed a higher level of IFN-γ, which is comparable with T cells stimulated with anti-CD3-CD28. Neither co-culture with Vehicle-treated TIL-DCs nor LN-DCs induced a significant increase of IFN-γ$^{high}$ T cells (Fig. 1i).

### CITE-seq reveals co-existence of distinct TME myeloid cells

To further characterize the immune cell transcriptome changes after MCT treatment at single-cell level, we employed Cellular Indexing of Transcriptomes and Epitopes by Sequencing (CITE-seq) single-cell transcriptome and mass cytometry approaches (Supplementary Fig. 2a, flow chart). For CITE-seq analysis, we stained tumor and tumor-associated immune cells with a 29-antibody panel representing well-known immune cell surface markers (Supplementary Table 2, immune Cell-ID). We manually gated these CITE-CD45$^+$ cells into myeloid (CD11b$^+$) and lymphoid (CD3$^+$) cell subsets (Supplementary Fig. 2b). Among the CD11b$^+$ myeloid cells, we identified cDCs and macrophages based on known markers that have been previously identified in human tumor-infiltrating immune cells[12–14], including tumor-associated dendritic cells (XCR1$^+$ cDC1, SIRP1a$^+$ cDC2, SiglecH$^+$ pDC) and tumor-associated macrophages (TAMs) (Supplementary Fig. 2b). RNA-based Uniform Manifold Approximation and Projection (UMAP) analysis revealed 17 different single-cell RNA clusters - here we refer to them as Seurat RNA Clusters (Fig. 2a, left). Broadly, these clusters exhibited known transcriptional signatures representing unique immune cell types (as labeled below UMAP and Supplementary Data 1). Interestingly, canonical DCs and TAMs identified conventional surface makers (immune Cell-ID) gating (Supplementary Fig. 2b) did not overlap well with any specific RNA clusters when projected on the RNA UMAP (Fig. 2a, right). This discordance could be partially attributed to the transcriptomal plasticity of myeloid cells[15,16].

Next, we examined marker genes of each RNA cluster identified (Fig. 2b and Supplementary Data 1). Cluster 0 expressed inflammatory (S100a8/9), immunosuppressive genes (Hcar2 and PD-L1), Ccr1, Nfkbia, and Nlrp3. Since these cells lacked DC maturation genes and were expressing Nlrp3, an immature DC marker[17], we categorized Cluster 0 as immature myeloid cells. Cluster 1 expressed DC progenitor genes (Trps1, Rel, and Etv3). Clusters 2, 6, and 11 expressed Keratin genes (Krt7, Krt8, and Krt18), suggesting their potential tumor origin. Clusters 3, 7, 8, and 10 gene signatures overlapped with both DC and macrophage gene signatures (Itgax, Csf1r, Irf8, Mrc1, Etv3, Cd74, H2-Ab1, and Cd86). Cluster 4 expressed MDSC gene signature (Arg1, Irf8, Itgam, and Lgals3). Cluster 12 expressed high proliferation and myeloid specific genes (ki-67, Cd86, and Cd72). Clusters 13 and 16 expressed fibroblastic

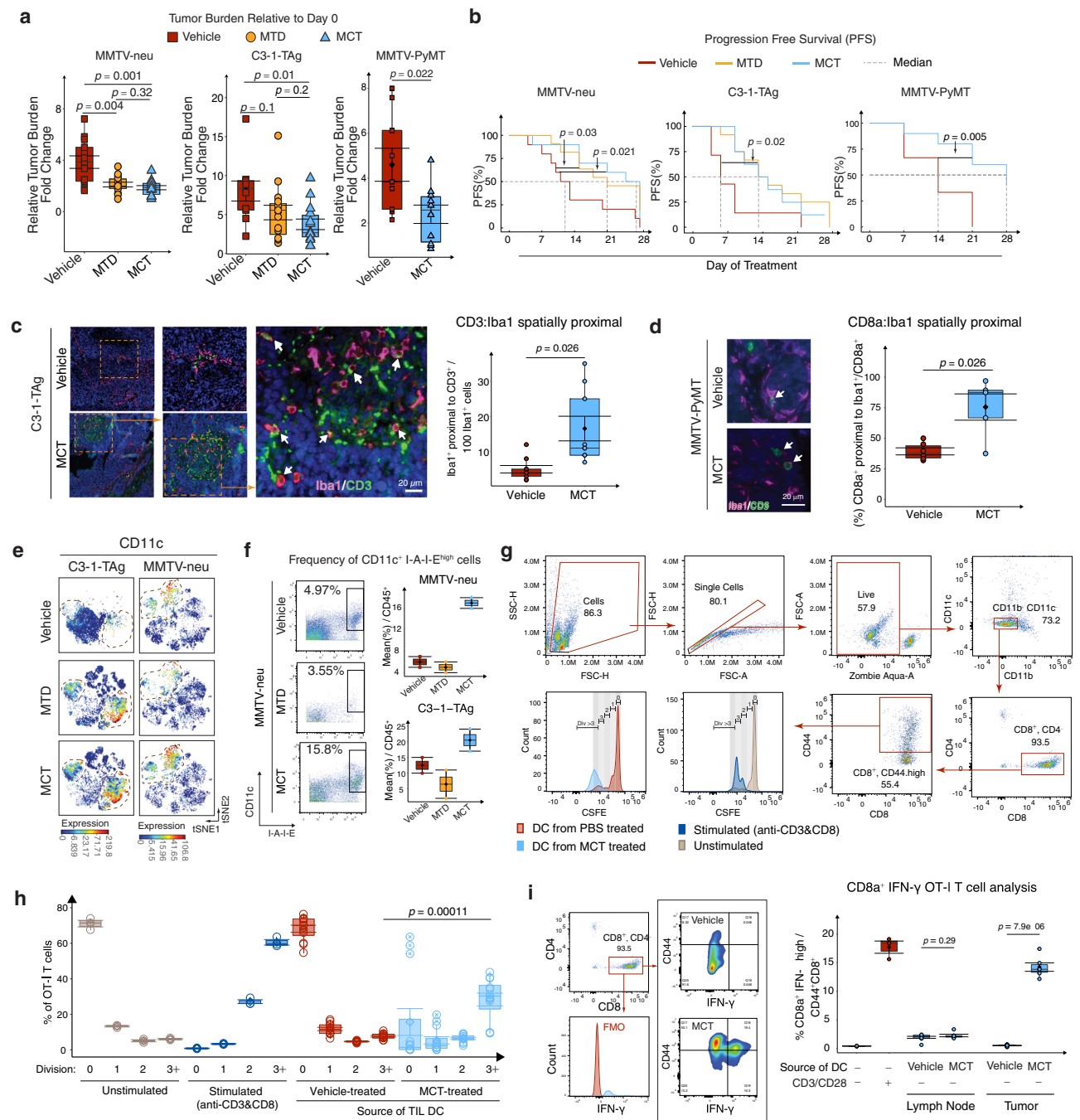

**Fig. 1 | MCT increases T - myeloid cell spatial proximity. a** MTD = Maximum tolerated Dose and MCT = Metronomic Dose. Box plots show tumor burden on Day 28 relative to Day 0 of treatment in MMTV-neu (left), C3-1-TAg (center), and MMTV-PyMT (right) mice. **b** PFS (%) indicates the tumor volume doubling time in MMTV-neu (left), C3-1-TAg (center), and MMTV-PyMT (right) (generated using the Survimer R package, log rank *p*-value). **a**, Immunofluorescence **b** 3–5 mice pooled over three cohorts (*n* = 10–13 per group). **c** Representative IF images of C3-1-TAg breast tumor tissues stained with Iba1, CD3, and DAPI. Box plots quantify spatial proximity (right) (*n* = 9 slides per group). **d** Representative IF images of MMTV-PyMT breast tumor tissues stained with Iba1, CD8, and DAPI. Box plots quantify Iba1⁺ and CD8⁺ cell spatial proximity (*n* = 5–8 slides per group). **e** CyTOF data t-SNE plots compare

CD11c expression in MMTV-neu and C3-1-TAg tumors upon treatment with either Vehicle, MTD, or MCT. **f** Biaxial plots (left) compare and box plots (right) quantify the mean percentage of CD11c⁺ I-A-I-E⁺ cells among all the CD45⁺ cells identified using CyTOF. In **e**, **f** CYTOF data, *n* = 1 million events per mouse (2 mice per group). **g** Gating strategy of DC-OT-|T cells co-culture assay. **h** Box plots show percentage of CD8⁺ T cells in division phases 0 to 3+ in different treatment strategies upon co-culture with tumor-infiltrating DCs. **i** Analysis of IFN-γ in OT-| T cells after co-culture. **h**, **i** 3–5 mice pooled over two cohorts (*n* = 3 to 10 per group). **a–i** In box plots, error bar = Mean ± SEM, center = Means (Diamonds) and Medians (Line), bottom and top boundaries of the box = 25 and 75th percentiles of the data, minima and maxima = lowest and highest data points, and two-sided *T*-test *p*-values.

and endothelial genes (*Col1a2, Col1a1, Fgf7, and Fgf2*). Cluster 14 comprises cells with classic cDC gene signature (*Btla4, Siglech, Flt3l, P2ry14*, and *Irf8*). Clusters 5, 9, and 15 expressed lymphoid specific gene signature (*Cd3e, Cd3g*, and *Cd3d*). Cluster 5 hosted *Gzmb⁺ Cd8a⁺* T cells and 15 hosted Nk T cells that expressed *Fasl* and *Gzmb*.

Among the different immune cell types, myeloid clusters (transcriptionally identified) constituted nine unique Seurat Clusters (Fig. 2c, d). Clusters 0 and 4 expanded from 26% and 7.9% in Vehicle to 43.5% and 14.8% in MCT respectively upon MCT treatment (Fig. 2d). The MCT treatment led to downregulation of mitochondrial genes (*mt-Cytb*,

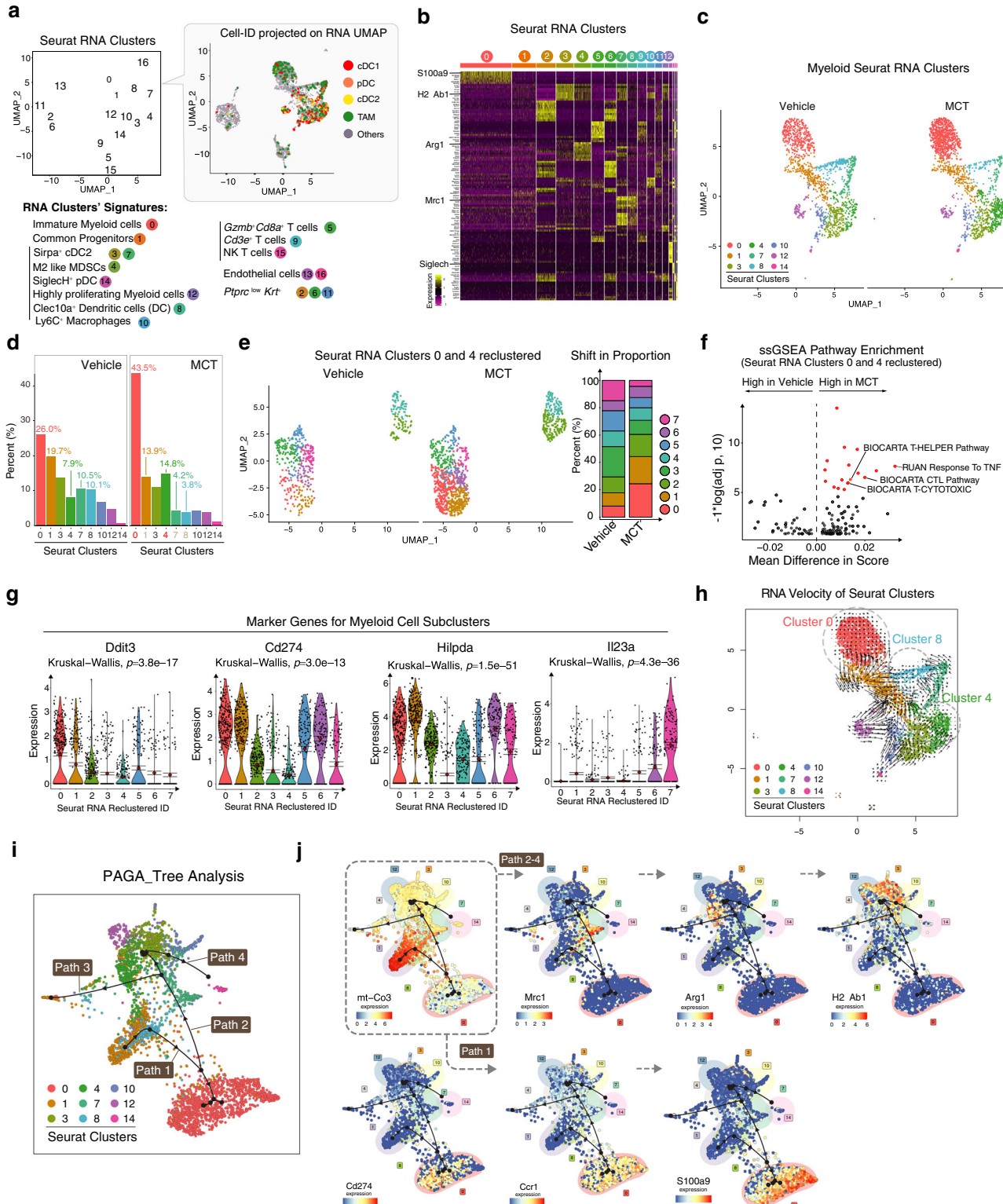

**Fig. 2 | CITE-seq reveals distinct myeloid subsets co-exist in MCT-treated TME.**
**a** UMAP clusters of TNBC derived single cells: RNA defined clusters (left) and Cell-ID
defined clusters of TADCs and TAMs (right) (in both the UMAPs: dots = cells, Cell-ID
UMAP: colored dots = TADs and TAMs, and gray dots = non TADCs and non TAMs).
**b** Heatmap shows expression of top differentially expressed genes (DEGs) among
RNA Seurat Clusters. **c** UMAP clusters of transcriptionally defined myeloid cells
split by treatment. **d** Bar plot shows (%) proportion of myeloid Seurat Clusters
projected in **c**. **e** UMAP (left) split by treatment projects reclustered Seurat Clusters
0 and 4. Stacked bar plot (right) shows shifts in the proportion of different clusters
identified after reclustering. **f** Volcano plot of differentially expressed gene path-
ways (DEGPs) between MCT and Vehicle-treated clusters in **e**. **g** Violin plots show
expression of marker genes of clusters from **e**. **h** UMAP projects RNA velocity
(length of arrow tail) of myeloid Seurat Clusters. **i** PAGA tree predicts trajectory
paths of myeloid Seurat Clusters. **j** PAGA trajectory tree projects marker genes of
trajectory clusters 7, 4, and 3 in Paths 2-4 and Clusters 0, 1 in Path 1. In **a–j**, n = cells
pooled from 3 biological repeats per treatment group. **g** In violin plots, error bar =
Mean ± SEM, center = Means (Diamonds) and Medians (Line), and Kruskal-Wallis
p-values. Also see associated Fig. S2 and Supplementary Table 2 and Data 1–4.

*mt-Nd*2, and *mt-Nd*4) and up-regulation of *Fth1, Cstb, Hilpda,* and *Cd274* in myeloid cells (Supplementary Fig. 2c). Gene set enrichment analysis showed up-regulation of ribosome, protein chain elongation, and TGFβ pathways in MCT-treated myeloid cells (Supplementary Fig. 2d and Supplementary Data 2). Next, we subsetted two clusters (Clusters 0 and 4) and found eight unique transcriptional UMAP clusters (Fig. 2e, left and Supplementary Data 3). Among these eight clusters, Clusters 0, 1 expanded while Clusters 7 proportions contracted after MCT (Fig. 2e, right). ssGSEA Pathway Enrichment analysis further showed an enriched T-cell helper, cytotoxic, and tumor necrosis factor (TNF) pathways after MCT treatment (Fig. 2f and Supplementary Data 4). Interestingly, Clusters 0 and 1 cells were marked by high expression of genes related to the immune system's response to inflammation (*Ddit3*)[18], immune checkpoint (*Cd274*), and hypoxia-induced anti-inflammatory mitochondrial metabolism process in myeloid cells (*Hilpda*)[19]. Cluster 7 cells express a high level of *Il23a*, resembling the gene signature of myeloid suppressor cells (MSDC) observed in castration-resistant prostate cancer[20](Fig. 2g and Supplementary Fig. 2e).

Coupling the RNA velocity[21] (Fig. 2h) and the PAGA tree trajectory[22] analyses (Fig. 2i, j), we further examined the evolution trajectories (cellular status transition) of myeloid Seurat RNA Clusters. Interestingly, despite the fact that Cluster 0 and 4 are the myeloid clusters which expanded the most in MCT tumors, Clusters 0, 4, and 8 had the lowest RNA velocity as depicted by the length of the arrow's tails. This suggests that these clusters are the three major transcriptome anchor points of myeloid cells co-existing in the tumor microenvironment (Fig. 2h, encircled clusters). PAGA tree analysis further delineated the specific trajectory paths that lead to the convergence of various myeloid cells into the Clusters 0, 4, and 8 (Fig. 2i). Cluster 1 (Common Progenitors) had two main future states or trajectory paths - Path 1 and Path 2 (Fig. 2i, arrow heads). Through Cluster 8, Cluster 1 evolved along Path 1 into Cluster 0 (immature MDSC-like cells expressing *Cd274, Ccr1,* and *S100a9*) or bifurcated into Path 2 (Fig. 2j). Along the trajectory Path 2, Cluster 7 cells (*Mrc1*+) emerged and then evolved into Cluster 3 (*H2-Ab1* and *Cd86* high DCs) and Cluster 4 (*Arg1* + M2-like MDSCs) (Fig. 2j). The Path 3 was an offshoot of Path 2 that then returned into a sub-portion of Cluster 1. Cluster 10 (cDC) continued evolving into *H2-Ab1*-expressing Cluster 14 (cDC2) via Path 4 (Fig. 2j). Collectively, our data revealed that tumor-infiltrating myeloid cells after MCT treatment showed highly heterogeneous transcriptome status, balancing the immune stimulatory and immunosuppressive environments in the tumor microenvironment (TME).

To strengthen the clinical relevance of our findings by which the emergence of trajectory Path 1 myeloid cells (cluster 0 cells) after chemo treatment contribute to breast cancer immune microenvironment in human breast cancer context, we performed in silico analysis using Bassez et al. dataset[23]. The goal of our analysis is two fold: (1) whether myeloid clusters change upon chemo treatment; (2) whether marker genes identified in our mouse model, e.g. trajectory Path 1 and Path 4 markers, express in myeloid cells identified from human breast tumors. First, we subsetted out myeloid cells (ITGAM^high cells) among all the immune cells (PTPRC^high and KRT8^low). UMAP based dimension reduction method clustered each single cell into four Seurat RNA clusters, 0–3 (Supplementary Fig. 2f). We observed that certain myeloid cell clusters (not all) increased upon chemo treatment. Cluster 1 and 2 myeloid cells increased after chemo from 26.7% (Cluster 1) and 5.6% (Cluster 2) in the Pre-chemo-treated tumors to 38.6% and 21.86% in the Chemo-treated cohort respectively (Supplementary Fig. 2f). Next, we examined the expression of trajectory Path 1 and Path 4 marker genes in human Cluster 1 and 2 myeloid cells (Supplementary Fig. 2g). Among these two clusters, Cluster 2 has a higher level of Mouse Path 1 (Cluster 0) marker genes (Supplementary Fig. 2g, left). On the other hand, Path 4 appears to decrease upon chemotherapy potentially due to maximum dosing (MTD), not MCT, used in the clinic (Supplementary Fig. 2g, right).

## Mapping tumor-associated myeloid cell gene regulatory network

MCT treatment led to an increase of Cluster 0 cells stopping at the end of trajectory Path 1, suggesting an intermediate cellular status between myeloid progenitors (Cluster 1 and 8) and fully differentiated lineages (Clusters 3, 7, 10, 12, and 14). First, to characterize the identity of Cluster 0 cells, we compared their gene signature with the marker genes of MDSCs[5], M1/M2 macrophages[24], Monocyte-derived DCs[25], Mature DCs, and regulatory DCs[15] (Fig. 3a). Cluster 0 cells showed minimal transcriptional overlap with above known tumor-associated myeloid cells types, except overexpressing inflammatory response genes *S100a8/9* and myeloid differentiating gene *Cebpb*, and pathogen recognizing gene *Cd14* (Fig. 3a and Supplementary Fig. 3a–h). However, classic MDSC marker genes such as *Ly6c, Ly6g,* and *Arg1* were not expressed by Cluster 0 cells. Additionally, *Ccr1* inflammatory response gene (expressed by a subset of MDSCs[26,27]) was differentially expressed by Cluster 0 cells. Next, to explore the transcriptomal programs that drive specific myeloid cell cluster's evolutionary trajectory, we performed Single-cell Regulatory Network Inference and Clustering (SCENIC, detailed in Methods, Fig. 3b Schematic)[28]. Using SCENIC analysis, we identified multiple active regulons either generally enriched to all the myeloid clusters or specific to one cluster or a subset of clusters (Fig. 3c). Notably, *Nfkb1*, known to regulate immunological responses to infections and diseases, was active in all the clusters. Cluster 0 specifically had highly active *Cebpb, Bhlhe40,* and *Atf3* that regulate cellular growth, differentiation, and immune response during stress. On the other hand, *Bcl11a* essential for plasmacytoid DC differentiation[29] was highly specific to Cluster 14. Interferon signal regulating gene *Irf8* along with general cellular growth regulators such as *Fos, Egr1,* and *Jun* were active in all the clusters with intermediate cellular state - Clusters 12, 3, 4, 10, 1, 7, and 8. Interestingly, we observed an opposite trend with *Foxp1* and *Fosl2* activity across the clusters. While *Foxp1* which is upregulated upon DC maturation was highly active in certain clusters including Cluster 14, where *Fosl2* activity was lower and vice versa (Fig. 3c, lower portion). We then projected some of these regulons with activity specific to a trajectory path/cluster on UMAP (Fig. 3d). Clearly, Cluster 0 hosted cells with highest *Cebpb* and *Fosl2*[30] activity specific to PAGA tree trajectory Path 1 while Cluster 14 hosted cells with *Bcl11a* activity. Similarly, *Foxp1* and *Irf8* were specific to trajectory Paths 2 and 4.

To dissect the regulon specificity for each of the clusters related to trajectory Path 1 and 4, we identified the specific regulons for each of the Seurat clusters along the trajectory Path 1 and 4 (Fig. 3e and Supplementary Data 5). Stat2, Sp1, and Stat1 scored the highest NES of 9.36, 9.67, and 23.6 in Clusters 1, 8, and 0 respectively. Stat1, E2f1, and Irf7 scored the highest NES of 15.9, 17.4, and 8.52 in Clusters 3, 4, and 14 respectively (Fig. 3e). To validate the predicted activation status of STAT1 regulon after MCT treatment, we co-stained C3-1-TAg and MMTV-PyMT tumors with Iba1 (myeloid marker) and pStat1 (Ser727) – an activation marker for STAT1[31] (Fig. 3f). MCT treatment expanded the Iba1+pStat1+ cells (white arrows) from a mean of 6 and 8% (Vehicle treated) to 31 and 33% of all the Iba1+ cells in C3-1-TAg (left) and MMTV-PyMT (right) respectively (Fig. 3g). Furthermore, we co-stained human TNBC breast cancer patient samples with Iba1 (myeloid marker) and pStat1 (Ser727). Compared with naive tumor, neoadjuvant ddACT chemotherapy treatment expanded the tumor-infiltrating Iba1+pStat1+ cells (white arrows) from a mean of 13% (Vehicle treated) to 25% (chemo-treated) of all the Iba1+ cells (Supplementary Fig. 3i, j and Supplementary Data 6).

## MCT-induced CXCL16 mediates TME dynamics and ICB efficacy

As Path 4 comprises diverse myeloid cell clusters, we questioned whether these myeloid clusters share any common changes upon MCT treatment. First, we grouped cells from Path 4 clusters (Seurat Cluster 3, 7, 8, and 10) and compared marker genes to Cluster 0 cells (Fig. 4a).

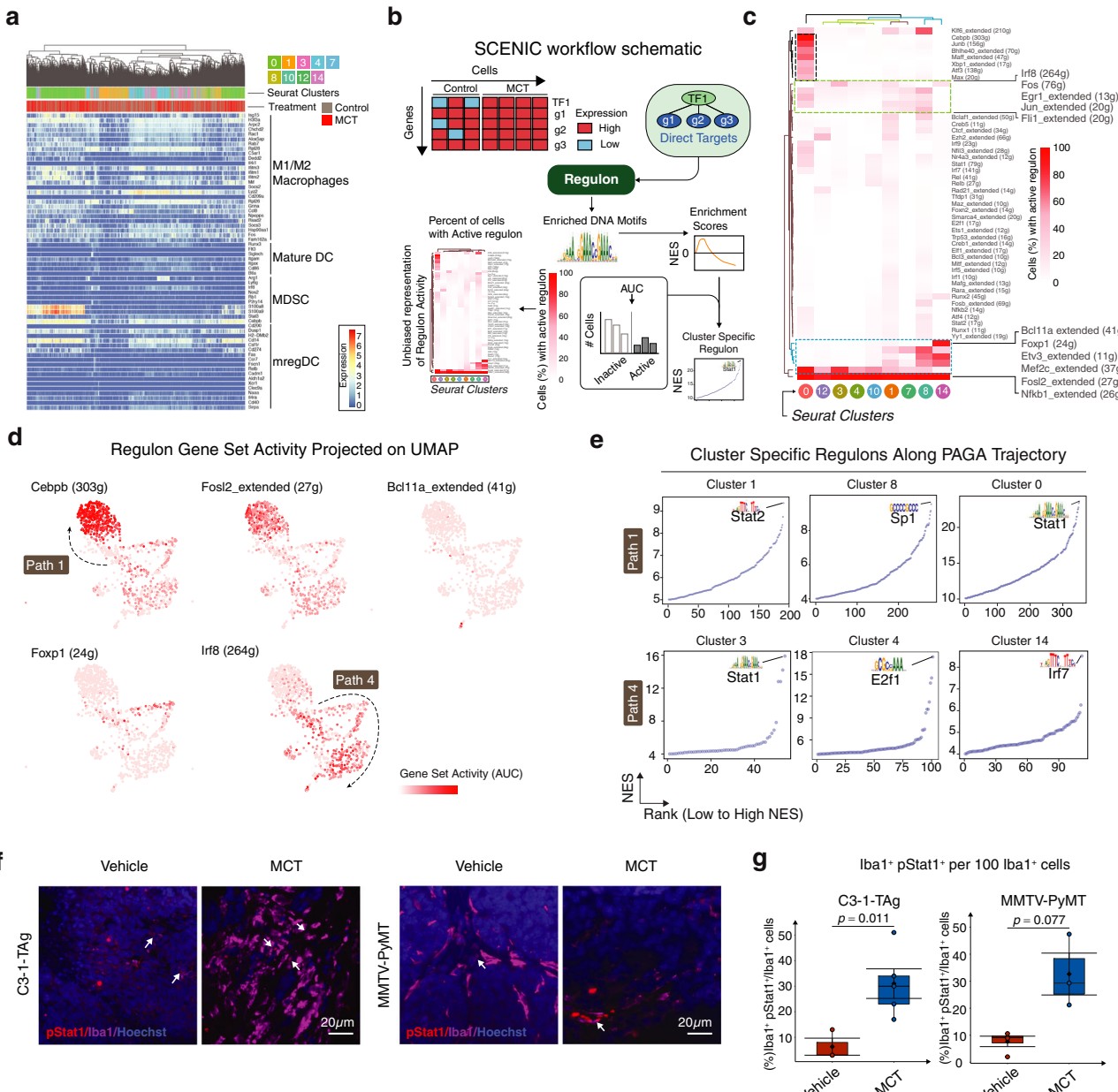

**Fig. 3 | Mapping tumor-associated myeloid cell transcriptome regulatory networks using Single-cell Regulatory Network Inference and Clustering (SCENIC). a** Heatmap shows expression of indicated cell type marker genes by myeloid Seurat clusters. **b** Schematic shows SCENIC workflow. **c** Heatmap shows percent of cells in a cluster with active regulon. **d** UMAP projects the regulon gene set activity. **e** Scatter plots show regulons specific to (top row) Clusters 1, 8, and 0 (left to right), (bottom row) Clusters 3, 4, and 14 (left to right) and their NES. Each dot is a representative of one NES (see Methods). **f** Representative IF images of murine C3- 1-TAg tissues stained with pStat1 (Ser 727), Iba1, and CD3 and MMTV-PyMT tissues with pStat1(Ser 727) and Iba1. **g** Box plot shows quantification of pStat1⁺ Iba1⁺ cells infiltrating the C3-1-TAg and MMTV-PyMT tumor (Vehicle and MCT-treated, $n = 3–5$). In box plots, error bar = Mean ± SEM, center = Means (Diamonds) and Medians (Line), bottom and top boundaries of the box = 25 and 75th percentiles of the data, (whiskers) minima and maxima = lowest and highest data points, and two-sided *T*-test *p*-values. Also see associated Fig. S3 and Supplementary Data 5 and 6.

Mitochondrial genes (*mt-Co1*), DC maturation genes (*Cd83*), and hemoglobin genes (*Hba-a1*) were highly upregulated in Path 4 clusters while immune response modulator (*Fth1*), immune checkpoint protein (*Cd274*), and anti-inflammatory gene (*Hilpda*) were higher in Cluster 0. In addition, *Cxcl12* and *Ccl3* were upregulated in Cluster 0 cells and *Ccl2*, *Cxcl10*, and *Cxcl16* were upregulated in Path 4 clusters (Fig. 4a). Interestingly, among the Path 4 cluster-specific marker genes, *Cxcl16* expression levels were significantly up-regulated upon MCT treatment (Fig. 4b, left and Supplementary Fig. 4a). Similar pattern of CXCL16 up-regulation in myeloid cells was also observed by IF staining of CXCL16 of TNBC breast cancer with ddACT regimen (Supplementary Fig. 4b).

*Cxcl16* levels are known to increase in exhausted tissue micro-environments and are correlated with high expression of interferon gamma[32]. In line with this notion, IFN receptor (*Ifnγr1* and *Ifnγr2*) expression levels were higher in Path 4 clusters cells upon MCT treatment (Fig. 4b, right).

Cxcl16 has been reported to be a chemokine attracting immune cells such as T cells[33,34], macrophages[35], neutrophils[36], and monocytes[37] in various disease contexts. To examine the potential role of myeloid-derived CXCL16 in regulating tumor infiltration of bone marrow-derived immune cells (BMD cells), we performed an intratumoral CXCL16 neutralization experiment (Fig. 4c and Methods). In this

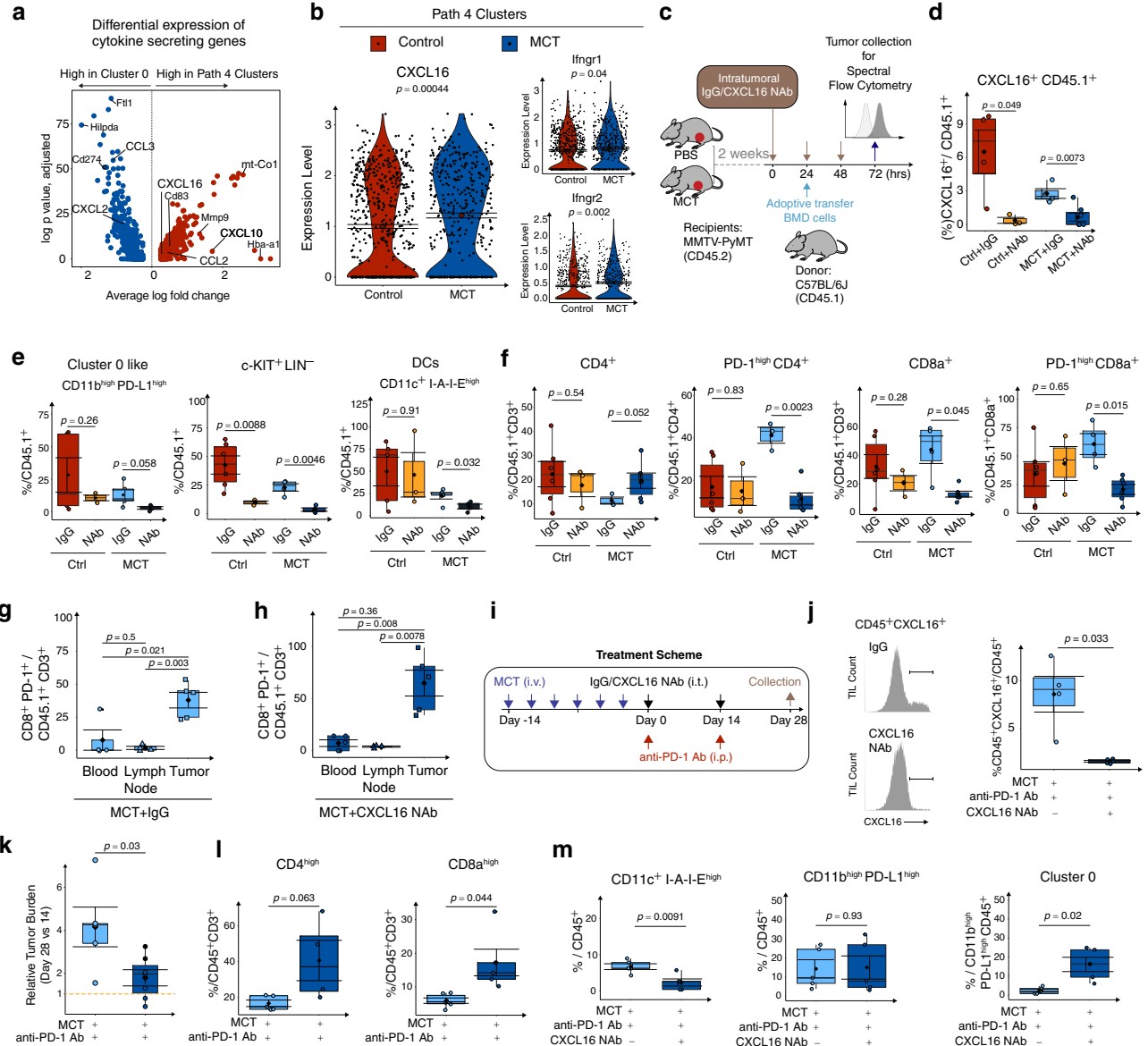

**Fig. 4 | MCT-induced CXCL16 mediates intratumoral immune dynamics and ICB efficacy. a** Volcano plot shows genes expressed differentially between Cluster 0 and DC clusters. **b** Violin plots show the expression of CXCL16, IFN-γr1, and IFN-γr2 by the DC clusters. **c** Schematic shows the intratumoral IgG or CXCL16 neutralizing antibody (NAb) injection into CD45.1⁺ cell adoptive transfer recipients treated with either PBS or MCT. **d, e** Box plots show the infiltration of CXCL16⁺ CD45.1⁺ cells, CD11b^high PD-L1^high, c-KIT⁺LIN⁻, and CD11c⁺I-A-I-E^high cells in PBS and MCT-treated tumors upon CXCL16 NAb injection. **f** Box plots show changes in CD4⁺, PD-1^high CD4⁺, CD8a⁺, and PD-1^high CD8a⁺ T cells in PBS and MCT-treated tumors upon CXCL16 NAb injection. Box plots quantify PD-1⁺CD8a⁺ cells among the CD3⁺ CD45.1⁺ cells in MCT + IgG treated blood, lymph node, and tumor (**g**), in MCT + CXCL16 NAb treated blood, lymph node, and tumor (**h**). **i** Schematic shows combinatorial

treatment with anti-PD-1 antibody and CXCL16 NAb in C3-1-TAg mice. **j** Histograms (left) show CXCL16⁺ CD45⁺ cell frequency and box plot (right) quantify CXCL16⁺ CD45⁺ cells in different treatment conditions. Box plots quantify (**k**) the tumor burden on Day 28 relative to Day 14, **l** CD4 and CD8a T cells associated with tumors, and **m** CD11c^high I-A-I-E^high, CD11b^high PD-L1^high, and Cluster 0 cells in different treatment conditions. In (**b**) violin plot, error bar = Mean ± SEM, center = Means (Diamonds) (A-B, *n* = 3 biological replicates). **d–m** In box plots, *n* = 3 to 5 biological replicates, error bar = Mean ± SEM, center = Means (Diamonds) and Medians (Line), bottom and top boundaries of the box = 25 and 75th percentiles of the data, (whiskers) minima and maxima = lowest and highest data points, and two-sided *T*-test *p*-values. Also see associated Fig. S4.

experiment, after treating the mice either with Vehicle or MCT for 2 weeks, the mice were intratumorally (i.t.) injected with three doses of either IgG or CXCL16 neutralizing antibody (NAb). On the second day of antibody injection, CD45.1⁺ BMD cells were adoptively transferred into the tumor-bearing mice (Fig. 4c). Interestingly, CXCL16 NAb injection significantly decreased the overall percentage of tumor-infiltrated CXCL16⁺ CD45.1⁺ cells in MCT-treated tumors (Fig. 4d and Supplementary Fig. 4c). Next, to explore the impact of CXCL16 NAb on lineage dynamics of tumor-infiltrating lymphocytes (TILs), we looked into percentages of CD45.1⁺ progenitor-like c-KIT⁺LIN⁻, Cluster 0-like

CD11b^high PD-L1^high, and CD11c^high I-A-I-E^high DCs (Supplementary Fig. 4d, gating strategy). CXCL16 NAb injection did not significantly alter the infiltration of CD11b^high PD-L1^high cells in PBS tumors while the decreasing trend was more apparent in the MCT-treated tumors (Fig. 4e, left). In contrast, CXCL16 NAb i.t. injection significantly decreased the infiltration of c-KIT⁺LIN⁻ CD45.1⁺ cells in both PBS and MCT-treated tumors (Fig. 4e, center). More interestingly, the infiltration of CD11c⁺ I-A-I-E^high DC into the MCT-treated tumors (Fig. 4e, right) but not in PBS-treated, also significantly reduced after CXCL16 NAb i.t. injection. Above results suggest the CXCL16 in the TME has a more prominent

role in attracting progenitor-like c-KIT+LIN- myeloid cells as well as peripheral DCs into the MCT-treated tumors.

CXCL16+ DCs are known to interact with T cells, especially CXCR6+ T cells[38]. Hence, we examined the T-cell infiltration into the tumors upon CXCL16 neutralization.

CXCL16 NAb did not alter the infiltration of CD4+ T cells in the PBS-treated tumors but increased in MCT-treated tumors (Fig. 4f, left). Among these CD4+ T cells, CXCL16 NAb significantly decreased PD-1high CD4+ T cells in MCT-treated tumors (41% in IgG to 11% in NAb) (Fig. 4f, middle-left). In addition, CXCL16 NAb more significantly reduced the CD8a+ T cells infiltration in the MCT-treated tumors (44.6% in IgG and 12% in NAb group), but not PBS-treated tumors (Fig. 4f, middle-right). Similarly, among these CD8a+ T cells, CXCL16 NAb reduced the PD-1high CD8+ T-cell infiltration specially in MCT-treated tumors (57% in IgG to 20.5% in NAb group) (Fig. 4f, right). CXCL16 NAb did not significantly alter the infiltration pattern of CD4+ CXCR6high or CD8+ CXCR6high cells in MCT-treated tumors (Supplementary Fig. 4f). Collectively, our results suggest that the increase of CXCL16, mostly derived from Path 4 clusters cells, lead to an increase of PD-1high T cells in the MCT-treated tumors.

To trace the source of the exhausted PD-1high T cells, we cross compared PD-1+ CD8a+ CD45.1+ T cells in blood, tumor-draining lymph nodes, and tumor tissues (Fig. 4g, h). We observed that in IgG treated mice (Fig. 4g), CD45.1+ PD-1+ CD8+ T-cell percentage was 8% (in blood), 2% (in lymph nodes), and 38% (in tumor tissues). In CXCL16 NAb treated mice (Fig. 4h), CD45.1+ PD-1+ CD8+ T-cell percentage was 7%, 4%, and 65% respectively. There are significantly higher percentage adoptively transferred CD45.1+ T cells that showed an exhaustion phenotype in the tumor compared with either blood or lymph. This observation suggests a model of T-cell exhaustion after entering the tumor microenvironment.

Previous studies showed CXCL16 blockage has therapeutic potential in curbing thyroid tumor growth[35]. To investigate the therapeutic potential of combining immunotherapy and CXCL16 blockage, we combined PD-1 checkpoint inhibitor with CXCL16 NAb (Fig. 4i, Treatment scheme). The C3-1-TAg TNBC primary tumor-bearing mice were primed with MCT for two weeks and then treated with anti-PD-1 antibody in combination with either IgG or CXCL16 NAb. The mean percentages of tumor-associated CXCL16+ CD45+ cells among all the CD45+ cells decreased from 8.5% in IgG to 2.1% in CXCL16 NAb group (Fig. 4j). Intriguingly, the mean relative tumor volume significantly decreased from 4 in IgG to 1.8 with combo treatment of CXCL16 NAb and anti-PD-1 antibody compared with anti-PD-1 alone (Fig. 4k and Supplementary Fig. 4g). Next, we examined TME dynamics under CXCL16 NAb treatment, including CD4+ and CD8+ T cells, CD11chigh I-A-I-Ehigh DCs, CD11bhighPD-L1high, and Cluster 0 cells (Supplementary Fig. 4h-i, gating strategy). Both CD4high T (16.4–40.5%) and CD8high (5–17%) T cells significantly expanded upon combo treatment compared with anti-PD-1 single treatment (Fig. 4l) partially explains the reduced tumor size. Upon CXCL16 NAb treatment, the CD11c+ I-A-I-Ehigh DCs decreased (10% in IgG to 3% in CXCL16 NAb). Moreover, while the percentage of CD11bhighPD-L1high cells among CD45+ cells did not change, treatment of CXCL16 NAb led to an increase of Cluster 0 cells under MCT plus anti-PD-1 dual treatment (Fig. 4m, right), suggesting a shift in homeostasis of the trajectory towards Path 1 Cluster 0 (Cd274+, Cd14+, S1008/9+) compensatory to the inhibition of CXCL16 signaling.

## STAT1 regulates PD-L1 expression in Cluster 0 cells

To examine whether Cluster 0 cells could suppress the T-cell activity, we performed an in vitro co-culture assay with MCT-tumor-infiltrating Cluster 0 cells with T cells activation beads. To enrich the Cluster 0 cells from tumor-infiltrating immune cells, we first examined the top 20 differentially up-regulated genes (Supplementary Data 1, Cluster 0 marker tab). CITE-sequencing data indicates Ccr1, PD-L1, Csf3r, Msrb1, Ccl3, Clec4d, and S100a are among differentially expressed genes by

Cluster 0 cells. Among all these DE genes, Cd14, Ccr1, and PD-L1 encode cell surface proteins which could be used to sort cells fluorescently tagged antibodies recognizing above cell surface proteins. We found that both Cluster 0 cells and Ccr1high cells have a high degree of transcriptome similarities (Fig. 5a, b). While Ccr1 expression on Cluster 0 is modest, we reasoned that sorting out Ccr1high cell population using magnet-based method will result in a population largely enriched with Cluster 0 cells. Of all the CD45+ tumor-infiltrating immune cells, about 10.7% were CD11b+CCR1high (Fig. 5c, biaxial plot). 74.4% of naive CD4+ T cells proliferated more than once when co-cultured with T-cell activation beads and IL2 (Fig. 5d, left box plot, gating shown in Supplementary Fig. 5a). In the presence of CD11b+CCR1high myeloid cells, the percentage of proliferated CD4+ T cells did not change (mean of 74.2%). However, in the presence of CCR1low myeloid cells in the co-culture, the percentage of proliferated CD4+ T cells significantly decreased (mean of 28.9%). Similarly, 54.7% of naive CD8+ T cells proliferated more than once when co-cultured with T-cell activation beads and IL2. While the presence of CD11b+CCR1high myeloid cells did not change (mean of 78.6%) their proliferation, CCR1low myeloid cells decreased the naive CD8+ T-cell proliferation significantly (mean of 23.6%) (Fig. 5d, right box plot). This observation suggested that CCR1high (Cluster 0) myeloid cells do not have MDSC-like properties which inhibit T-cell proliferation. Instead, co-culture with CCR1high myeloid cells led to a significant increase in the ratio of proliferating to non-proliferating PD-1high CD4+ or CD8+ T cells (Fig. 5e). These results suggest Cluster 0 cell pushes T cell toward hyper-proliferative PD-1high phenotype which might suggest a dysfunctional status of T cells[39]. To further comprehend the transcriptome of tumor-infiltrating T cells, we identified CITE-CD45+ and CITE-CD3+ TIL cells from our CITE-seq data and clustered them into 10 Seurat Clusters, of which 1 and 2 showed a high expression of Cd3e. There are three major clusters of T cells, Cd8a+ T cells (Cluster 0), Cd4+ T cells (Cluster 1), and Foxp3+ CD4+ T cells (Cluster 2). We plotted a heatmap for top differentially expressed genes between different T-cell clusters (Supplementary Fig. 5b, left). Cluster 0 T cells expressed CD8. Thus, we focused our analysis on this cluster. Next, we compared gene expression between control of MCT treatment in each T-cell cluster (Supplementary Fig. 5b, right). PD-1, Lag3, and CTLA4 were differentially increased upon MCT treatment in Cluster 0 T cells. We also plotted markers for precursors of exhausted T (Tpex) cells, including Tcf7, Sell (encodes CD62L), Myb[40]. While Tpex markers are only moderately expressed in Cluster 0T cells, we observed an overall higher expression in MCT-treated group, correlating with T-cell exhaustion markers (Supplementary Fig. 5b, right). Above gene signatures analyses collectively suggested that MCT treatment reshapes a subcluster of tumor-infiltrating CD8+ T cells towards enrichment of Tpex phenotype.

We next analyzed Cluster 0 specific DGE (differential gene expression), DGEP (differential gene expression pathway), and Stat1 targetome. Upon MCT treatment, Cluster 0 cells up-regulated marker genes related to innate immunity (Gna13) and immune suppression (Cd274)[41] (Fig. 5f and Supplementary Fig. 5c). Furthermore, GSEA analysis of Cluster 0 specific DGE correlated with a downregulation of adaptive immune system pathways, including CD8+ T-cell activity, and dendritic cell maturation (Supplementary Fig. 5d). Cluster 0 cells also up-regulated expression of GPCR ligand binding and GPCR activation pathways that are critical for immune cell functionality[42] and TGFβ pathway, known to create an immune suppressive TME[43] (Supplementary Fig. 5d). Since STAT1 is the major transcription factor that drives Cluster 0 transcriptome (Fig. 3e, SCENIC analysis), we further examined potential gene targets of STAT1 (Targetome) that are differentially expressed by Cluster 0 (Fig. 5g). SCENIC based gene regulatory network analysis (GRN)[28] correlated more than 200 genes (NES > 3) under STAT1 targetome (Supplementary Data 7). Interestingly, in addition to Cd274, Cluster 0 specific STAT1 targetome also contains genes that are immune-related, including Tspo, Ifitm3, Cd164,

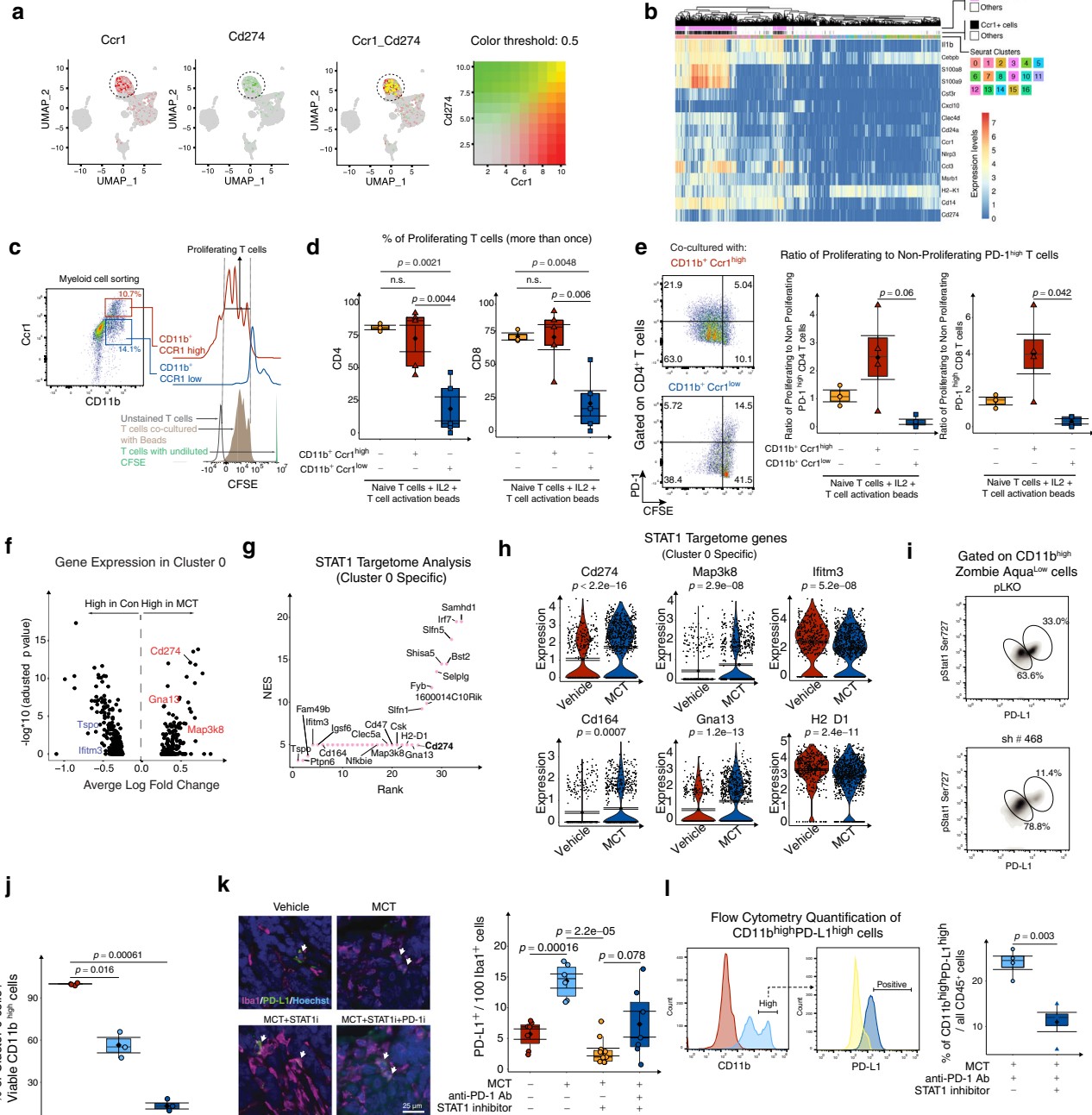

**Fig. 5 | STAT1 regulates PD-L1 expression in Cluster 0 cells and influences T-cell activation status. a** Dual plot shows expression of Ccr1 and Cd274 in immune cells. **b** Heatmap shows the marker genes of Cluster 0 and compares their expression between Ccr1+ cells, Cluster 0, and other myeloid cell subsets. **c** Gating strategy to identify FACS sorted CD11b^high CCR1^high, CD11b^high CCR1^low myeloid cells, and CFSE levels on CD8+ T cells in co-culture. **d** Box plots show percentage of proliferating CD4 and CD8 T cells in co-culture with T-cell activation beads with or without myeloid cells. **e** Biaxial plots (left) show the gating strategy to identify the PD-1 levels on CFSE diluted and proliferating T cells. Box plots (right) show the ratio of PD-1^high proliferating and non-proliferating CD4 and CD8 T cells. **f** Volcano plot shows DGE on Cluster 0 cells when MCT-treated. **g** Scatter plot shows the potential gene targets (NES > 3 and p < 0.05 DEG between MCT and Vehicle treated) in Cluster 0 specific STAT1 targetome. **h** Violin plots show expression of Cd274, Map3k8, Ifitm3, Cd164, Gna13, and H2-D1 genes by Cluster 0 cells (each dot is a cell and cells are from n = 3 biological repeats pooled together). **i** Biaxial plots show the

levels of pStat1 in CD11b+ cells in different transfection conditions. **j** Box plot shows the percentage of pStat1+ PD-L1+ CD11b+ viable cells after shRNA STAT1 transfection relative to pLKO transfection (n = 5 biological repeats). **k** Representative IF images of Iba1 and PD-L1 stained tumors treated with either Vehicle, MCT, MCT + STAT1 inhibitor, or MCT + STAT1 inhibitor + PD-1 inhibitor (left). Box plot (right) shows the quantified Iba1+ PD-L1 + cells of all the Iba1+ cells (n = 3 to 5 biological repeats). **l** Histograms show flow cytometry gating strategy and box plots show quantification of PD-L1+ CD11b+ cells of all the CD45+ immune cells in dual-therapy (MCT + PD-1 inhibitor) and tri-therapy (MCT + PD-1 inhibitor + STAT1 inhibitor) treated TNBC tumors (n = 4 biological repeats). **a**, **b**, **f**, **g** n = 3 biological replicates. **c**–**l** In box and violin plots, error bar = Mean ± SEM, center = Means (Diamonds) and Medians (Line), bottom and top boundaries of the box = 25 and 75th percentiles of the data, (whiskers) minima and maxima = lowest and highest data points, and two-sided T-test p-values. Also see associated Fig. S5.

*Clec5a*, *H2-D1*, *Gna13*, and *Irf7* (Fig. 5g). MCT significantly increased the expression of a T-cell co-inhibitory signal, *Cd274*[41], Map kinase protein *Map3k8*, metastasis promoter, *Cd164*[44], and *Gna13* G protein[45] (Fig. 5h). Expression of interferon-regulatory genes *Ifitm3*[46], histocompatibility antigen *H2-D1*, innate immune-related *Clec5a*, immunomodulatory genes *Cd47*, *Tspo*, and *Irf7* decreased with MCT treatment (Fig. 5h and Supplementary Fig. 5e). Next, we validated *STAT1*'s role in regulating tumor-associated PD-L1⁺ myeloid cells by knocking down *STAT1* in tumor-associated CD11b⁺ cells. Tumor-bearing C3-1-TAg mice were treated with MCT for 28 days. Tumor-associated CD11b⁺ cells were then transfected with lentivirus carrying either pLKO shRNA control or *STAT1* targeting shRNAs (Methods section). The addition of Stat1 shRNA did not significantly change the viability of the singlets (Supplementary Fig. 5f, g). Compared with the pLKO control shRNA group, shRNA *STAT1* led to a decrease in percentage of CD11b⁺ cells with high phospho-Stat1 (pStat1) levels (Fig. 5i and Supplementary Fig. 5f, gating strategy). The viable Cluster 0 cell number decreased significantly upon *Stat1* KD to 56% in sh468 construct and to 19% in sh2253 construct relatively when compared to the 100% in pLKO group (Fig. 5j). Furthermore, multigene association analysis of TCGA and GTex projects' RNA-seq data showed a stronger association between *STAT1* and *Cd274* expression in the invasive breast carcinoma ($R = 0.72$) compared with the correlative pattern in healthy mammary gland tissue ($R = 0.3$). Similarly, the association between *STAT1* and *Cd274* was stronger in cancerous tissues in multiple cancer types (Supplementary Fig. 5h).

Given the close association between PD-L1 levels and *STAT1* (phosphorylation) levels in Cluster 0 cells, we treated mice with a combination of *STAT1* inhibitor Fludarabine[47] with or without PD-1 inhibitor along with MCT. PBS-treated tumors served as a control group for MCT-treated tumors. Immunofluorescence results suggested that out of every 100 Iba1⁺ cells the frequency of PD-L1⁺ Iba1⁺ cells increased from 5% in Vehicle-treated tumors to 14% in MCT (Fig. 5k). Furthermore, combining STAT1 inhibitor with MCT decreased their frequency to 3% and combining STAT1 and PD-1 inhibitors with MCT decreased the frequency to 10% (Fig. 5k, box plot). Flow cytometry results suggested that among all the CD45⁺ cells, the mean frequency of PD-L1⁺ CD11b⁺ cells significantly decreased from 24% to 11% when STAT1 inhibitor was added to the regimen (Fig. 5l).

## Modulating STAT1 enhances anti-tumor responses of anti-PD-1

To examine the impact of PD-1 inhibitor on tumor growth, we treated MMTV-neu and C3-1-TAg spontaneous tumor-bearing mice with PD-1 antibody after two weeks of MCT treatment (Fig. 6a, Treatment Regimen 1). While MCT treatment significantly reduced tumor size, the subsequent anti-PD-1 treatment alone was not sufficient to further enhance the antitumor efficacy of MCT in both tumor models (Fig. 6b). Next, we designed a sequential treatment regimen of CXCL16 NAb and STAT1 inhibitor to test possible enhanced efficacy of sensitizing MCT-primed tumor to anti-PD-1 treatment. The C3-1-TAg mice bearing primary breast tumors were MCT-primed for 2 weeks and then treated with anti-PD-1 antibody in combination with either IgG or CXCL16 NAb prior to STAT1 inhibitor treatment (Fig. 6c, Treatment Regimen 2 and Supplementary Fig. 6a). However, one week treatment of CXCL16 NAb prior to STAT1 inhibitor (Treatment Regimen 2, Seq) was only able to maintain a stable disease over the course of treatment (Fig. 6d). The potential increase of Cluster 0 cells after CXCL16 NAb might counteract the overall anti-cancer response. Thus, we further hypothesized that targeting STAT1 right after MCT-priming could potentially improve anti-PD1 treatment efficacy (Fig. 6e, Treatment Regimen 3). Compared with the baseline efficacy (MCT + anti-PD1, Dual), adding STAT1 inhibitor (Tri-therapy, Tri) significantly improved anti-tumor efficacy of anti-PD1 treatment (Dual-therapy, Dual) (Fig. 6f, $p = 0.0098$) with an increased percentage of tumor-associated T cells (Fig. 6g and Supplementary Fig. 6b) when compared to Dual therapy used in

regimen 3. Furthermore, the DC percentage (Fig. 6h and Supplementary Fig. 6b) increased and Cluster 0 cells (CD11b⁺ PD-L1⁺ CCR1⁺, Fig. 6h and Supplementary Fig. 6c) percentage decreased in Tri-therapy (Tri) used in regimen 3 when compared to Sequential (Seq) treatment used in regimen 2. This suggests the potential shift of myeloid cell trajectory Path 1 (Cluster 0 dominated) to Path 4 (DC dominated) when applying the Tri-therapy (Tri) scheme.

Overall, the tri-therapy improved the PFS when compared to other strategies in either of the regimens (Fig. 6i, Pink line). The H-score of the Ki-67 nuclear stain decreased most significantly from 120 in dual-therapy to 72 in tri-therapy (regimen 3) treated tumors (Fig. 6j).

## Discussion

MCT regimen was originally conceived as an alternative chemotherapy regimen to MTD with potential anti-angiogenic benefits[48]. In addition to the expected anti-angiogenic effect, MCT stimulates multiple anti-tumorigenic immune cell subsets in the context of breast cancer[49]. Low-dose chemotherapy has been demonstrated to have immune stimulatory properties in the TONIC metastatic TNBC clinical trial and improved overall response rate in 20% (35% of Doxorubicin treated) of participants to PD-1/PD-L1 blockade[50]. Despite a generally low response rate of breast cancer to ICB, this study highlighted the clinical significance of using chemotherapy as a "primer" for immune check-point blockades even in the late-stage metastatic setting. However, while the improvement seen in the TONIC study is encouraging, overall anti-tumor efficacy remains limited. A large portion of patients were still not responsive to anti-PD1 ICB treatment[50]. In our triple-negative breast cancer preclinical setting, addition of anti-PD-1 immunotherapy to MCT regimen was only marginally improved anti-tumor efficacy (Fig. 6a, b, regimen 1), suggesting other potential immune suppressants within the TIME that restrict the full potential of ICB's efficacy. To reveal this dynamic equilibrium of TIME after MCT, we used CITE-seq and trajectory analysis to map the single-cell immune landscape of breast tumor after low-dose MCT-priming (Fig. 2). It is increasingly appreciated, besides tumor-T-cell direct engagement through immune checkpoint molecules[51], that the maximal benefits of ICB treatment requires T-DC crosstalk (a "licensing" model)[4]. Tumor-infiltration by DCs also correlates with immunotherapeutic efficacy[2,4]. T-DC crosstalk sensitizes the tumor to ICBs[4] and ensures durable responses in melanoma and lung cancer[4,52]. Our analysis showed that MCT enhances T and myeloid cell spatial proximity (Fig. 1c–e), expands antigen presenting cDCs (Fig. 1g, h). However, MCT-priming also parallelly enables the evolution of a subset of paradoxical myeloid cells (Cluster 0 cells), expressing both immune-stimulating and immune-suppressive transcriptomal profile (Figs. 2b and 3a and Supplementary 3a–h, 5d, e). In contrast to an over-simplified binary models of either immune stimulating or suppressive tumor microenvironment, this co-existence of antigen presenting and immuno-suppressive myeloid cells underscores the notion of de novo immune homeostasis achieved in MCT-primed TIME, highlighting a resilient nature of TIME to re-establish homeostasis in response to the systemic stimulation – a tenet in biology at the organismal level[53].

CXCL16 has been shown to improve tissue tolerance, and recruit T cells in various physiological and pathological contexts[33,36,54,55]. In ovarian, thyroid, and lung cancers CXCL16 promotes tumor progression[32,35,56]. Immunogenic chemotherapy promoted CAR-T-cell recruitment by increased CXCL16 secretion in lung cancer[56]. Furthermore, CXCL16-CXCR6 interaction promoted tolerance to cardiac allograft transplants and successful fetal development[54,55]. In our study, we observed that MCT treatment increased CXCL16 in the DCl specific trajectory paths (Fig. 4). Neutralization of CXCL16, resulted in decreased tumor-infiltration of immune cells, specifically CD45⁺ CD11b⁺ PD-L1^high cells, c-Kit⁺Lin⁻ progenitor cells, PD-1^high T cells (Fig. 4d-f). Reducing the recruitment of peripheral PD1^high and PD-L1^high

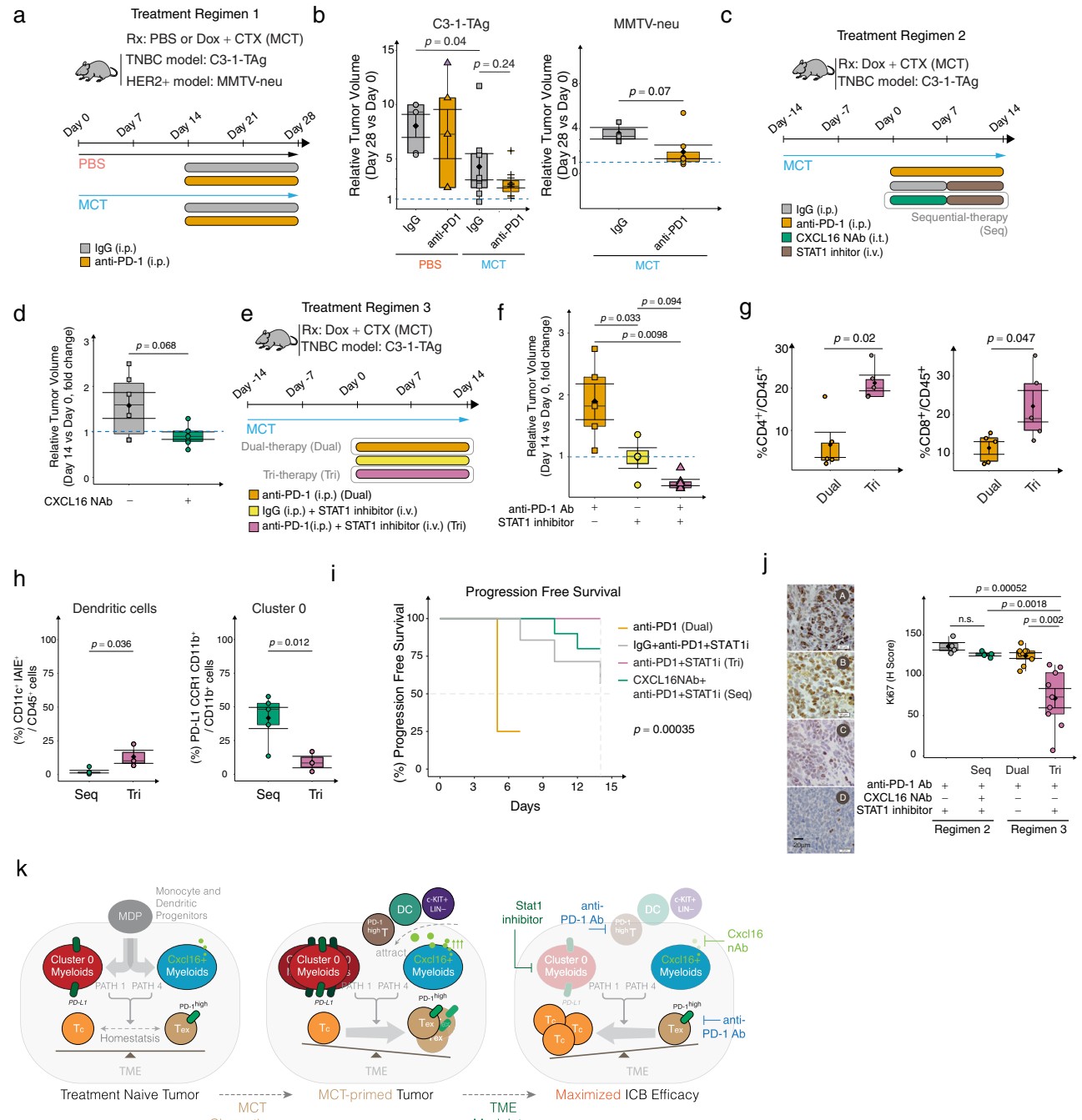

**Fig. 6 | Modulating STAT1 signaling enhances anti-PD-1 antibody mediated anti-tumor responses. a** Treatment scheme for synergizing anti-PD-1 with MCT (Regimen 1). **b** Box plots show tumor volumes on Day 28 of treatment (Regimen 1) relative to Day 0 in C3-1-TAg (left) and MMTV-neu (right) mice. *n* = 3–5 mice pooled over three cohorts. **c** Treatment scheme for synergizing CXCL16 NAb with STAT1i response in MCT and anti-PD-1 treated mice (Regimen 2). **d** Box plots show tumor volumes on Day 14 of treatment (Regimen 2) relative to Day 0 in C3-1-TAg mice. *n* = 3–5 mice pooled over three cohorts. **e** Treatment scheme for synergizing STAT1i and anti-PD-1 antibody response (Regimen 3). **f** Box plots show tumor volumes on Day 14 of treatment (Regimen 3) relative to Day 0 in C3-1-TAg mice. *n* = 3–5 mice pooled over three cohorts. Box plots quantify flow cytometry identified **g** T cells in Dual and Tri-therapy and **h** dendritic cells and Cluster 0 cells in Seq-therapy and Tri-

therapy. **i** Progression-free survival probability in different treatment regimens showing time taken for the tumor volume to increase from Day 0 of treatment to Day 14. **j** Representative IHC images of Ki-67 stained TNBC tissues treated with either Regimen 2 or 3 (left). Box plot (right) quantifies H-Score (*n* = 3–5 biological replicates per group). **a–j** In box plots, *n* = 3–5 biological replicates, error bar = Mean ± SEM, center = Means (Diamonds) and Medians (Line), bottom and top boundaries of the box = 25 and 75th percentiles of the data, (whiskers) minima and maxima = lowest and highest data points, and two-sided *T*-test *p*-values. Survival plot was generated using the Survimer package in R. **k** Schematic shows the proposed-model of MCT-mediated immune modulation in TNBC. Also see associated Fig. S6.

---

cells to TIME partially explains the improvement of the efficacy of anti-PD-1 antibody upon intratumoral neutralization of CXCL16 (Fig. 4i, j). However, the tumor continues to progress (Fig. 4i, Relative Tumor Burden >1.5 fold increase, Day 28 vs Day 14), potentially due to a

significant increase of Cluster 0 cells after CXCL16 neutralization (Fig. 4k), suggesting the dynamic nature of the TIME in response to immune modulator CXCL16. Therefore, in order to maximize the ICB's anti-cancer efficacy, a holistic and dynamic view of TIME is needed.

Immune cell plasticity drives both anti- and pro-tumor immunity to maintain the TIME homeostasis. Immune plasticity, exemplified in the highly plastic myeloid cells, could be leveraged to improve the efficacy of anticancer therapies[53]. Through tumor-associated myeloid cell subset trajectory inference and SCENIC regulon analyses, our study revealed the emergence of immune deterring myeloid cluster 0 after MCT, which is transcriptionally driven by *STAT1* regulon. Interestingly, it has been shown that eIF4F-regulated translation of *STAT1* mRNA is a critical step of IFN-γ-induced PD-L1 expression on melanoma tumor cells, partially contributing to tumor immune evasion mechanisms[57], suggesting a translational significance of modulating eIF4F-STAT1 pathway as a potential cancer immunotherapeutic modality in melanoma patients[57]. Complementary to Cerezo et al. study, our study further demonstrated STAT1 signaling as a dominant regulon driving the tumor-infiltrated immature myeloid cells (Cluster 0) after MCT-priming (Fig. 3c–e). The role of STAT1 signaling in TME has been previously suggested in Renca monoclonal cancer model, where a high expression of inflammatory gene signature driven by STAT1 signaling in TIME is correlated with the tumors' responsiveness to ICBs[58]. In the MCT-primed breast cancer context, we observed similar inflammatory signatures in TIME cells (Fig. 3). However, despite an enriched inflammatory signature in the TME, MCT-priming only marginally improved the response of breast cancer to anti-PD-1 treatment (Fig. 6b). Our single-cell trajectory analysis shed the light on the root cause of this suboptimal outcome and demonstrated an intrinsic dual-role of chemotherapy priming - inducing a potentially immunogenic inflammatory signature while concurrently stimulating *STAT1* regulon in TIME to facilitate local immune tolerance through upregulating immunosuppressants due to CXCL16^high cells (Fig. 4) and Cluster 0 immature myeloid cells (Fig. 5). Such MCT-induced tolerance imposed by CXCL16^high cells and Cluster 0 myeloid cells partially explains the only subtle improvement of therapeutic efficacy seen on combining MCT with anti-PD-1 (Fig. 6b), highlighting the nature of local immune equilibrium regulated by collective factors. Therefore, modulating *STAT1* - a master regulator of collective immunoregulatory factors, using *STAT1* inhibitor in MCT-primed breast tumor effectively shifted the immune homeostasis in the favor of immunogenicity, which resulted in a much improved response to ICB (Fig. 6f, j).

In summary, enabled by multimodal single-cell analysis, we uncovered the pro-tumoral immune suppressive myeloid cells within the MCT-primed tumor microenvironment. Our study provided mechanistic insights and preclinical rationale for modulating the *STAT1* regulon as a neoadjuvant regimen for MCT-primed breast tumors to achieve the maximum clinical benefit of the anti-PD1 immunotherapy.

## Methods

### Ethical approval statement
All the mouse experiments were performed as per University of Notre Dame (Protocol # 18-05-4687) and UT Southwestern (Protocol #2022-103340) Institutional Animal Care and Use Committee (IACUC) approved protocols. The study with TNBC clinical specimens in this publication was approved by the UT Southwestern Institutional Review Board (IRB# STU102010-051).

### Mice
To model the clinical scenario, we used spontaneous tumor-bearing immunocompetent mouse strains, MMTV-neu, MMTV-PyMT, and C3-1-TAg, representing HER2 overexpressing, adenocarcinoma, and TNBC in clinic respectively. For MMTV-neu, FVB/N-Tg(MMTVneu)202Mul/J (Jax Stock, 002376) homozygous males were crossed with homozygous females. For spontaneous tumor developing MMTV-PyMT B6 mice, FVB/N-Tg(MMTV-PyVT)634Mul/J carrying MMTV-LTR driving the mammary gland specific Polyoma middle T antigen is backcrossed to C57BL/6 J (Jax Stock, 022974). For spontaneous TNBC tumor

development, C3-1-TAg males hemizygous for Tg(C3-1-TAg) (Jax Stock, 013591) were crossed with noncarrier (FVB) females. For non-tumor-bearing C3-1-TAg mice, C3(1)/Tag-REAR (Jax Stock, 030386) mice (Hemizygous crossed with noncarrier) were used as tumor transplantation recipients as well as used for peripheral blood analyses. All mice were bought from the Jackson Laboratory and bred at Freiman Life Sciences Center, University of Notre Dame or at UT Southwestern Medical Center. Female mice aged between 3 to 8 months were used for all the experiments. Our study was carried out in a specific pathogen-free/SPF facility. All the experiments were performed as per University of Notre Dame's Institutional Animal Care and Use Committee (IACUC) approved protocols. The mice were provided with food and water 24 h. They were maintained on a 12 h day and 12 h night schedule. Mice were euthanased by carbon dioxide inhalation using the Euthanex system, followed by cervical dislocation. The maximum tumor size (allowed) 1.5 cm was not exceeded. Both the experimental and control mice cohorts were age and cage matched and assigned to different treatment groups randomly.

### Drugs and treatment dosages
Doxorubicin (DOX, MedChem Express, HY-15142) and Cyclophosphamide (CTX, MedChem Express, HY-17420) were used in the chemotherapy cocktail. The clinically used doses of DOX and CTX were converted from human to mouse equivalents using FDA-approved formula, HED (mg/kg) = Animal Dose (mg/kg) × 0.08. Chemotherapy cocktail was administered intravenously either once a week (MTD) or thrice a week (once every two and a half days) (MCT) for a period of 4 weeks. The dosage of DOX used was 2 mg/kg body weight and CTX 20 mg/kg body weight for the MTD cohort and a third of the MTD dose for the MCT cohort. For a PD-1 inhibitor treatment trial, both PD-1 inhibitor (BioXcell, BE0146) and IgG (BioXcell, BE0087) antibodies were injected intraperitoneally once a week at a dose of 150 μg per mouse. The PD-1 inhibitor was dosed at 6 mg/kg body weight, which is a mouse equivalent dose of PD-1 inhibitor clinical human dose. Stat1 inhibitor (MedChem Express, HY-B0028) was given at 8.4 mg/kg body weight (derived from clinically used human doses and calculated as per mouse equivalents) dose in MCT strategy intravenously along with MCT as a cocktail prepared right before administration.

### Single-cell isolation for CyTOF and CITE-seq analysis
Commercially available Collagenase Type 1 enzyme powder (Thermo Fisher Scientific, 17100-017) dissolved in DMEM High Glucose Medium (1% Penicillin/Streptomycin and 10% Fetal Bovine Serum) (Thermo Fisher Scientific, 11965092) was used to digest the tumors. The tumors were enzymatically digested for 60 min after mincing. Live cells were enriched using Ficoll solution (GE Healthcare, 17-5446-02). For density gradient based enrichment of cells, 4 mL of cell suspension was added gently as a layer on the top of 3 mL of Ficoll Paque solution in a 15 mL conical tube. The conical tube was centrifuged at $500 \times g$ with the brake off at 18 °C. All the cell layers were collected. The pellets with blood cells and granulocytes were subjected to red blood cell lysis and granulocytes separated. Red blood cells (RBCs) were lysed using the ACK Lysis buffer (Lonza, 10-548E). Dead cell removal kit (Miltenyi Biotec, 130-090-101) was used to remove dead cells. CD45 magnetic beads (Miltenyi Biotec, 130-052-301) and or CD11b magnetic beads (BioLegend, 480109) were used to isolate the CD45$^+$ and CD11b$^+$ cells. For isolating splenocytes, the spleen was mashed in heat-inactivated RPMI medium. RBCs were lysed in 1× red blood cell lysis buffer, ACK Lysis Buffer (Lonza 10-548E). Either CD45 magnetic beads or T-cell isolation kit (Miltenyi Biotec, 130-095-130 or BioLegend, 480024) was used to isolate cells of interest. Cells were incubated at 4 °C to maintain their viability and passed through a 40 μm cell strainer (VWR, 76327-098) to dissociate single cells from clumps. Single cells were maintained viable and separate in buffers with 5% Bovine Serum Albumin (BSA) and 2 mM Ethylene Diamine Tetra Acetic Acid (EDTA).

### OT-I plasmid cloning and transfection

An endoplasmic reticulum targeting SIINFEKL gene fragment was synthesized from Twist Biosciences. The fragment was then PCR-amplified using primers flanked by *attB1* and *attB2* recombination sequences. The resulting amplicon was cleaned up with DpnI treatment followed by 10% PEG, 10 mM $MgCl_2$ precipitation. For Gateway cloning, the attB1-ER-SIINFEKL-attB2 product was reacted with pDONR-CBX (Addgene # 29634) using BP Clonase II (Thermo Scientific, 11789020) followed by Proteinase K treatment. 2 μL of reaction mix was transformed into Lucigen Endura chemically competent *E. Coli* (Lucigen, 60240-2) and plated on an LB plate with Kanamycin. Colonies containing the entry vector were miniprepped using the Monarch™ Plasmid Miniprep Kit (NEB, T1010S). Next, the entry vector was reacted with pPB-PGK-Destination (Addgene # 60436) using LR Clonase II (Thermo Scientific,11791020) to generate the desired expression vector. The sequencing validated PB-ER-SIINFEKL construct was co-electroporated along with the Super PiggyBac transposase (System Biosciences, PB210PA-1) into parental E0771 mouse breast cancer cells (CH3 BioSystems) using the Neon Transfection system (Life Technologies, Invitrogen™ MPK5000). Cells were selected with 500 μg/mL Hygromycin before mammary fat pad injection and OT-I expression was validated via Flow cytometry on Cytek Northern Lights.

### Blood collection for peripheral immune cell quantification

Whole blood was collected from an anesthetized mouse (100 μL) into an anticoagulant coated syringe. RBCs were lysed using ACK Lysis Buffer (Lonza 10-548E) and the blood was further diluted at 1:100 ratio in the cell staining buffer (Biolegend, 420201). The cells were stained with CD45 flow antibody and proceeded for flow cytometry to analyze the frequency of CD45$^+$ immune cells.

### Flow cytometry and fluorescence assisted cell sorting

Following single-cell isolation, cells were washed in the cell staining buffer (Biolegend, 420201 or Tonbo, TNB-4222-L500). For experiments on cell surface antibody staining, $F_C$ and $F_{ab}$ receptors were blocked by incubation with 100 μL cell staining buffer and Miltenyi Biotec, 130-092-575 for 30 min at 4 °C and then stained with antibodies of interest for 15 min at 4 °C. For intracytoplasmic staining, the cells were blocked for non-specific binding using the cell staining buffer and then were fixed and permeabilized using True Phos Perm Buffer (BioLegend, 425401 or Tonbo, TNB-1213) before antibody staining. The antibodies used were validated by the manufacturer (BioLegend Inc. comply with the requirements of ISO 13485:2016 or by BD Biosciences, Cell Signaling, or Abcam). The antibodies used for analysis on FC500 were: Zombie Aqua viability dye (BioLegend, 423101, 1:200), PE/APC anti-mouse CD45 antibody (BioLegend, 103105/103112, 30-F11, 1:100), Alexa Fluor 488 anti-mouse CD11c antibody (BioLegend, 149021, N418, 1:50), PE/Cy7 anti-mouse CD11c (BioLegend, 117318, N418, 1:50), APC/Cy7 CD11b antibody (BioLegend, 101225, M1/70, 1:100), APC anti-mouse CD103 (BioLegend, 121413, 2E7, 1:20), APC/Cy7 CD8 anti-mouse antibody (BioLegend, 100713, 53-6.7, 1:100), PE anti-mouse PD-L1 (BioLegend, 124308, 10F.9G2, 1:100), Alexa Fluor 488 pStat1 (Ser 727) (Biolegend, 686410, A15158B, 1:50), PE anti-CD44 anti-mouse antibody (BioLegend, 10323, IM7, 1:100). For analysis on Cytek Flow cytometer, Myeloid cell specific markers used were: BV421 anti-mouse CD86 (BioLegend,105031, GL-1, 1:100), BV570 anti-mouse CD11c (BioLegend,117331, N418, 1:50) or 11b (BioLegend,101233, M1/70, 1:100), BV605 anti-mouse CD11b (BioLegend,101257, M1/70, 1:100) or 11c (BioLegend,117333, N418, 1:50), BV711 anti-mouse PD-L1(BioLegend,124319, 10 F.9G2, 1:100), FITC anti-mouse Ccr1 (BioLegend,152505, S15040E, 1:50), PE anti-mouse CXCL16 (BD, 566740, 12-81, 1:20), PE-Dazzle/594 I-A-I-E (BioLegend,107647, M5/114.15.2, 1:100), PerCP-Cy5.5 anit-mouse pStat1(p727) (BioLegend,686415, A15158B, 1:100), PE-Cy7 CD45 (BioLegend,157206, S18009F, 1:100); Lineage specific markers used were Pacific Blue anti-mouse Lin (BioLegend,133305, 17A2; RB6-8C5; RA3-6B2; Ter-119; M1/70, 1:100) and BV510 anti-mouse c-KIT (BioLegend,105839, 2B8, 1:100); T-cell specific markers used were BV 421 PD-1 (BioLegend,135217, 29F.1A12, 1:100), BV510 CD69 (BioLegend,104531, H1.2F3, 1:100), FITC anti-mouse CD45 (BioLegend,103122, 30-F11, 1:100), PE/Cy5 CD4 (BioLegend,100410, GK1.5, 1:100), PE-Dazzle 594 CD8a (BioLegend,100762, 53-6.7, 1:100), PerCP-Cy5.5 CXCR6 (BioLegend,151120, SA051D1, 1:100), PE-Cy7 anti-mouse CD3e (BioLegend,100319, 145-2C11, 1:100), APC/Fire 750 anti-mouse CD3 (BioLegend, 100248, 17A2, 1:100), immune cell marker BV785 CD45.1 (BioLegend,110743, A20, 1:100), and T-cell activation marker BV711 IFN-γ (BioLegend, 505835, XMG1.2, 1:50), BV570 CD44 (BioLegend, 103037, IM7, 1:100), BV785 anti-mouse/human KLRG1 (BioLegend, 138429, 2F1/KLRG1, 1:100), APC CD49 (BioLegend, 142606, HMa1, 1:100), Pacific Blue CD45 (BioLegend, 103126, 30-F11, 1:100), and PE anti-mouse CD103 (BioLegend, 121406, 2E7, 1:20). For tetramer panel APC/Fire 750 anti-mouse CD3 (BioLegend, 100248, 17A2, 1:100) and BV421 Flex-T™ H-2 K(b) OVA (SIINFEKL) Tetramer (BioLegend, 280051, 1:100) were used. Following staining, cells were washed and resuspended in a 250–300 μL cell staining buffer. The cells were sorted on either a BD Biosciences 1108 FACS Aria III sorter or analyzed on BD FC500 flow cytometer or Cytek Northern Lights (Cytek NL 3000 and Cytek Northern Lights 3000 V/B/R - 38 channel flow cytometer (N7-00008)).

### Cytometry Time of Flight analysis

One million single cells (tumor and tumor-associated immune cells) were resuspended in Maxpar PBS (Fluidigm, 201058). Dead cells were incubated with 0.75 μM Cisplatin for 5 min (Fluidigm, 201064), washed with Maxpar cell staining buffer (Fluidigm, 201068). FC receptors were blocked with TruStain FcX in 100 μL MaxPar cell staining buffer for 30 min at room temperature. Cells were then washed and stained with a cocktail of metal-conjugated antibodies for 30 min at room temperature and washed in a MaxPar cell staining buffer. Optimal concentrations were determined for each antibody by titration and the primary antibodies used are listed in Supplementary Table 1. Cells were resuspended and fixed in 1.6% paraformaldehyde prepared in MaxPar cell staining buffer for 20 min and washed in MaxPar cell staining buffer. Nuclei were labeled by incubating fixed cells in 1:4000 DNA intercalator (Fluidigm, 201192B) dissolved in MaxPar Fix and Perm Buffer (Fluidigm, 201067) for an hour at room temperature or overnight at 4 °C. Following nuclear labeling, cells were washed once in MaxPar cell staining buffer and twice in MaxPar Water (Fluidigm, 201069). Samples were brought to 500,000 particulates per mL in MilliQ water containing 0.1 × EQ beads (Fluidigm, 201078) and run in 450 μL injections on a CyTOF2 instrument.

CyTOF data was either analyzed using Cytobank online software or FlowJo. Cells were identified as events with high DNA (EQbeads served as negative control) and viable cells as events with high DNA and low Cisplatin. Once the viable cells were gated, CD45$^+$ cells were identified as immune cells. These immune cells were downsampled to the number of immune cells present in MTD tumors for viSNE analysis, as MTD hosted the least number of immune cells. CD45$^+$CD11b$^+$ cells were identified as myeloid cells and CD45$^+$CD3e$^+$ cells as Lymphoid T cells. Myeloid cells further parsed into Neutrophils, Monocytes, and Dendritic (DC) cells. The Cell-IDs of these cell subsets are: Neutrophils: Ly6C$^{low}$Ly6G$^{high}$, Monocytes: Ly6C$^{high}$Ly6G$^{high}$, NK cells: Nk1.1$^+$, DCs: CD11c$^{high}$ I-A-I-E$^{high}$. T cells further parsed into CD4$^+$ helper T cells and CD8a$^+$ cytotoxic T cells. Activated T cells were identified as CD3e$^+$CD44$^+$ and regulatory cells as CD3e$^+$CD25$^+$. This gating strategy is illustrated in Fig. S1f.

### Cellular Indexing of Transcriptomes and Epitopes analysis

CITE-seq integrates cell surface protein and its transcriptome measurements as single-cell readouts using oligonucleotide-based

antibodies[59]. Antibodies used to phenotype different immune cells and their specific functional status are listed in Supplementary Table 2. Three different biological tumors were used for both the treatment conditions. A million immune cells were isolated using CD45 micro magnetic beads (Miltenyi Biotec, 130-052-301) from each tumor. Multiplexed with CITE-seq antibodies (CITE or ADT antibodies) and Cell hashing antibodies (HTOs, to identify samples), 20,000 cells were encapsulated using 10x Genomics Chromium single-cell 3′ library and gel bead kit V3 (PN-1000092). On this chip, single cells were captured into emulsions with beads (GEM) to prepare cDNA libraries. Along with the primers for cDNA amplification PCR, primers for ADTs (CITE-seq antibodies) and the HTOs (HTOs 1–6, hashtag antibodies) amplification were included to increase the yield of libraries. Sequencing was performed at the IU Center for Medical Genomics for sequencing on Illumina NovaSeq 6000, S2, platform. The number of reads were 754 million and 380 million for gene expression and cell surface antibody expression respectively. The raw output from the sequencer was demultiplexed into sample specific mRNA, ADT, and HTO FASTQ files and were processed using Cell Ranger v3.1 and 6.

Seurat v3.0[60] R package was used for downstream multimodal analysis. Each sample was demultiplexed based on HTO hashtagging. High quality singlets were subsetted based on mitochondrial gene content less than 20% and gene expression RNA features between 200 and 8000[15,60]. The gene expression matrix was normalized and highly variable features were selected for downstream analysis. Cells were downsampled (2356 per treatment condition) to have equal number of cells in either Vehicle or MCT-treated group for downstream analyses. Clusters formed from such quality controlled 2356 cells per treatment condition were referred to as Seurat Clusters. RNA assay and gene expression matrix were used for Principal Component Analysis (PCA) and Unified Manifold Approximation and Projection (UMAP). Clustered cells were further analyzed for differential gene expression, gene set enrichment (ssGSEA), trajectory analysis, and SCENIC.

## CITE-seq gating strategy
After identifying singlets, *Ptprc* high and CD45 high cells were identified as immune cells. After downsampling (542 single cells per treatment condition), they were further parsed into CD11b$^+$ myeloid cells and CD3e$^+$ T cells. Then Ly6C$^{high}$ Monocytes were gated out. Further, CD24$^{high}$ F4/80$^{high}$ cells were gated to identify CD11b$^+$ and CD11c$^+$ TAMs. On the other hand, CD24$^{high}$ F4/80$^{low}$ cells were gated to identify CD11b$^+$ DCs and CD103$^+$ DCs (together TADs). They were referred to as TADs and TAMs. The lymphoid cells with CD3e high expression further parsed into CD4 high and CD8a high cells. This gating strategy is illustrated in Fig. S2B and has been adopted from previous publications[13,14].

## Dynverse trajectory inference analysis
Dynverse package[22] was used for single-cell trajectory inference. All the myeloid clusters which shared transcriptional similarities with TADCs and TAMs (Seurat Clusters 0, 1, 3, 4, 7, 8, 10, 12, and 14) were selected as Dyno subclusters for trajectory inference. Cluster markers identified were: *S100a9*, *mt-Co3*, *Arg1*, *H2-Ab1*, *Cd74*, *Mrc1*, *Ccl8*, *Ly6c2*. These markers were selected as the start and end points of trajectories. Then, PAGA tree analysis function was used to infer the trajectory of relevant Seurat Clusters.

## Single-Cell Regulatory Network Inference and Clustering analysis
SCENIC version 1.1.2.1[28,61] was used for building the regulons and targetomes/gene regulatory networks (GRN). The analysis was performed using R software. For efficient analysis, the entire data set was downsampled to 1000 cells. For building the GRN, transcriptional factors and their top 10 potential gene targets were predicted via regression-based network inference. Putative regulatory regions of these positively regulated targets were searched and the enriched motifs were identified. The motifs with a NES > 3 were considered significant and positive regulators were chosen with pearson 0.03 (these were the values set by default).

In some instances, for separating regulons with greater confidence in predicting motifs, NES ≥ 4 or 5 was chosen. We analyzed the data in two different iterations (1) All the myeloid cells pooled together: for Fig. 3c, d, split by Seurat Clusters (used to generate heatmap in Fig. 3c) and (2) Individual Myeloid Seurat Cluster specific regulon analysis (Fig. 3e and Supplementary Data 5). Each dot in the scatter plot represents a specific NES. We realized that multiple motifs of a regulon could have a NES and hence not to clutter the plot, we chose to represent each regulon with its highest NES (represented along the Y axis) with a dot. The scatter plots were generated after modifying the entire gene sets listed in their respective Supplementary Data 5. For Cluster 0 *Stat1* targetome analysis: (1) The entire gene list is presented in Supplementary Data 7. (2) The scatter plot in Fig. 5g, represents *Stat1* gene targets with NES > 3 and DEG in MCT with $p < 0.05$.

## T-cell proliferation and suppression assay
After splenocytes were isolated as mentioned above (Single-cell isolation method section), Pan T Cell Isolation Kit II, mouse (Miltenyi Biotec, 130-095-130 or BioLegend, 480024) was used to isolate T cells. Using Fluorophore Assisted Cell Sorting (FACS), CD11c$^+$ cells were isolated from CD45$^+$ cells for proliferation. Using Fluorophore Assisted Cell Sorting (FACS), CD11b$^+$ Ccr1$^{high}$ and CD11b$^+$ Ccr1$^{low}$ cells were sorted from CD45$^+$ cells for suppression assay. For both the assays, CD45$^+$ cells were isolated using magnetic beads (Miltenyi Biotec, 130-052-301). APC or APC/Cy7 anti-mouse CD11b (BioLegend, 101212 or 101225, M1/70) PE anti-mouse Ccr1 (BioLegend, 152507, S15040E) markers were used for FACS sorting. T cells were stained with CFSE stain (ThermoFisher, Catalog#C34554, 1:1000 of stock solution) following manufacturer's instructions. For suppression assay, T cells with T-cell activation beads and either CD11b$^+$ Ccr1$^{high}$ or CD11b$^+$ Ccr1$^{low}$ cells were cultured in 1:1 ratio in the heat-inactivated RPMI medium supplemented with IL2 for 72 hrs. As experimental controls, T cells were either cultured alone or with T-cell activation beads (Dynabeads™ Mouse T-Activator CD3/CD28 for T-Cell Expansion and Activation, Thermo Fisher Scientific, 11456D). After 72 h of culture, the cells were stained with CD3e (BioLegend, 100319, 145-2C11), CD8a (BioLegend, 100762, 53-6.7), CD4 (Biolegend, 100410, GK1.5), PD-1 (BioLegend, 135217, 29F.1A12), and CXCR6 (BioLegend, 151120, SA051D1) anti-mouse antibodies and proceeded for Spectral Flow Cytometry (Cytek NL 3000). For OT-I co-culture activation assays, $1 \times 10^5$ T cells were plated per well in a 96-U well plate. Positive control wells were pre-treated with anti-CD3 (5 μg/ml in PBS, Thermo Scientific, 16-0032-82, 17A2) while negative control wells were treated with PBS for 2 hours at 37 °C. Anti-CD28 antibody (3 μg/ml in PBS, Thermo Scientific, 16-0281-82, 37.51) was added to positive control wells. For co-culture wells, $1 \times 10^5$ of isolated CD11c cells were added in addition to the T cells. All wells received 20 ng/ml recombinant mouse IL-2 (BioLegend, 575404) and were incubated for 72 h.

## Adoptive transfer of CD45.1$^+$ cells
Tibiae, Fibulae, and Femurs were isolated from CD45.1$^+$ B6 mice into heat-inactivated RPMI medium. The bones are then crushed with a pestle and washed and passed through a 40 μm cell strainer (VWR, 76327-098). RBC lysed and dead cells were separated (using dead cell removal kit, Miltenyi Biotec, 130-090-101). The live cells were then resuspended in a sort buffer to inject 1 million cells per B6 MMTV-PyMT mouse retro-orbitally. For adoptive transfer of CD45.1$^+$ CD3$^+$ T cells, the CD3 isolation beads (BioLegend Cat # 480024) were used to enrich T cells from CD45.1$^+$ spleens and injected retro-orbitally.

**Intratumoral injection of IgG or CXCL16 neutralization antibody**
In all, 10 ng of IgG or CXCL16 neutralizing antibody was dissolved in diluent to make upto 10 μL of the antibody cocktail and delivered at a constant rate (5 μL per min) into the tumor using a Harvard Syringe Pump at a 45° angle. The tumors more than 10 × 10 mm in size were given 20 μL of the antibody. The tumors less than 5 × 6 or greater than 12 × 13 mm³ in size were not included in the study. For a combinatorial PD-1 inhibitor with CXCL16 NAb (Figs. 4 and 6), 30 ng in 10 μL of CXCL16 was given per dose per tumor.

**shRNA lentiviral transfection assay**
Bacterial stocks for pLKO or shRNA mouse *Stat1*(Bacterial Glycerol Stock Sequence (same as Construct 1) 1 # CCGGGCTGCCTAT-GATGTCTCGTTTCTCGAGAAACGAGACATCATAGGCAGCTTTTTG, Bacterial Glycerol Stock Sequence (same as Construct 2) 2# CCGGGCTGTTACTTTCCCAGATATTCTCGAGAATATCTGGGAAAGTA ACAGCTTTTTG) constructs were purchased from Sigma Aldrich. The plasmids were packaged into lentiviral particles using the 2nd generation packaging plasmid (Addgene). 10 ml of virus containing media were concentrated with Lenti-X concentrator (Takara Bio, 631231) to a titer sufficient to decrease pStat1 levels by 25% (Mean pStat1 + CD11b + cells: pLKO: 100, Construct 1# 75, Construct 2# 75). MCT-treated CD11b⁺ cells were isolated from tumors using MojoSort Mouse CD11b Selection Kit (BioLegend, 480109). For transfecting every million primary CD11b⁺ cells, 1 ml of the viral concentrate was used along with 5 μg/mL of polybrene. The cells were transfected for 72 hrs prior to analysis by flow cytometry. For flow cytometry analysis, the cells (post transfection) were washed with a cell staining buffer, blocked for 20 min on ice in dark and stained with APC/Cy7 CD11b antibody (Bio-Legend, 101225, M1/70) and PE anti-mouse PD-L1 (BioLegend, 124308, 10F.9G2) flow antibodies and incubated for 20 min in dark on ice. The cells were further fixed and permeated using a phospho permeable buffer. Later, cells were washed, stained with Alexa Fluor 488 pStat1 (Ser 727) (Biolegend, 686410, A15158B) for 20 min, washed, and resuspended in a cell staining buffer before analyzing on the flow cytometer.

**Multiple gene association data analysis**
For multiple gene association analysis, GEPIA web server was used. GEPIA uses RNA-seq data from TCGA and GTex projects. Here we compared *Stat1* and *Cd274* correlation between healthy breast, blood, ovarian, skin, and lung tissues and cancerous tissues (BRCA, CHOL, LAML, HNSC, OV, SKCM, SARC). We observed that *Stat1* and *Cd274* correlate stronger in cancerous conditions ($R = 0.71$) than in healthy states ($R = 0.18$; Fig. S5h).

**Tissue processing**
For formalin-fixed, paraffin-embedded tissues (FFPE), the tumor tissues covered in cassettes were incubated in 10% nebulized buffered formalin overnight at room temperature. The tissue cassettes were then incubated in 70% ethanol for a week before being processed in the Leica Tissue Processor using an 8-hour program specific for fat rich tissues. To perform staining, 4 μm thick tissue ribbons were sectioned using Leica Tissue Microtome. These tissue ribbons were then adhered to SuperFrost slides (VWR, 48311-703) and dried overnight before being processed for staining. FFPE sections of human TNBC were obtained from UT Southwestern Harold C. Simmons Comprehensive Cancer Center Tissue Management Shared Resource (TMSR). Patient information is included as Supplementary Data 6.

**Tissue Immunohistochemistry**
FFPE tissue sections were deparaffinized by incubation in a 60 °C oven for 30 min and in Xylene for 10 min. Hydration was achieved by sequentially immersing slides for 10 min each, in staining dishes containing 100%, 95%, 80% ethanol, and ddH2O. Antigens were unmasked

by boiling slides in 1× Sodium citrate solution (10 mM Sodium citrate, pH 6.0) in a pressure cooker for approximately 20 min (first 10 min without weight at 350 °C and next 10 min with weight at 240 °C). After antigen retrieval, tissues were blocked in 3% H2O2 in phosphate buffered saline (PBS) for one hour at room temperature to quench endogenous peroxidase. To quench non-specific antibody binding, the tissue sections were blocked in horse serum for an hour. Tissues were then stained with primary antibodies overnight in a humidity chamber at 4 °C. Following overnight incubation, the tissues were washed in TBST three times 30 min each before an hour of incubation with secondary antibodies. The Vectastain Elite ABC-HRP Kit RTU (Vector Laboratories, PK-7200) was used to develop the peroxidase before incubating with the ImmPACT-DAB chromogen. The primary antibodies used were Ki-67 (dilution 1:800, clone D3B5, Cell Signaling Technology, CST 12202S) and anti-CD3 (dilution 1:500, Abcam, ab 16669, Rabbit mAb, SP7).

**Tissue Immunofluorescence**
FFPE tissue sections were first incubated in a 60 °C oven for 30 min and washed twice with Xylene for 5 min. Hydration was achieved by sequentially immersing slides, for 10 min each in 100%, 95%, and 80% ethanol. Then the slides were washed in ddH2O before proceeding to the antigen retrieval step. Antigens were unmasked by boiling slides in 1× Sodium citrate solution (10 mM Sodium citrate, pH 6.0) in a pressure cooker for approximately 20 min (first 10 min without weight at 350 °C and next 10 min with weight at 240 °C). After cooling to room temperature, slides were washed three times with PBS in vertical staining jars. Dewaxed specimens were blocked by incubation with NaOH and 4.5% H2O2 for an hour. Then the slides were incubated with Odyssey blocking buffer (LI-COR Biosciences, 927-40150) or 2.5% Donkey Serum in PBS for 60 mins by applying the buffer to slides as a droplet at room temperature; evaporation was minimized by using a humidity chamber. Slides were incubated overnight with primary antibodies (both fluorophore conjugated and unconjugated). The unconjugated primaries were further incubated with secondaries for 2 h before washing three times with PBS, 5 min each. Finally, slides were incubated with Hoechst 33342 (1 mg/ml) in Odyssey blocking buffer for 30 min and washed three times in PBS, 5 min each. A set of immune cell markers and proteins of interest were used to stain breast tissues (and Spleen for positive control when appropriate) which are treated either with PBS, MCT, MTD, anti-PD-1, CXCL16 NAb, Stat1 inhibitor or a combination of either of these. They are: anti-Iba1 (dilution 1:2000, Abcam ab5076, Goat polyclonal immunogen to NP_001614 and NP_116573), Phosphorylated STAT1 Stat1 - Ser 727(dilution 1:200, BioLegend 686405, A15158B), anti-CD3 (dilution 1:500, Abcam,ab 16669, Rabbit mAb, SP7), anti-CD8 (dilution 1:250, CST 98941S, Rabbit mAb, D4W2Z), anti-PD-L1 (dilution 1:500, CST 64988S, Rabbit mAb, D5V3B), and anti-CXCL16 (dilution 1:500, R&D Systems MAB503-100, 142417).

**Statistical analysis**
All common quantitative data were analyzed and plotted as box plots or survival curves using the R program. $p < 0.05$ is significant in all our analyses. Seurat, ssGSEA, Dynverse, and SCENIC analyses were also performed in the R program as described in each dedicated section. Flow cytometry and CyTOF data was analyzed using FlowJo software. viSNE analyses of CyTOF were performed using Cytobank. For quantitative image analysis, positively stained cells (myeloid, lymphoid, and their spatial proximity with markers such as PD-L1 and CXCL16) or Ki67 high to low cells were counted manually in a given field of view using ImageJ.

**Reporting summary**
Further information on research design is available in the Nature Portfolio Reporting Summary linked to this article.

## Data availability

All CITE-sequencing Data has been deposited at GEO (GSE158888). Source data are provided with this paper.

## Code availability

The R code for data analysis is available at: https://github.com/S-Zhang-Lab/MCT-myeloid upon publication. Source data are provided in this paper.

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

## Acknowledgements

This work was partially supported by Dee Family endowment and CPRIT RR220024 award (S.Z.). We are grateful for the technical support of the following core facilities: Notre Dame Genomics and Bioinformatics Core Facility, Notre Dame Freimann Life Sciences Center, Indiana University School of Medicine Center for Medical Genomics Core Facility, and the University of Texas Southwestern Medical Center Tissue Core. We are also grateful for the technical guidance and discussion with Dr. Tuoqi Wu and Dr. Chen Yao at UT Southwestern Medical Center.

## Author contributions

B.P., S.R.F., I.H.G., E.A., X.L., A.K.M., Q.W., R.A.N., S.M.G., E.N.H., and S.Z. performed the experimental studies. B.P. and S.Z. carried out the data analysis. Y.P. and C.L. provided human tissue sample curation. B.P. and S.Z. conceived the study concept. S.Z. supervised the work.

## Competing interests

The authors declare no competing interests.
