## [Peer Review File · Nature Communications]

Targeting CXCL16 and STAT1 augments immune checkpoint blockade therapy in triple-negative breast cancerEditorial Note: Parts of this Peer Review File have been redacted as indicated on p9 and p28 to remove third-party material where no permission to publish could be obtained.

REVIEWER COMMENTS

Reviewer #1 (Remarks to the Author):

Palakurthi et al. presented a lengthy study on the the role of CXCL16 and STAT1 in maximising the anti-tumour response of immune checkpoint blockade in myeloid cells. The study is length with many techniques but it suffers from many gaps that needs to be addressed before it can be considered for publication in any high impact journal.

The main shortcoming of the article is that it is not focused at the clinical level to any specific cancer, however in both the abstract and the introduction it seems to refer to breast cancer, however breast cancer is a complex disease with multiple different subtypes. Having said that, if the focus of this study is on breast cancer, the authors only considered the triple negative using the C3-1-TAg mouse model and HER2 positive using the MMTV-neu model as mentioned in their methods but there are still the luminal and triple positive subtypes.

Therefore the study requires more focus on the role of CXCL16 and STAT1 in a specific cancer rather than pan cancers. But if they want to do pan cancers then they should make the study more structured and show the mechanism of CXCL16 and STAT1 pathways in those cancers.

Extensive work was carried out in the study as highlighted in the methods section, however the manuscript will benefit greatly from testing the biomarkers on human model and I suggest the authors test the biomarkers identified from the bioinformatics in silico analysis on few of the breast cancer cell line models such ask BT474, MCF7, MDAMB231, SKBR3 as well as others. The article: <https://www.ncbi.nlm.nih.gov/pmc/articles/PMC5665029/pdf/jcav08p3131.pdf> has good review of all the BC cell lines in table 1. I suggest the authors choose different cell lines that represent the different BC subtypes and carry out some of the validation on those cell lines in a similar manner that was done in the study. Also if possible then to validate those in vivo on some patients' biopsies from those who were treated with chemotherapy but this is not essential.

In summary the manuscript has some interesting data but it needs to be more focussed on a disease or if it is across cancers it has to be made clear whether the CXCL16/STAT1 axis are involved in all cancers and if the molecular mechanism differ across the different cancers.

Other minor points:

- 1) in the abstract the phrase "overall balance" is rather vague and better to use homeostasis as the immune system is always active and dynamic
- 2) In the introduction the first sentence "Cancer is a systemic disease" could be better rephrased as cancer is a heterogeneous disease
- 3) There are many methods used in the study and it is difficult to follow the flow of how the methods are linked so a suggestion is to include work flow diagram to summarise the methodology. This can be included in the results with the different numbers fo genes isolated from the study
- 4) Include the clone number for the antibodies used in flow, IHC and IF
- 5) In the multiple gene association analysis, it is not clear why some cancerrs are included but others not. In addition what is the disease counterpart for the blood from healthy individuals will that be leukaemias? Also this type of analysis needs p value as well as R value otherwise it will be too vague
- 6) In tissue process generally it is better to section the FFPE specimen at 3um to ensure minimal overlap between disease and normal cells. Therefore the authors are advised to explain that the 4um capture the tumour areas well
- 7) For IHC plesae include the dilutions used especially for Ki67 and CD3

8) In the statistical analysis section the authors need to mention that $p < 0.05$ is considered to be significant unless otherwise

Reviewer #2 (Remarks to the Author):

The manuscript by Bhavana and colleagues focuses on maximizing the efficacy of anti-PD-1 immunotherapy in chemotherapy-primed breast cancer models through fine-tuning of the immunogenic and tolerogenic balance in the tumor microenvironment shaped by tumor-infiltrating myeloid cells. The authors postulate that tumor-associated myeloid cells, which are highly plastic, prevent ICB from reaching its full therapeutic potential. By leveraging single-cell transcriptomic trajectory analyses (MCT) after chemotherapy priming, they show that metronomic chemotherapy leads to accelerated T cell exhaustion through CXCL16-mediated recruitment of myeloid cells and expansion of STAT1-driven PD-L1 expressing myeloid cells. Indeed, combination treatment with anti-CXCL16 and a STAT1 inhibitor prevents T cell exhaustion, thereby enhancing the efficacy of anti-PD-1 ICB. This is a novel and potentially important finding with clear clinical implications. However, the studies are limited to mouse models of breast cancer and no attempt is made to examine human data. Furthermore, the exact roles of CXCL16 in cluster 4 and STAT1-PD-L1 in cluster 0 are not mechanistically well explained. Finally, there are major concerns with experimental design and interpretation of data that should be addressed.

Comments:

1) The single cell seq data in the cell clusters are confusing. For example, *irf8* is present in many cell clusters, which is difficult to understand since *Irf8* is a cDC1 specific transcription factor.

2) T cell proliferation based on CFSE is not clearly displayed and not explained mechanistically. In Figure 1H, the gating strategy of CD8+ T cells and CFSE dilution are difficult to decipher. Without specific antigen loading of the DCs, why should syngeneic tumor-infiltrating DCs activate naïve CD8+ T cells in vitro? It is also difficult to understand how this in vitro culturing system can model the in vivo behavior of these CD11c+MHCII+ cells in the tumor setting? Moreover, the functional activation status of the activated CD8+ T cells should be indicated on the basis of IFN γ , Granzyme B staining.

3) In Figure S2A, CITE-seq, the gating and definition of cDC1 and cDC2 are inadequate. cDCs should be defined with lineage-specific transcription factors. The discordance between the cell-surface and transcriptional status cannot be due to a high level of transcriptional plasticity of myeloid cells (Figure 2A and S2A).

4) Path 4 analysis highlights the role of CXCL16 expressing cells in determining tumor-infiltrating T cell fate. However, as shown in Figure 4A, CXCL16 expression is modest, which casts doubt on the utility of CXCL16 as a target for ab blockade. In Figure S4C, neither the CXCL16 expressing cell type nor the sorting strategy of these cells is clear. Are these myeloid cells? In Figure S4C, the CFSE staining in the culture with CXCL16 cells exhibits almost no CFSE dilution (ie, proliferation) compared with bead stimulation. How did the authors manage to gate a single peak into several divisions? How can the authors exclude the possibility that CXCL16 is secreted by other cells?

In Figures 4c and 4d, which display the results of adoptive transfer of CD45.1 BMD into anti-CXCL16 ab treated mice, the readout is CXCL16+ cells in CD45.1+ cells, but is the intent to focus on CXCR6+ cells? In Figure S4e, the gating strategy needs to be clarified or corrected. Also, what is the gating strategy of CD45.1+ T cells to explain the bars in Figure 4F? The time-dependent tumor growth curve needs to be shown to explain the statistical analysis in Figure 4I.

5) By using a single cell surface marker such as CCR1 to identify individual clusters without validation is not acceptable. In Figure S3, the explanation for selecting CCR1 to identify cluster 0 is not clear, because the CCR1 expression level on this cluster is not high, and the percentage of CCR1+CD11b+ cells in myeloid cells is only 10% (Figure 5A).

6) Figure 5G is confusing because, in pLKO, the % of pSTAT1+PD-L1high among CD11b+ cells is almost 100%. What is this experiment intended to show?

7) Figure S5A, the CD3+ T cell gating strategy is not valid, because there is no clear separation of T cells in TILs. The CD3+ T cells should be further gated with CD4 and CD8 together but not separately.

8) The time-dependent tumor growth curve should be shown in Figures 6B and 6F. Without this information, it is difficult to judge the data quality in these figures

Reviewer #3 (Remarks to the Author):

The authors describe the modulation of the tumor immune microenvironment in breast cancer after metronomic chemotherapy (MCT). They show that MCT treatment improves survival and promotes T cell recruitment and T-myeloid cell proximity in different breast cancer models. However, MCT is also associated with T cell exhaustion and the recruitment of immunosuppressive myeloid cells that are dependent on STAT1. They propose that the inhibition of STAT1 could be an effective combinatory treatment to enhance the benefit of both standard therapy and immune checkpoint blockade (ICB) in breast cancer.

The paper is interesting and could be relevant in the immunotherapy field, but some of the data are technically not convincing and do not fully support the authors' conclusions.

Specific comments:

1) The authors suggest that myeloid/T cell proximity is promoted after MCT treatment. In Figure 1E, the IF picture does not show that. Figure 1C shows a general decrease of Iba1+ cells after MCT treatment and this is not consistent with Figure 3F where Iba1+ cells are increased in the MCT treated sample. I would suggest changing the term co-localization, since it generally refers to proteins expressed by the same cell type.

2) The gating strategy shown in S1F, S1G, S1I should be improved. What are the authors showing in Figure S1F and S1G? It should be quantified. The strategy used to gate NK cells in Figure S1I is not correct: NK cells express CD11b at a lower level compared to myeloid cells, but not 100% of NK cells are CD11b+; NK cells are NK1.1+CD3-. Finally, the analysis of DC populations in tumor draining lymph nodes would be interesting.

3) The authors use the term "TIL" when referring to CD11b+ cells and DCs. TILs should be only used for lymphocyte populations.

4) The CITE-seq data are interesting but clusters are poorly defined: appropriate gene signature should be used. DCs and macrophages could be discriminated using Zbtb46 expression. Are the cells in Cluster 14 pDCs? In Figure 2F the authors show the enriched annotated pathways for the myeloid subclusters and they are all related to T cell functions. This should be clarified.

5) The authors suggest that CXCL16+ cells stimulate T cell function (page 12), but the Figure (Figure S4C) shows the opposite: the CFSE dilution is increased without CXCL16+ cells. Also, the gating strategy and plots showing the sorting of CXCL16+ cells would be interesting.

6) In Figure 4, the authors show the effect of anti-CXCL16 antibody treatment in the breast cancer tumor model. How do the authors explain the reduced recruitment of CXCL16+ cells between the control and the MCT-treated group (Figure 4D)? The MCT treatment is shown to be associated to an increased frequency of PD1high CD4 T cells and this is reduced by the Ab treatment (Figure 4E). Is this the case for the second experimental setting (Figure 4G)? The authors suggest that the Ab treatment attracts exhausted PD1+ T cells. The fact that they are recruited is not proven. Where would the reservoir of exhausted T cells be? It would be more likely that the treatment modulates PD1 induction within the tumor. Tumor growth is reduced by the CXCL16 treatment (Figure 4H). How is tumor growth upon CXCL16 treatment in the experiment shown in Figure 4C?

7) In Figure 5H, the authors show an IF quantification of PD-L1 expression upon STAT1 inhibition. First, in the IF picture there is a difference in Iba1 total cells after STAT1 inhibition. Then, how is tumor growth in these groups? The flow analysis of PD-L1 expression in the conditions shown in Figure 5H would corroborate the finding.

8) The ex vivo culture of tumor-associated CD11b+ cells and gene knock down (Figure 5G) is a challenging experiment. Can the authors show cell viability and the CD11b+ cell subpopulations?

Point-by-point Responses to the Reviewers' Comments

RE: NCOMMS-22-02938-T

We would like to thank the editor and the reviewers for valuable and constructive comments. In this revision, we have carefully considered and thoroughly addressed all of the reviewers' comments and revised our manuscript accordingly with our best efforts. We have provided additional evidence, modified our data analyses, and performed new experiments to address concerns of reviewers comprehensively. In this revised manuscript, we have generated 44 new panels of figures, including additional *in vivo* and *ex vivo* experiments, and *in silico* analysis.

First, we agree with the reviewer and the editor, that validating the human relevance of our data is important for a translational-oriented study like ours. We want to point out that the goal of our preclinical study is to prospectively model the tumor's response to metronomic treatment and immunotherapy treatment. Such prospective human clinical trials are not yet available in the clinic. Currently, there are no paired (pre-/post treatment) tumor samples with single cell profiling data derived from metronomic chemotherapy or immunotherapy from ongoing clinical trials. Despite the above limitations, we have obtained raw data of the recently published single cell profiling human study ¹. This recent patient dataset was derived from human breast cancer patients treated with neoadjuvant chemotherapy followed by immune checkpoint blockade. As detailed in our response to Reviewer 1's comments below, we were able to validate some of the findings from our mouse models in the breast cancer patient datasets. Moreover, following the reviewer's suggestion, we have modified the manuscript text including the title in a way such that it reflects our work's primary focus on triple-negative breast cancer.

Second, to address the technical/methodical concerns, we have added new experimental data using tumor antigen-specific, T cell and Ova tumor-antigen co-culture systems. We have also taken supportive evidence from existing literature to support how this *in vitro* co-culture system in some degree mimics *in vivo* DC and T cell interaction. We have also clarified the rationale behind using Ccr1 as an alternative marker to sort Cluster 0 cells for the co-culture experiments. In addition, we have comprehensively improved the gating strategy

Third, in response to Reviewer 3's comment, we have conducted an additional *in vivo* adoptive transfer experiment and analyzed tumor draining lymph nodes to gain further insight on the mechanisms of T cell exhaustion upon CXCL16 NAb. Furthermore, we provided additional results from STAT1 knockdown experiments and demonstrated that STAT1 plays a more significant role in Cluster 0 myeloid cells as compared to other tumor infiltrating myeloid cells.

We deeply appreciate the reviewers' critical and constructive comments. These critiques helped us strengthen our conclusions through additional experiments and careful revisions. We believe our manuscript has been improved significantly. Please see below for our detailed point-to-point response (in blue text) to each comment.

Reviewer #1 (Remarks to the Author):

Palakurthi et al. presented a lengthy study on the the role of CXCL16 and STAT1 in maximising the anti-tumour response of immune checkpoint blockade in myeloid cells. The study is length with many techniques but it suffers from many gaps that needs to be addressed before it can be considered for publication in any high impact journal. The main shortcoming of the article is that it is not focused at the clinical level to any specific cancer, however in both the abstract and the introduction it seems to refer to breast cancer, however breast cancer is a complex disease with multiple different subtypes. Having said that, if the focus of this study is on breast cancer, the authors only considered the triple negative using the C3-1-TAg mouse model and HER2 positive using the MMTV-neu model as mentioned in their methods but there are still the luminal and triple positive subtypes. Therefore the study requires more focus on the role of CXCL16 and STAT1 in a specific

cancer rather than pan cancers. But if they want to do pan cancers then they should make the study more structured and show the mechanism of CXCL16 and STAT1 pathways in those cancers.

Answer: We thank Reviewer 1 for this suggestion. We have modified the manuscript text to reflect that our work is focussed on triple-negative breast cancer.

Extensive work was carried out in the study as highlighted in the methods section, however the manuscript will benefit greatly from testing the biomarkers on human model and I suggest the authors test the biomarkers identified from the bioinformatics in silico analysis on few of the breast cancer cell line models such as BT474, MCF7, MDAMB231, SKBR3 as well as others. The article:

<https://www.ncbi.nlm.nih.gov/pmc/articles/PMC5665029/pdf/jcav08p3131.pdf> has good review of all the BC cell lines in table 1. I suggest the authors choose different cell lines that represent the different BC subtypes and carry out some of the validation on those cell lines in a similar manner that was done in the study. Also if possible then to validate those in vivo on some patients' biopsies from those who were treated with chemotherapy but this is not essential.

In summary the manuscript has some interesting data but it needs to be more focussed on a disease or if it is across cancers it has to be made clear whether the CXCL16/STAT1 axis are involved in all cancers and if the molecular mechanism differ across the different cancers.

Answer: We thank Reviewer 1 for the valuable comments and suggestions. We agree that the significance of our study will be greatly enhanced by demonstrating the clinical relevance of our findings to patient samples. It will be valuable to explore whether the cell cluster markers identified from our single cell CITE-seq analysis using mouse models can serve as biomarkers to guide clinical treatment. However, using human cell lines might not be an ideal approach in our project's context. This is because our project focuses on the tumor-immune microenvironment interactions. Such interactions could not be faithfully modeled by using human cell lines only or *in vivo* human-in-mouse settings in an immune-deficient mouse model. Thus, we have to rely on the spontaneous mouse tumor with intact immune system to study the question of tumor-immune interactions.

We agree that the mouse model has its own limitations. In order to relate our findings to clinical scenarios, a comparative analysis between mouse data and human sample is necessary. Given our experimental setting (chemo-treated breast cancer patient), we reasoned that most informative analysis should be comparing mouse-derived gene signature to patient sample data sets that 1) contain both breast cancer samples collected before and after chemotherapy, preferably MCT treatment and 2) provide single cell resolution in order to unbiasedly analyze/compare multiple tumor microenvironmental immune cell types. After extensive search of available sequencing datasets, fortunately, we obtained the raw data from Bassez A study published recently (Bassez A. et al, *A single-cell map of intratumoral changes during anti-PD1 treatment of patients with breast cancer. Nature Medicine volume 27, pages 820–832 (2021)*)¹. This study aims to understand why only a subset of tumors respond to immune checkpoint blockade (ICB). This study includes single cell transcriptome analysis of paired tumor biopsy before and after receiving platinum-based neoadjuvant chemotherapy immediate before anti-PD1 treatment (n = 11). We are cautious in interpreting the correlative study as there are caveats in using patients data. One caveat of validating our data using this patient dataset is the neoadjuvant chemotherapy treatment regimen patient is not the same as the doxorubicin plus cyclophosphamide (AC) regimen we used in our study. Furthermore, their treatment is a maximum dosing (MTD) strategy and not metronomic regimen. Despite the above caveats, this single cell dataset represents the closest patient single cell data with chemotherapy treatment currently available.

PTP R1C1. Bassez single cell dataset analysis. (A) UMAP split by treatment groups shows different myeloid cell subsets. (B) Bar plot shows the percentage distribution of different Seurat clusters shown in (A). (C) Heatmaps show enrichment of Mouse Path 1 (left) and Path 4 (right) marker genes in Clusters 2 and 1 respectively.

To strengthen the clinical relevance of our findings by which the emergence of trajectory Path 1 myeloid cells (cluster 0 cells) after chemo treatment contribute to breast cancer immune microenvironment in human breast cancer context, we performed *in silico* analysis using Bassez dataset. The goal of our analysis is two-fold: 1) whether tumor infiltrating myeloid clusters change upon chemo treatment; 2) whether marker genes identified in our mouse model, e.g. trajectory Path 1 and Path 4 markers, express in myeloid cells identified from human breast tumors.

First, we subsetted out myeloid cells (ITGAM high cells) among all the immune cells (PTPRC.high and KRT8.low). UMAP based dimension reduction method clustered each single cell into four Seurat RNA clusters, 0 to 3 (Fig. PTP R1C1.A). We observed that certain myeloid cell clusters (not all) increased upon chemo treatment. Cluster 1 and 2 myeloid cells increased after chemo from 26.7% (Cluster 1) and 5.6% (Cluster 2) in the Pre-chemo treated tumors to 38.6% and 21.86% in the Chemo treated cohort respectively (Fig. PTP R1C1.B). Next, we examined the expression of trajectory Path 1 and Path 4 marker genes in human Cluster 1 and 2 myeloid cells (Fig. PTP R1C1.C). Among these two clusters, Cluster 2 has a higher level of Mouse Path 1 (Cluster 0) marker genes (Fig. PTP R1C1.C, left). On the other hand, Path 4 appears to decrease upon chemo treatment potentially due to maximum dosing (MTD) of the chemo (Fig. PTP R1C1.C, right). It is encouraging to observe the myeloid cell transcriptome shifting in patients cohorted treated with chemotherapy, which supports the value of using mouse models to examine the temporal-dependent changes in tumor microenvironment (TME), particularly in the context of treatment. Beyond the scope of our current preclinical study, future prospective clinical trials using metronomic low dose chemo (MCT) strategy followed by single cell profiling analysis need to be conducted to fully examine the TME changes in full under the MCT and immunotherapy.

Other minor points:

1) in the abstract the phrase "overall balance" is rather vague and better to use homeostasis as the immune system is always active and dynamic

Answer: We have replaced "overall balance" with "homeostasis" throughout the manuscript.

2) In the introduction the first sentence "Cancer is a systemic disease" could be better rephrased as cancer is a heterogeneous disease

Answer: We have replaced "Cancer is a systemic disease" with "Cancer is a heterogeneous disease".

3) There are many methods used in the study and it is difficult to follow the flow of how the methods are linked so a suggestion is to include work flow diagram to summarise the methodology. This can be included in the results with the different numbers to genes isolated from the study

Answer: We have included a workflow diagram to summarize the methodology to help follow the flow of methods used in our project. The diagram is included in the revised Fig. S2A and is shown as Fig. PTP R1C Minor 3.

PTP R1CMinor3 Schematic of research workflow.

4) Include the clone number for the antibodies used in flow, IHC and IF

Answer: We thank Reviewer 1 for this comment. It is indeed important to know the clone numbers of the antibodies used in various methods in the study (Flow, IHC, and IF). We have tabulated the different antibodies used in the study along with their manufacturer's information and catalog number. This information has been included in the Methods section in the revised manuscript.

Fluorophore/Antibody	Manufacturer, Cat#	Clone
PE/APC anti-mouse CD45	BioLegend, 103105/103112	30-F11
AF 488 anti-mouse CD11c	BioLegend, 148021/117311	N418
APC/Cy7 anti-mouse CD11b	BioLegend, 101225	M1/70
APC anti-mouse CD103	BioLegend, 121413	2E7
APC/Cy7 CD8 anti-mouse	BioLegend, 100713	53-6.7
PE anti-mouse PD-L1	BioLegend, 124308	10F.9G2
AF488 anti-pStat1 (Ser 727)	Biolegend, 686410	A15158B
PE anti-CD44 anti-mouse	BioLegend, 10323	IM7

BV421 anti-mouse CD86	BioLegend,105031	GL-1
BV570 anti-mouse CD11c	BioLegend,117331	N418
BV570 anti-mouse CD11b	BioLegend,101233	M1/70
BV605 anti-mouse CD11b	BioLegend,101257	M1/70
BV605 anti-mouse CD11c	BioLegend,117333	N418
BV711 anti-mouse PD-L1	BioLegend,124319	10F.9G2
FITC anti-mouse Ccr1	BioLegend,152505	S15040E
PE anti-mouse CXCL16	BD, 566740	12-81
PE-Dazzle/594 I-A-I-E	BioLegend,107647	M5/114.15.2
PerCP-Cy5.5 anti-mouse pStat1(p727)	BioLegend,686415	A15158B
PE-Cy7 anti-mouse CD45	BioLegend,157206	S18009F
Pacific Blue anti-mouse Lin	BioLegend,133305	17A2 ; RB6-8C5 ; RA3-6B2 ; Ter-119 ; M1/70 ;
BV510 anti-mouse c-KIT	BioLegend,105839	2B8
BV421 anti-mouse PD-1	BioLegend,135217	29F.1A12
BV510 anti-mouse CD69	BioLegend,104531	H1.2F3
FITC anti-mouse CD45	BioLegend,103122	30-F11
PE/Cy5 anti-mouse CD4	BioLegend,100410	GK1.5
PE-Dazzle/594 anti-mouse CD8a	BioLegend,100762	53-6.7
PerCP-Cy5.5 anti-mouse CXCR6	BioLegend,151120	SA051D1
PE-Cy7 anti-mouse CD3e	BioLegend,100319	145-2C11
BV785 anti-mouse CD45.1	BioLegend,110743	A20
Anti-CD3, Flow, IHC-P, WB, mIHC	Abcam,ab16669	Rabbit mAb, SP7
CD8 α , Mouse Specific	CST,98941	Rabbit mAb, D4W2Z
PD-L1, Mouse Specific for IHC	CST,64988	Rabbit mAb, D5V3B
Mouse Specific CXCL16 Antibody	R&D, 503-CX	142417

5) In the multiple gene association analysis, it is not clear why some cancers are included but others not. In addition what is the disease counterpart for the blood from healthy individuals will that be leukaemias? Also this type of analysis needs p value as well as R value otherwise it will be too vague

Answer: We thank the reviewer for asking the rationale behind the cancer types included in the analysis. Stat1 is known to have a dual role in cancer, tumor promoter and suppressor. We have tried to include cancers having Stat1 association both as tumor suppressor and promoter in our analysis. We have included the following cancer types in our multiple gene association analysis: Breast invasive carcinoma (BRCA), Cholangiocarcinoma (CHOL), Acute Myeloid Leukemia (LAML), Head and Neck squamous cell carcinoma (HNSC), Ovarian serous cystadenocarcinoma (OV), Skin Cutaneous Melanoma (SKMC), Sarcoma (SARC). The table below includes the references showing association of these cancer types with STAT1 expression.

Cancer type	Association	References
Breast	Dual role	PMID: 22320867, 24058806
Blood	Dual role	PMID: 15217838
Ovarian	Dual role	PMID: 33834023
Skin	Dual role	PMID: 24464587, 25690042
Head and Neck	Dual role	PMID: 21207025, 16449678

In some cancer types, the role of presence of Stat1 has been shown to have a controversial role in cancer progression and in some cases high Stat1 correlates with a poor prognosis. Hence we have not included our pooled multi-tissue type analysis (revised Fig. S5H) of the correlation of Stat1 and CD274.

Cancer type	Prognosis with high Stat1 levels	References
Uterine (endometrial)	Poor	PMID: 25267067
Renal, Glioma, Lung	Poor	PMID: 32426049

Per reviewer's suggestion, we have added the R and P-value of Spearman correlation in the revised Fig. S5H.

6) In tissue process generally it is better to section the FFPE specimen at 3um to ensure minimal overlap between disease and normal cells. Therefore the authors are advised to explain that the 4um capture the tumour areas well

Answer: We agree with Reviewer 1 that specimen's thickness is important to ensure minimal overlap between multiple layers of cells. To validate that the results derived from 4µm sections are not significantly different from 3µm sections spatially, we repeated immunofluorescence analysis using 3µm sections. We stained C3-1-TAg untreated breast tumor sections (3µm) with CD8a antibody. When we quantified the number of CD8a+ cells in sections of either 4 or 3µm, we did not find a statistically significant difference between the infiltration of CD8a+ cells (PTP R1C_Minor6).

PTP R1C_Minor6 Capture of tumor microenvironment by tissues of different thicknesses (3 and 4 microns).

(A) IF images of 3 and 4 micron thickness. (B) Box plot quantifies the number of CD8a cells infiltrating the tumors.

7) For IHC please include the dilutions used especially for Ki67 and CD3

Answer: We thank Reviewer 1 for pointing out the missing information of the antibody dilutions. The dilutions used for IHC antibodies were: Ki67 1:800, CD3 1:500, and CD8 1:250 and have been included in the manuscript.

8) In the statistical analysis section the authors need to mention that $p < 0.05$ is considered to be significant unless otherwise

Answer: We thank Reviewer 1 for pointing out the missing information. We have included the statement “ $p < 0.05$ is considered to be significant unless otherwise” in the statistical analysis section.

Reviewer #2 (Remarks to the Author):

The manuscript by Bhavana and colleagues focuses on maximizing the efficacy of anti-PD-1 immunotherapy in chemotherapy-primed breast cancer models through fine-tuning of the immunogenic and tolerogenic balance in the tumor microenvironment shaped by tumor-infiltrating myeloid cells. The authors postulate that tumor-associated myeloid cells, which are highly plastic, prevent ICB from reaching its full therapeutic potential. By leveraging single-cell transcriptomic trajectory analyses (MCT) after chemotherapy priming, they show that metronomic chemotherapy leads to accelerated T cell exhaustion through CXCL16-mediated recruitment of myeloid cells and expansion of STAT1-driven PD-L1 expressing myeloid cells. Indeed, combination treatment with anti-CXCL16 and a STAT1 inhibitor prevents T cell exhaustion, thereby enhancing the efficacy of anti-PD-1 ICB. This is a novel and potentially important finding with clear clinical implications. However, the studies are limited to mouse models of breast cancer and no attempt is made to examine human data. Furthermore, the exact roles of CXCL16 in cluster 4 and STAT1-PD-L1 in cluster 0 are not mechanistically well explained. Finally, there are major concerns with experimental design and interpretation of data that should be addressed.

Response: We thank the reviewer for acknowledging the novelty of our finding and providing critical while constructive comments to help to improve our manuscript. In this revised manuscript, we have carefully addressed each comment. Our detailed answers as follows:

Comments:

1) *The single cell seq data in the cell clusters are confusing. For example, irf8 is present in many cell clusters, which is difficult to understand since Irf8 is a cDC1 specific transcription factor.*

Answer: We agree with the reviewer that Irf8 has been widely considered as a cDC1 specific transcription factor. Yet, with the resolving power of single cell transcriptome analysis, our CITE-seq data suggested a continuum of various Irf8 regulon activity levels - shown as Gene Set Activity (AUC) - across subsets of myeloid cells (Fig. 3C-D). We believe such a continuum reflects a plastic and consistent evolving nature of myeloid cells and is consistent with the previous research showing that the transcription factor Irf8 has a significant role in myeloid cell lineage commitment, development, and function. The following evidence, including (a) published research articles, (b) publicly available single-cell data, and (c) our manuscript data, help suggest that Irf8 is essential for different myeloid cell clusters along with cDC1.

(a) Published research articles: In the table below we listed articles and the work underscoring the importance of Irf8 in differentiation of monocytes, macrophages, and importantly all splenic dendritic cell subsets.

Journal	PubMed ID	Conclusion
Blood	PMID: 22238324	In Irf8 (-/-) mice BM, cell-intrinsic defects were observed in the formation of common dendritic progenitors and all splenic DC subsets . Irf8 (-/-) mice common myeloid progenitors and Irf8 (-/-) all lymphoid Progenitors produced more neutrophils in vivo than the progenitors derived from wild type mice at the expense of DCs.
Cellular and Molecular Immunology	PMID: 34480145	Irf8 (-/-) mice have fewer CDPs which also give rise to cDC2s, Mon-DCs.
Nature Communications	PMID: 25236377	Irf8 impacts monocyte macrophage progenitor differentiation through C/EBP α . Irf8 protein is highly expressed in the monocyte macrophage progenitors. Irf8 ^{-/-} mice also lack Ly6C⁺ monocytes .
Nature Reviews Immunology	PMID: 17259967	Irf8 is required for the differentiation of CMP through Monoplast and Monocyte into Macrophages.
Journal of Experimental Medicine	PMID: 27001747	Irf8 / Irf1 /Stat1/PU.1 and Irf1 /Stat1/PU.1, both the combinations in binding to the DNA, were observed to be involved in controlling the macrophage transcription programs in the CHIP seq data. Their Bone Marrow cells for CHIP Seq analysis were derived from C57BL6/B6 WT, BXH2 (Irf8 ^{m/m}), and Irf1 ^{-/-} mice.

(b) Publicly available single-cell data: Recent single-cell data on triple-negative human patient samples shows expression of *Irf8* in different myeloid cell clusters including different DC subtypes ¹. In Bassez et al.¹, shown in Fig. PTP R2C1, the authors show intratumoral changes in TNBC upon chemotreatment. Here in Fig. PTP R2C1 A (see the next page), the violin plot shows the expression of *Irf8* in different cell types including myeloid cells, dendritic cell subsets, T cells, and B cells. In PTP Fig. R2C1 B, we subclustered untreated patient data and compared the expression of *Irf8* to other marker genes for myeloid, lymphoid, and non-immune (PTPRC low) cell types. We observed that *Irf8* was expressed in clusters other than ITGAE/ITGAX (CD103/CD11C) high ITGAM (CD11B) cells.

PTP R2C1 Irf8 expression: (A) Violin plot shows Irf8 expression in different cell types when clustered as demonstrated in the article. (B) Heatmap compares Irf8 expression with other well known markers for myeloid and lymphoid cell types in untreated patient cohort.

(c) Our data shows expression of Irf8 at both gene expression (marker gene analysis, Supplementary Table 3) and regulon level (Single-cell gene regulatory network analysis, Regulon Analysis, Fig. 3C-D). In the DGE analysis, Irf8 is differentially expressed in Cluster 3 (0.6 fold increased) which are cDCs, Cluster 4 (0.2 fold increase) which are MDSCs (Irf8 is essential for the expansion of MDSCs²), Cluster 10 (0.6 fold increase) which are DCs, Cluster 12 (0.5 fold increase) which are Mki67 high and Cd86 high myeloid cells, Cluster 14 (2 fold increase) which are cDCs, Cluster 15 (0.3 fold increase). We have included marker gene analysis in the Supplementary Table 3. In the Regulon analysis, Irf8 is actively the highest in the Cluster 3 and 14 which are DC specific clusters. This is supported by our gene regulatory network analysis shown in Fig. 3C-D.

2) T cell proliferation based on CFSE is not clearly displayed and not explained mechanistically. In Figure 1H, the gating strategy of CD8+ T cells and CFSE dilution are difficult to decipher. a) Without specific antigen loading of the DCs, why should syngeneic tumor-infiltrating DCs activate naïve CD8+ T cells in vitro? b) It is also difficult to understand how this in vitro culturing system can model the in vivo behavior of these CD11c+MHCII+ cells in the tumor setting? c) Moreover, the functional activation status of the activated CD8+ T cells should be indicated on the basis of IFN, Granzyme B staining.

Answer: We thank the reviewer for the above insightful and valuable comment. In this revised manuscript, we have replaced the original Figure 1H with a set of new panels Figure 1H-I in the revised manuscript (The new figures have been reproduced below as Fig.PTP R2C2 for reviewer's easy reference). The new figure, we believe, provides stronger experimental evidence of T cell proliferation under our study context and helps decipher gating strategy of CD8+ T cells and CFSE dilution. We have also quantified the levels of IFN- γ levels on T cells to examine the functional activation status of T cells.

For Reviewer's point a): Prior research has shown that syngeneic tumor-infiltrating DCs have the ability to stimulate naïve CD8+ T cells in vitro even without specific antigen loading in order to improve the T cell homeostasis³. To provide further evidence the tumor antigen-mediated T cell activation via dendritic cells (DCs), we used OT II tumor-antigen specific co-culture system and repeated our experiments.

PTP R2C2: (A) Flow biaxials and histogram (top) indicate the gating strategy for CD11c+ mCherry+ cells. Box plot (bottom) quantifies the mCherry+ CD11c+ cells in Vehicle and MCT-treated CD11c+ DCs. (B) Gating strategy for CD4 and CD8a OT II cells used in (C). (C) Histograms (left) show the CFSE levels on CD8a OT II cells. Boxplot (right) quantifies CD8a T cells that have proliferated more than 3 times in coculture with CD11c+ cells. (D) Gating strategy for IFN- γ levels on CD8a T cells (top) and boxplot quantifies their levels.

We first established mammary tumors by injecting E0771 tumor cells expressing a fusion mCherry protein and chicken ovalbumin 323-339 (mCherry-Ova 323-339) peptide to the C57B6 host. For the co-culture experiment, we isolated CD11c+ DCs from mCherry-Ova tumor. Two groups of DCs were isolated for subsequent T cell co-culture: 1) MCT-treated DCs: CD11c+ DCs isolated from mCherry-Ova tumor after MCT-treatment for four weeks. 2) Vehicle-treated DCs: CD11c+ DCs isolated from mCherry-Ova tumor after Vehicle treatment as control for four weeks. Total CD3+ T cells (including OT-II CD4+ T cells and CD8+ T cells) from OT-II transgenic mice were co-cultured with either Vehicle-treated DCs or MCT-treated DCs for 72 hrs. These OT-II

transgenic mice express the mouse alpha-chain and beta-chain T cell receptor that pairs with the CD4 coreceptor and is specific for chicken ovalbumin 323-339 (Ova323-339) in the context of I-Ab.

We assessed the uptake of tumor antigen (mCherry-Ova) by tumor-infiltrating CD11c+ DCs. We found that there is no significant difference in mCherry-Ova expression levels between Vehicle and MCT-treated tumor-infiltrating CD11c+ DCs (PTP. Fig. R2C2 A). Next, we examined T cell expansion/proliferation after co-cultured with tumor-infiltrating CD11c+ DCs. As shown in Fig. PTP R2C2, we observed that CD8+ T cells (gating strategy shown in PTP. Fig. R2C2 B) co-cultured with MCT treated DCs (CD11c+ cells) proliferated more than those cultured with Vehicle-treated DCs (Fig. PTP R2C2, C boxplot). Specifically, when naïve T cells from OT-II mice were cultured with activation beads, 69.6% of CD8 T cells proliferated more than 3 times compared with only 4% divided without co-culture. When co-cultured with MCT-treated DCs, 17% of CD8 T cells proliferated more than 3 times compared to 2.5% in the Vehicle group (T co-culture with vehicle-treated DCs). These data suggest a tumor antigen specific activation of CD8 T cells via DCs and OTII CD4 cells. The new data has been included in our revised manuscript Fig. 11.

For Reviewer's point b): In vitro T cell co-culture system has been extensively used to study T cell responses to drugs⁴ and myeloid-derived suppressor cells (MDSC) mediated inhibition of T cell proliferation^{5,6}. Below, we summarize literature that used an in vitro co-culture system to study the T cell and dendritic interactions. In this summary we have included literature on OT II T cell interactions with OT II peptide specific myeloid cells (including dendritic cells). We believe that this literature supports how our *in vitro* system is a valid model to study mechanisms of DC-T cell interactions.

Ge, Qing, et al. published in PNAS, 2002 (PMID: 11854473): In this article researchers elucidate the mechanisms underlying homeostatic T cell proliferation and differentiation. Due to the difficulty in manipulating in vivo systems, they approached an in vitro co-culture system. They show that naïve CD8+ T cells in the presence of syngeneic non antigen-specific dendritic cells proliferate in TCR and peptide-MHC specific manner. Furthermore, their results suggest that: (a) dendritic cells (but not other antigen-presenting cell types) provide peptide-MHC complexes to help proliferate T cells and (b) dendritic cell secreted factors are essential for such T cell proliferation.

Binnewies, Mikhail, et al. published in Cell, 2019 (PMID: 30955881): In this article researchers used OT II peptide specific tumor cell injection mouse model, B16-mCherry-OVA tumor bearing mice. Here, they isolated tumor-infiltrating dendritic cells (DCs) and co-cultured them with tumor-antigen naïve OT II T cells to learn which cell types can prime, activate, and proliferate naïve OT II T cells. Their results suggested that conventional DC2 type of dendritic cells help proliferate the naïve OT II T cells the most among other antigen-presenting myeloid cells.

For Reviewer's point c): To understand the functional activation status of the activated CD8+ T cells we stained the T cells with antibodies recognizing intracellular IFN- γ . Here, we stimulated the T cells with PMA + Ionomycin as a positive control. We observed that PMA + Ionomycin stimulation induced expression of IFN- γ in 99% of CD8a+ T cells (Fig. PTP R2C2, D), which is significantly higher than T cells that were unstimulated (0.1%). Co-culture with T cell activation beads led to 87% T cell expresses high levels of IFN- γ . Despite an increase in proliferation of CD8+ T cells when co-cultured with MCT-treated DCs (Fig. PTP R2C2 C), we did not observe a significant increase in IFN- γ level (Fig. PTP R2C2 D).

3) In Figure S2A, CITE-seq, the gating and definition of cDC1 and cDC2 are inadequate. cDCs should be defined with lineage-specific transcription factors. The discordance between the cell-surface and transcriptional status cannot be due to a high level of transcriptional plasticity of myeloid cells (Figure 2A and S2A).

Answer: We thank Reviewer 2 for this valuable suggestion. To take the advantage of CITE-seq data, which couple the cell surface marker with the transcriptome of each cells, we have used gene expression of *Irf4*, *Irf8*, and *Batf3* along with surface expression of CD11b, Ly6C, CD11c, I-A-I-E, and CD103 to classify cDC1 and cDC2. The gating strategy is shown in PTP Fig. R2C3A. While we agree that the marker-based gating has its limitations, our SCENIC regulon analysis of the gated DC showed an enrichment of *Bcl11*, *Irf8*, and *Runx2* (Fig. 3C), which strongly suggests the cell's DC identity. For instance, *Bcl11a* essential for plasmacytoid DC differentiation⁷ was highly specific to Cluster 14 cDCs. Interferon signal regulating gene *Irf8* along with general cellular growth regulators such as *Fos*, *Egr1*, and *Jun* were active in progenitor-like cells (Cluster 1) and DCs (Clusters 3, 7, and 10).

Given the similarities between the newer gating strategy (as Reviewer 2 suggested) to our current strategy in clustering on RNA-based UMAP, we have not modified our original gating strategy in our revised manuscript. We agree with the reviewer regarding the limitation of gating strategy and data interpretation. Therefore, in this revised manuscript, we have modified our text (Page 10) to suggest potential reasons for discordance between cell surface and transcriptome status which is as follows. “*This discordance could be partially attributed to the transcriptomal plasticity of myeloid cells*^{3,8}.”

PTP R2C3. cDCs identified using TFs. (A) Gating strategy used to identify cDCs. (B) cDCs (colored dots) projected on Seurat RNA UMAP. Rest of the other cell types are shown as gray dots.

4) Path 4 analysis highlights the role of *CXCL16* expressing cells in determining tumor-infiltrating T cell fate. However, as shown in Figure 4A, *CXCL16* expression is modest, which casts doubt on the utility of *CXCL16* as a target for ab blockade.

Answer: The rationale of focusing *CXCL16* in our mechanistic study as follows: Fig. 4A volcano plot shows the differential expression of *CXCL16* between Cluster 0 cells and DCs. The goal of Fig. 4A analysis is to identify marker genes for Cluster 0 and Path4 cluster cells. More importantly, after identifying a list of marker genes, we asked the question which marker gene of the Path4 cluster shows a significant increase in expression upon MCT (Fig. 4B). We reasoned that genes that significantly changed in response to MCT treatment might modulate MCT therapeutic response. This is an important rationale behind utilizing *CXCL16* as a target for blockade in MCT-treated tumors. Indeed, intratumoral injection of anti-*CXCL16* antibody significantly changed the tumor's response to anti-PD1 treatment (Fig. 4G-K).

In Figure S4C, neither the CXCL16 expressing cell type nor the sorting strategy of these cells is clear. Are these myeloid cells?

Answer: CXCL16 is a chemokine with a transmembrane domain. It expresses mainly as transmembrane form on the antigen-presenting cells, dendritic cells, CD19+ B cells, and CD14+ monocytes/macrophages⁹. It functions as a transmembrane adhesion molecule and as a membrane metalloprotease-cleaved soluble chemoattractant for CXCR6-bearing cells⁹. In our experiments we specifically looked into cell surface CXCL16 which has been shown to predominantly express on DCs and macrophages¹⁰. In Fig. S4C, CXCL16+ cells were enriched using magnetic-bead based sorting. We stained the isolated single cells from either vehicle or MCT-treated tumor with CXCL16 PE flow antibody (PE Rat Anti-Mouse CXCL16, BD 566740). The magnetic PE nanobeads were then added and cells bound to PE+ nanobeads were sorted to enrich for CXCL16+ cells. To examine what cell types express CXCL16 in cells from digested tumor tissue, we performed flow cytometry analysis on all cells isolated before magnet enrichment. When tumor dissociated cells were stained for CD45, CD11b, and CXCL16 markers, we found that most of CXCL16 high cells are also CD45+CD11b+ cells (77%) (PTP. Fig. R2C4 A).

How can the authors exclude the possibility that CXCL16 is secreted by other cells?

Answer: We agree with the reviewer that other cell types might also express CXCL16. Examining the secretory form of CXCL16 among different tumor microenvironment cell types *in situ* is technically very challenging. From single cell CITE-seq analysis (Fig. S4B. Also see PTP R2C4 B below), we observed that Seurat cluster 3, 4, 7, 10, 12 express the higher level of CXCL16. Based on the cluster marker genes, we believe the cell type identities are (See main Fig. 2A and PTP R2C4 C below):

- Cluster 3: DCs
- Cluster 4: M2-like MDSCs
- Cluster 7: DCs
- Cluster 10: DCs
- Cluster 12: Proliferating myeloid cells

Those cells might contribute to the overall pool of secretory form CXCL16 in the TME.

To examine the CXCL16 protein expression, we performed flow cytometry to detect cell surface bound CXCL16. Tumor dissociated single cells were stained with CD45, CD11b, and CXCL16. We gated for CD45+ cells and then plotted them against CD11b and CXCL16. We observed that CXCL16 was observed almost exclusively on CD11b+ CD45+ cells and only ~4% of total CD45+ cells were CD11b+ CXCL16+ (measured based on the cell surface form of CXCL16) (PTP. Fig. R2C4D and revised Fig. S4C).

PTP R2C4 - CXCL16 expression across different cell types. (A) Flow analysis of CXCL16⁺ cells. (B) Violin plot showing the relative expression of CXCL16 mRNA. (C) Cell type identity classified by Seurat Clusters. (D) Flow analysis of CXCL16⁺ cell among CD45⁺ cells.

In Figure S4C, the CFSE staining in the culture with CXCL16 cells exhibits almost no CFSE dilution (ie, proliferation) compared with bead stimulation. How did the authors manage to gate a single peak into several divisions?

Answer: The peak of undiluted CFSE⁺ T cells (Figure S4C, also reproduced below in PTP. Fig. R2C4-continued A) refers to the total T cells stained with CFSE, but without co-culturing (undivided T cells, “0” division), where the CFSE intensity peak is at $>1 \times 10^6$. This sets a reference point (negative control) and helps us gate cells expressing diluted CFSE. For the positive control, CFSE stained T cells were co-cultured with activation beads. After 72 hours, most of the T cells have had divided more than 2 times with a diluted CFSE intensity of $\sim 1 \times 10^4$ (PTP. Fig. R2C4-continued A, Dividing T cells). This gating strategy was applied to all the DC-T cells co-culture experiments.

In Figures 4c and 4d, which display the results of adoptive transfer of CD45.1 BMD into anti-CXCL16 ab treated mice, the readout is CXCL16⁺ cells in CD45.1⁺ cells, but is the intent to focus on CXCR6⁺ cells?

Answer: From single cell CITE-seq analysis, we identified that the DC (Seurat cluster 3) expresses a higher level of CXCL16 and CXCL16 is further increased after MCT treatment (Fig. 4B). We hypothesized that CXCL16 might be functionally important for a changed tumor immune microenvironment after MCT treatment. Our focus for the experiments presented in Figs 4c and 4d is to understand how neutralizing CXCL16 would impact or modulate tumor-infiltrating immune cells from the peripheral immune cell pool.

PTP Figure R2C4 -continued. (A) Gating strategy for identifying proliferated T cells (B) Dual plot of Cxcr6 and Cd3e from CITE-seq analysis. (C) Line plot for Fig. 4I. (D) Gating strategy of CD45.1⁺ progenitors and myeloid cells in IgG and NAb treated tumors. (E) Gating strategy of CD45.1⁺ T cells in IgG and NAb treated tumors.

The receptor for the CXCL16 is CXCR6, which is known to be mainly expressed in T cells. Our CITE-seq analysis also demonstrated that CXCR6 expression is restricted on T cells (PTP Figure R2C4 -continued, B). In the following analysis, instead of probing for CXCR6, we explicitly examined the infiltration of peripheral CD45.1⁺ T cells after anti-CXCL16 NAb. As expected, we observed a significant reduction of CD8 T cell tumor infiltration, but not CD4 T cells. These results have been presented in our original Fig. 4F.

In addition to T cells, to our surprise, we found a significant decrease CXCL16⁺ CD45.1⁺ cells infiltrating the tumor (Fig. 4D). As discussed above in our answers to comment #4, transmembrane CXCL16 expresses mainly on antigen-presenting cells, dendritic cells, CD19⁺ B cells, and CD14⁺ monocytes/macrophages. Our anti-CXCL16 NAb result suggests neutralizing intratumor CXCL16 modulated peripheral myeloid cell infiltration. The detailed analysis of the cell types (e.g. cluster 0 like myeloid cells and c-Kit⁺ progenitor like cells) that were impacted by anti-CXCL16 NAb were presented in the original Fig. 4E.

In Figure S4e, the gating strategy needs to be clarified or corrected. Also, what is the gating strategy of CD45.1⁺ T cells to explain the bars in Figure 4F?

Answer: For Figure S4E, we did the following: We identified adoptively transferred cells as CD45.1⁺ cells. Of these CD45.1⁺ cells, we gated for (1) c-Kit⁺ Lin⁺ cells, (2) CD11c⁺ I-A-I-E⁺ cells, and (3) CD11b⁺ PD-L1⁺ cells. The gating strategy of CD45.1⁺ T cells was presented in Figure S4F (Lymphoid cell compartment gating strategy)

The time-dependent tumor growth curve needs to be shown to explain the statistical analysis in Figure 4I.

Answer: In this revision, we have included the time dependent growth curve of tumor volumes in Fig. 4I as shown as a line plot (PTP. Fig. R2C4-continued, C) as well as revised Supplementary Figure S.4H

5) By using a single cell surface marker such as CCR1 to identify individual clusters without validation is not acceptable. In Figure S3, the explanation for selecting CCR1 to identify cluster 0 is not clear, because the CCR1 expression level on this cluster is not high, and the percentage of CCR1+CD11b+ cells in myeloid cells is only 10% (Figure 5A).

Answer: We thank the reviewer for this comment. Our rationale is as follows: CITE-sequencing data indicates *Ccr1*, *PD-L1*, *Csf3r*, *Msrb1*, *Ccl3*, *Clec4d*, and *S100a* are among differentially expressed genes by Cluster 0 cells. Among all these DE genes, *Cd14*, *Ccr1* and *PD-L1* encode cell surface proteins which could be used to sort cells fluorescently tagged antibodies recognizing above cell surface proteins. We found that both Cluster 0 cells and *Ccr1*^{high} cells have a high degree of transcriptome similarities as shown in Fig. PTP R2C5. dual feature plot (A) and heatmap (B). While *Ccr1* expression on Cluster 0 is modest, we reasoned sorting out *Ccr1*^{high} cell population using a magnet-based method will result in a population largely enriched with Cluster 0 cells.

PTP R2C5 CCR1 and Cluster 0 cells gene expression : (A) Dual plot shows coexpression of Ccr1 and Cd274 in immune cells. (B) Heatmap shows the marker genes of Cluster 0 and compares their expression between Ccr1+ cells. Cluster 0, and other myeloid cell subsets.

6) Figure 5G is confusing because, in pLKO, the % of pSTAT1+PD-L1high among CD11b+ cells is almost 100%. What is this experiment intended to show?

Answer: We apologize for the confusion. pLKO is set to be our control where the Stat1 levels remain unperturbed. We have equated the levels of pStat1+PDL1+ CD11b+ myeloid cell percent to 100% for the pLKO group. This helped us to calculate the relative change in the pStat1+PDL1+ cells upon Stat1 Knockdown (in both the constructs). This way we get relative values (relative to pLKO control, also mentioned in the Figure legend) which we believe are more comparable than absolute cell number counts across the biological repeats. In the revised Figures, we took only viable CD11b+ and reanalyzed results (Fig. 5J).

7) Figure S5A, the CD3+ T cell gating strategy is not valid, because there is no clear separation of T cells in TILs. The CD3+ T cells should be further gated with CD4 and CD8 together but not separately.

Answer: We thank the reviewer for this critique. We have reanalysed our data as per the reviewer's suggestion. We have adjusted the gating strategy and we believe it is a better representation of CD3+ T cell separation and gating them further into CD4 and CD8 T cells. We have reflected these changes in our main Fig.S5A (reproduced below as PTP R2C7A) in this revised manuscript.

PTP R2C7. T cell suppression Assay. (A) Gating strategy to identify CD4 and CD8 T cells. (B) Gating strategy for sorting Ccr1 high and Ccr1 low myeloid cells and histograms show CFSE on proliferated T cells in coculture with the myeloid cells. (C) Boxplots show % of proliferated CD4 and CD8 T cells in different co-culture conditions. (D) Biaxial plots (left) show PD-1 levels on CFSE low CD8 T cells. Box plots (right) show the ratio of CFSE low PD-1 high to CFSE high PD-1 high T cells.

8) The time-dependent tumor growth curve should be shown in Figures 6B and 6F. Without this information, it is difficult to judge the data quality in these figures

Answer: We thank the reviewer for this comment. We have graphed the time-dependent tumor growth curve as follows:

Related to Fig. 6B:

PTP Fig. R2C8. A: Line chart of absolute individual tumor size changes over time.

PTP Fig. R2C8. B: Line chart of relative tumor size changes over time

Related to Fig. 6F:

PTP Fig. R2C8. C: Line chart of absolute individual tumor size changes over time

PTP Fig. R2C8. D: Line chart of relative tumor size changes over time

PTP Figure R2C8. Tumor growth curves for Figure 6B and 6F. Line plots show (A) Absolute and (B) Relative (to the starting tumor volume) tumor volume changes for treatment groups in Figure 6B. Line plots show (C) Absolute and (D) Relative tumor volume changes for treatment groups in Figure 6F.

Reviewer #3 (Remarks to the Author):

The authors describe the modulation of the tumor immune microenvironment in breast cancer after metronomic chemotherapy (MCT). They show that MCT treatment improves survival and promotes T cell recruitment and T-myeloid cell proximity in different breast cancer models. However, MCT is also associated with T cell exhaustion and the recruitment of immunosuppressive myeloid cells that are dependent on STAT1. They propose that the inhibition of STAT1 could be an effective combinatory treatment to enhance the benefit of both standard therapy and immune checkpoint blockade (ICB) in breast cancer. The paper is interesting and could be relevant in the immunotherapy field, but some of the data are technically not convincing and do not fully support the authors' conclusions.

Specific comments:

1) The authors suggest that myeloid/T cell proximity is promoted after MCT treatment. In Figure 1E, the IF picture does not show that. Figure 1C shows a general decrease of Iba1+ cells after MCT treatment and this is not consistent with Figure 3F where Iba1+ cells are increased in the MCT treated sample. I would suggest changing the term co-localization, since it generally refers to proteins expressed by the same cell type.

Answer: We thank Reviewer 3 for pointing out the inconsistency. To clarify, the intent of our original Figure 1E IF panel only shows the IF staining quality, but not representatives of an increased myeloid cell as shown in the quantification. We agree with the reviewer that the term co-localization might be misleading. As suggested, we have changed the colocalization term to spatial proximity in our revised Figure 1C and Manuscript's text.

R3C1 Myeloid and T cell proximity. Representative IF images of C3-1-Tag breast tumor tissues stained with Iba1, CD3, and DAPI. Inset (far right) shows the CD3+ and Iba1+ cell colocalization indicated by arrows.

2) The gating strategy shown in S1F, S1G, S1I should be improved. What are the authors showing in Figure S1F and S1G? It should be quantified.

Answer: We thank Reviewer 3 for asking this question. In Fig. S1F and S1G, biaxial plots show gating strategy of highly viable CD45^{high} immune cells (as shown by Hoechst stain) (also reproduced below as Fig. PTP R2C2 A-B). We have quantified the Hoechst high CD45 high cells (Fig. PTP R3C2 C).

PTP Figure R3C2. (A) Flow cytometry gating strategy for identifying highly viable CD45 high cells. (B) CyTOF gating strategy for highly viable CD45 high cells. (C) Box plots show quantification of highly viable CD45 high cells identified in (A-B). (D) Corrected CyTOF gating strategy for NK cells. (E) Flow cytometry gating strategy for DCs infiltrating tumor draining lymph node (left) and box plot shows quantification of CD11c+ I-A-I-E+ CD45 high myeloid cells (right).

The strategy used to gate NK cells in Figure S1I is not correct: NK cells express CD11b at a lower level compared to myeloid cells, but not 100% of NK cells are CD11b+; NK cells are NK1.1+CD3-. Finally, the analysis of DC populations in tumor draining lymph nodes would be interesting.

Answer: We thank the reviewer for this insightful comment. We have corrected the NK cell gating strategy as suggested (revised Fig. S1I and Fig. PTP R3C2. D). In addition, we have analyzed the DC populations in the tumor draining lymph nodes and quantified the DCs (Fig. PTP R3C2. E). We observed a trend of increasing CD11b+/CD11c+/I-A-I-E+ DCs from 0.2% in the PBS-treated group to ~2% in MCT-treated mice among all the CD45+ cells analyzed.

3) The authors use the term “TIL” when referring to CD11b+ cells and DCs. TILs should be only used for lymphocyte populations.

Answer: We thank Reviewer 3 for the correction. We agree with the reviewer that TILs commonly refers to tumor infiltrating lymphon. To avoid confusion, in the revised manuscript, we have changed the term from TILs for myeloid cells to tumor-infiltrating immune cells throughout text.

4) The CITE-seq data are interesting but clusters are poorly defined: appropriate gene signature should be used. DCs and macrophages could be discriminated using Zbtb46 expression. Are the cells in Cluster 14 pDCs?

Answer: We thank the reviewer for this comment. As shown in Fig. PTP R3C4 A, our single cell CITE-seq analysis detected a low level of Zbtb46 gene expression (potentially due to sequencing depth), making it difficult to use Zbtb46 as a marker gene to discriminate between macrophages and dendritic cells. Instead, when examining the cluster-specific marker genes collectively, Cluster 14 cells differentially expressed Flt3, Siglech, Irf8 along with many other genes related to DC function (Fig. PTP. R3C4 B). Flt3 is known to promote the differentiation of myeloid cells in plasmacytoid DCs and Siglech is one of its markers^{3,11,12}. Furthermore, SCENIC regulatory analysis shows active Bcl11a transcription factor in Cluster 14 cells⁷ (Fig. 3C). Bcl11a transcription factor is essential for plasmacytoid DC differentiation⁷. Given the expression of Flt3, Siglech, Irf8, and Bcl11a, Cluster 14 is identified as pDCs.

PTP R3C4. DCs identified using Zbtb46. (A) Violin plots show expression of Zbtb46. (B) Volcano plot shows differentially expressed genes (DEG) in Cluster 14.

In Figure 2F the authors show the enriched annotated pathways for the myeloid subclusters and they are all related to T cell functions. This should be clarified.

Answer: We thank the reviewer for pointing out this interesting observation. Indeed, some T cell function related pathways are enriched with myeloid cells clusters after the MCT treatment, e.g. BIOCARTA_THELPER_PATHWAY and RUAN_RESPONSE_TO_TNF_DN. We delved into the genes contributing to these pathways using the ssGSEA website database¹³. The THELPER_PATHWAY includes genes such as ITGL (a myeloid cell activation gene¹⁴), ICAM1 (adhesion of myeloid cells¹⁵). RUAN_RESPONSE_TO_TNF_DN pathway includes CEBPG/A, ITGA6 which are related to myeloid cell differentiation. These observations suggest that the genes included in curated gene sets are not T cell exclusive. This interesting finding also helped us rationalize that the enrichment of these pathways might reflect the interactions between myeloid cells and T cells.

5) The authors suggest that CXCL16+ cells stimulate T cell function (page 12), but the Figure (Figure S4C) shows the opposite: the CFSE dilution is increased without CXCL16+ cells. Also, the gating strategy and plots showing the sorting of CXCL16+ cells would be interesting.

Answer: We thank the reviewer for this question/suggestion. In the co-culture set up we have T cells with activation beads as a positive control where the T cells divide. However, in the experimental samples, we have T cells co-cultured with CXCL16+ cells. As shown in the Fig. PTP. R3C5 A, the undiluted T cells' peak represents the unproliferated T cells without beads stimulation or co-culture (negative control). The T cells with activation beads show the clear peaks of T cells with diluted CFSE in different proliferative phases (i.e. 1, 2, and 2+). In summary:

- T cells only = negative control, T cells are not activated and shows highest level CFSE (undiluted)
- T cells + beads = positive control, the activation beads increase the T cell activation and expansion
- T cells + CXCL16+ cells = experimental samples.

When co-cultured with CXCL16+ cells, we observed different degrees of T cell proliferation. As shown in the PTP Figure R3C5 and revised Fig. 4SC, co-culture with CXCL16+ cells led to more T cells (both CD4 and CD8) with 1 or 2 divisions compared with T cell activation beads. Among the CD4+ T cells in the coculture with CXCL16+ cells, around 40% have divided twice whereas as few as <10% have not divided (remain undiluted). Similarly, among the CD8+ T cells in the coculture with CXCL16+ cells, around 60% have divided twice whereas as few as <10% have not divided (remain undiluted).

PTP Figure R3C5. (A) Gating strategy of T cells in different proliferative phases. (B) CFSE dilution compared between positive control and T cells cocultured with CXCL16+ cells. (C) Box plots quantify the CD4 (left) and CD8 (right) T cells in different division phases. (D) Gating strategy of CXCL16+ cells.

6) In Figure 4, the authors show the effect of anti-CXCL16 antibody treatment in the breast cancer tumor model. How do the authors explain the reduced recruitment of CXCL16+ cells between the control and the MCT-treated group (Figure 4D)?

Answer: Our single cell CITE-sequence data (Fig. 4A-B) shows increased levels of Cxcl16 mRNA expression mainly on the dendritic cells (DCs) cluster in MCT-treated tumors. Given CXCL16 is a potent chemoattractant regulating the chemotaxis, we hypothesized that an increased CXCL16 at the tumor microenvironment might modulate infiltration of peripheral immune cells. Thus, the goal of CD45.1 adoptive transfer experiment (Fig. 4C-F) is to examine the peripheral CD45.1 immune cell chemotaxis and potential tumor-infiltration modulated by the CXCL16 neutralizing within the tumor.

The reviewer 3 pointed out a very interesting observation that the baseline level of CD45.1 cell tumor infiltration was reduced in the MCT-treated group, as shown in IgG-treated control vs MCT (Fig. 4D). Given the CXCL16 CD45.1 levels tend to be lower in the MCT-treated tumors (Fig. 4D), we cannot exclude the possibility that overall CXCL16 levels and or specifically CD45+ CD11c+ I-A-I-E+ CXCL16+ levels are lower in MCT when compared to Vehicle-treated tumors. This result strongly suggests that MCT treatment does NOT promote the

peripheral CXCL16+ cell tumor infiltration, pointing to the possibility that increased CXCL16+ DCs within the tumor after MCT treatment are mostly derived from tumor resident immune cells.

The MCT treatment is shown to be associated to an increased frequency of PD1high CD4 T cells and this is reduced by the Ab treatment (Figure 4E). Is this the case for the second experimental setting (Figure 4G)?

Answer: CXCL16 NAb modulated the frequency of peripherally derived immune cells including DCs (Fig. 4E) and PD1 high CD4 and CD8 T CD45.1+ cells (Fig. 4F). To further investigate the therapeutic potential of combining immunotherapy with CXCL16 NAb, we designed Fig. 4G. The Fig. 4G-K is a separate set of experiments unrelated to Fig. 4C-F. There is no adoptive transfer of CD45.1 cells. Therefore, it is not directly comparable to Fig. 4F.

The authors suggest that the Ab treatment attracts exhausted PD1+ T cells. The fact that they are recruited is not proven. Where would the reservoir of exhausted T cells be? It would be more likely that the treatment modulates PD1 induction within the tumor. Tumor growth is reduced by the CXCL16 treatment (Figure 4H). How is tumor growth upon CXCL16 treatment in the experiment shown in Figure 4C?

Answer: We thank the reviewer for this important point. Our data suggest that peripheral derived T cells can infiltrate the tumor and this infiltration is impacted by neutralization of CXCL16. We agree that, despite the observed reduction of PD1 high CD4+ T cell modulation by CXCL16 NAb, the result does not exclude the possibility of the pre-existing (in the tumor) T cell exhaustion due to the NAb.

To understand if PD1+T cells are recruited into the tumor and learn where the exhausted T cell reservoir is, We used a CD45.1 CD3T cells specific adoptive transfer experiment. In this experiment, we treated the mice either with Vehicle or MCT for two weeks, then intratumorally (i.t.) injected three doses of either IgG or CXCL16 NAb (Fig. R3C6.A). On the second day of Ab injection, we adoptively transferred CD45.1 CD3 T cells into the tumor bearing mice. CXCL16 NAb i.t. injection significantly decreased the overall percentage of tumor-infiltrated CXCL16+ T cells in MCT-treated tumors within 48 hrs of adoptive transfer (Fig. R3C6.B). Among all the adoptively transferred CD45.1 T cells in the tumor, PD-1+ CD8a+ CD45.1+ T cells percentage increased from 13% in IgG to 80% in NAb in PBS-pre-treated mice. Similar increase was also detected in MCT-treated mice, from 38% in IgG to 67% in NAb group (Fig. R3C6.C-D).

Next, we cross compared PD-1+ CD8a+ CD45.1+ T cells in blood, tumor draining lymph nodes, and tumor tissues (Fig. R3C6.E-F). We observed that in IgG treated mice, CD45.1+ PD-1+ CD8+ T cell percentage was 8% (in blood), 2% (in lymph nodes), and 38% (in tumor tissues). In CXCL16 NAb treated mice, CD45.1+ PD-1+ CD8+ T cell percentage was 7%, 4%, and 65% respectively. There are significantly higher % adoptively transferred CD45.1 T cells that showed an exhaustion phenotype in the tumor compared with either blood or lymph. This observation suggests a model of T cell exhaustion after entering in the tumor microenvironment, which is consistent with previous research reporting that tumor tissues are infiltrated with CD8+ T cells ¹⁶ and get exhausted due to a constant exposure to tumor-antigen ¹⁷ and start to exhibit a higher PD-1 expression ¹⁸.

With this new evidence, we have modified our conclusions in the revised manuscript (page 17-18). We greatly appreciate the reviewer for this critical and constructive comment, which helped to strengthen our manuscript.

PTP R3C6 PD1 induction and infiltration. (A) Schematic shows the experiment setting. (B) Box plots show levels of CXCL16+ cells among all the singlets. (C) Gating strategy for PD-1 high CD8 T+ CD3e+ CD45.1+ T cells. (D) Box plots quantifying PD-1+CD8a+ cells among the CD3+ CD45.1+ cells in tumors (E) Box plots quantifying PD-1+CD8a+ cells among the CD3+ CD45.1+ cells in MCT+IgG treated blood, lymph node, and tumor (E), in MCT+CXCL16 NAb treated blood, lymph node, and tumor (F).

7) In Figure 5H, the authors show an IF quantification of PD-L1 expression upon STAT1 inhibition. First, in the IF picture there is a difference in Iba1 total cells after STAT1 inhibition.

Answer: We thank Reviewer 3 for this question. The IF picture in Fig. 5H is not a good representation of the number of Iba1+ cells throughout the section we examined, but rather a demonstration of PD-L1 IF staining quality. In this revised manuscript, we quantified the Iba1 positive cells across the tissue section. We observed a trend of reduction of overall Iba1+ cells after the treatment although the result is not statistically significant (PTP. Fig. R3C7 A). Furthermore, to specifically examine the impact of STAT1 on CD11b+ cell viability, we have quantified the CD11b+ viable cells upon Stat1 knockdown *in vitro* (please also see our answer to R3 Critique #8 below). We treated C3-1-TAg tumor-bearing mice with MCT for 28 days. After treatment, tumors were digested and single cells were isolated, and CD11b+ cells were enriched through magnet-based cell enrichment/isolation. The cells were plated in u bottomed 96 well plates. Then shRNA lentivirus of either pLKO or Stat1 (constructs 468 or 2253) were added. The cells were incubated with shRNA for 72 hrs before proceeding for flow cytometry. We observed that the overall viable CD11b cell percentage did not change upon Stat1 KD (PTP. Fig. R3C7 B-C). This genetic evidence suggests that STAT1 is not essential to maintain most types of CD11b+ cells' viability.

Then, how is tumor growth in these groups? The flow analysis of PD-L1 expression in the conditions shown in Figure 5H would corroborate the finding.

We have plotted the tumor growth curve under designated treatments (PTP R3C7 D). In this experiment, C3-1-Tag tumor was treated by MCT for 2 week before receiving anti-PD1 Ab only or Stat1 inhibitor only or combo. Both MCT+STAT1i or MCT+STAT1i+anti-PD1 Ab showed better efficacy than MCT+anti-PD1 only treatment group (Fig. PTP. R3C7 D). Furthermore, with Stat1 inhibition, the number of CD11b⁺/PD-L1^{high} cells among all CD45⁺ cells decreased (PTP. Fig. R3C7 E, flow analysis).

R3C7 (A) Quantification of Iba1+ cell across IF stained tumor sections. (B) Gating strategy for viable CD11b+ myeloid cells. (C) Box plots quantify Zombie aqua low viable CD11b+ myeloid cells after STAT1 shRNA (sh468 and sh2253). (D) Line plots indicate tumor volume changes in different treatment conditions. (E) Flow cytometry gating and quantification show the PD-L1 levels with and with out Stat1 inhibition.

8) The ex vivo culture of tumor-associated CD11b+ cells and gene knock down (Figure 5G) is a challenging experiment. Can the authors show cell viability and the CD11b+ cell subpopulations?

Answer: We thank Reviewer 3 for this comment. We treated C3-1-TAG tumor-bearing mice with MCT for 28 days. After treatment, tumors were digested and single cells isolated. The cells were plated in u bottomed 96 well plates. Then shRNA lentivirus of either pLKO or Stat1 (constructs 468 or 2253) were added. The cells were incubated with shRNA for 72 hrs before proceeding for flow cytometry. We tested the viability of CD11b cells using Zombie aqua. As shown in Fig. R3C8A, the singlets with low autofluorescence and low Zombie Aqua levels were gated as viable cells. Interestingly, the addition of Stat1 shRNA did not significantly change the viability of most singlets (Fig. R3C8B) and myeloid cells (Fig. R3C8C). The viable singlet percentages were 51% in pLKO, 51% in sh468, and 42% in sh 2253 (Fig. R3C8B). The viable myeloid cell percentages were 28, 24, and 27 in pLKO, sh468, and sh2253 groups respectively (Fig. R3C8C).

We also investigated the PD-L1^{high} pStat1^{high} Cluster 0 cells in viable CD11b cells (Fig. R3C8D). The viable Cluster 0 cell number decreased upon Stat1 KD to 56% in sh468 construct and to 19% in sh2253 construct relatively when compared to the 100% in pLKO group (Fig. R3C8E). We next looked into CD11c+/I-A-I-E^{high} cells (Fig. R3C8F) and CD11c+/I-A-I-E^{high}/CXCL16^{high} cells (Fig. R3C8G) among all viable CD11b+ cells, since

Stat1 Regulon score is high among trajectory Path 4 Cluster 3 dendritic cells (Fig. 3E). We observed that CD11b dendritic cell percentage did not significantly change in the sh468 and sh2253 group compared to the pLKO group, despite some degree of variations. Above analysis demonstrated that Stat1 plays a more significant role in the PD-L1^{high} pStat1^{high} Cluster 0 cells.

R3C8 Cell viability after shRNA-mediated STAT1 knockdown. (A) Gating strategy for viable CD11b+ myeloid cells. Box plots quantify (B) Zombie aqua low viable singlets (C) Zombie aqua low viable CD11b+ myeloid cells. (D) Gating strategy for PD-L1 high pStat1 high myeloid cells in pLKO shRNA treated cells on left and Stat1 shRNA468 construct on right. (E) Box plot quantifies viable PD-L1 high pStat1 high Zombie aqua low CD11b+ cells in different treatment groups. (F) % of CD11c+/I-A-I-E high dendritic cells among Zombie aqua low CD11b high cells (G) % of CD11c+/I-A-I-E high dendritic cells among Zombie aqua low CD11b high cells (H) % of CD11c+/I-A-I-E high dendritic cells among Zombie aqua low CD11b high cells

References

1. A single-cell map of intratumoral changes during anti-PD1 treatment of patients with breast cancer - PubMed. <https://pubmed.ncbi.nlm.nih.gov/33958794/>.
2. Veglia, F., Perego, M. & Gabrilovich, D. Myeloid-derived suppressor cells coming of age. *Nat. Immunol.* **19**, 108–119 (2018).
3. Maier, B. *et al.* A conserved dendritic-cell regulatory program limits antitumour immunity. *Nature* (2020) doi:10.1038/s41586-020-2134-y.
4. Faulkner, L. *et al.* The development of in vitro culture methods to characterize primary T-cell responses to drugs. *Toxicol. Sci. Off. J. Soc. Toxicol.* **127**, 150–158 (2012).

5. Wang, G. *et al.* Targeting YAP-Dependent MDSC Infiltration Impairs Tumor Progression. *Cancer Discov.* **6**, 80–95 (2016).
6. Lu, X. *et al.* Effective combinatorial immunotherapy for castration-resistant prostate cancer. *Nature* **543**, 728–732 (2017).
7. Ippolito, G. C. *et al.* Dendritic cell fate is determined by BCL11A. *Proc. Natl. Acad. Sci.* **111**, E998–E1006 (2014).
8. Satpathy, A. T., Wu, X., Albring, J. C. & Murphy, K. M. Re(de)fining the dendritic cell lineage. *Nat. Immunol.* **13**, 1145–1154 (2012).
9. Gough, P. J. *et al.* A disintegrin and metalloproteinase 10-mediated cleavage and shedding regulates the cell surface expression of CXC chemokine ligand 16. *J. Immunol. Baltim. Md 1950* **172**, 3678–3685 (2004).
10. Shimaoka, T. *et al.* Cell surface-anchored SR-PSOX/CXC chemokine ligand 16 mediates firm adhesion of CXC chemokine receptor 6-expressing cells. *J. Leukoc. Biol.* **75**, 267–274 (2004).
11. Reizis, B., Bunin, A., Ghosh, H. S., Lewis, K. L. & Sisirak, V. Plasmacytoid dendritic cells: recent progress and open questions. *Annu. Rev. Immunol.* **29**, 163–183 (2011).
12. Mitchell, D., Chintala, S. & Dey, M. Plasmacytoid dendritic cell in immunity and cancer. *J. Neuroimmunol.* **322**, 63–73 (2018).
13. Gene set enrichment analysis: A knowledge-based approach for interpreting genome-wide expression profiles | PNAS. <https://www.pnas.org/doi/10.1073/pnas.0506580102>.
14. Yang, H. *et al.* Transcriptome profiling of brain myeloid cells revealed activation of Itgal, Trem1, and Spp1 in western diet-induced obesity. *J. Neuroinflammation* **16**, 169 (2019).
15. Makgoba, M. W. *et al.* ICAM-1 a ligand for LFA-1-dependent adhesion of B, T and myeloid cells. *Nature* **331**, 86–88 (1988).
16. Kumagai, S. *et al.* The PD-1 expression balance between effector and regulatory T cells predicts the clinical efficacy of PD-1 blockade therapies. *Nat. Immunol.* **21**, 1346–1358 (2020).
17. Bally, A. P. R., Austin, J. W. & Boss, J. M. Genetic and Epigenetic Regulation of PD-1 Expression. *J. Immunol. Baltim. Md 1950* **196**, 2431–2437 (2016).
18. Ahmadzadeh, M. *et al.* Tumor antigen-specific CD8 T cells infiltrating the tumor express high levels of PD-1 and are functionally impaired. *Blood* **114**, 1537–1544 (2009).

REVIEWER COMMENTS

Reviewer #1 (Remarks to the Author):

The authors addressed most of the issues related to the manuscript and agreed to focus the study on triple negative breast cancer including changing the title of the manuscript.

However, the one key part that still needs addressing at least some validation on human subjects. The reason for this is that mouse immune system is very different from human immune system and if co-culturing human breast cancer cells lines with some immune cells is not feasible or does not reflect what we see in BC patients, then I strongly recommend the reviewers to test at least some of the markers on patient samples. Whilst re-analysis of data from single cell study: <https://pubmed.ncbi.nlm.nih.gov/33958794/> can illustrate some of the changes due to PD1, it is not sufficient to show changes via STAT1 and/or CXCL16.

Therefore, I believe that in order to validate some of the findings from this study in mice either the authors can explore the possibility of carrying out cross-sectional study using for example immunohistochemistry on markers they identified from this study and then obtain single cells based on the immunostaining e.g. using laser capture then carry out targeted molecular analysis e.g. using qRT-PCR to show co-expression of specific biomarkers obtained from this study. Alternatively, if this is not feasible, the authors may considering co-culturing of BC cell lines with different immune cell lines to mimic immune response in human breast cancer. If it is the former and getting IRB may take time a suggestion would be to look at breast cancer tissue microarray and enquire if there are samples from treated patients. Example of commercial source can be: <https://tristargroup.us/product/breast-cancer-100-tissue-microarray/>

Also there are few resources for academics such as: <https://cancer.uillinois.edu/researchers-clinicians/breast-cancer-working-group-tissue-microarray-access/> and would be good to enquire if they have an treated cases. This provide better citation for the study but also eliminates some of the doubts that may result from purely relying on mouse model for mechanistic insights into the treatment of breast cancer.

If obtaining patient samples from the above is not possible, the authors should try to find another independent transcriptome data set and show that their analysis on the PD1 single cell data set: <https://pubmed.ncbi.nlm.nih.gov/33958794/> is consistent across different and independent studies.

Reviewer #2 (Remarks to the Author):

The authors have provided additional data to address the criticisms and questions of the original review. While partly satisfactory, there remain a number of issues that have not been adequately addressed.

1. PTP R2C2: To investigate Ag-specific CD8+ T cell priming and proliferation induced by DCs, the authors used the EO771-Ova 329-339 expressing cell line, and isolated tumor CD11c+ DCs to incubate with these cells and T cells. However, this experiment cannot address CD8+ T cell proliferation, since Ova 329-339 is an MHCII-restricted peptide and OTII mice are CD4+ Ova-specific TCR transgenic mice. Therefore, the revised figure does not make any sense. Figure 2H-I (PTP R2C2) appears to be mislabeled, and the data misinterpreted. In Figure 2I, which includes CD8+ T cells, the original IFN γ intracellular cellular staining flow cytometry data should also be included.
2. Fig 1C-E: Since the authors use CD11c cells to address DC-T cell functional interaction, CD11c histological staining should be shown in Figure 1C-E.
3. CITE-seq analysis, The markers defining conventional DCs are not appropriate. To define cDC1s in mice, XCR1 and Clec9A should be used, and for human cDC1s, CD141, Clec9A should be used. For mouse cDC2s, SIRP1a, CD11b should be used, while CD1c should be used for human cDC2s. Siglech is a pDC marker in mice, and therefore cluster 14 in Figure 2B is not clearly defined. Ly6C

is a monocyte maker, which cannot be used to define conventional dendritic cells. In addition, the heterogeneous transcriptome status change after MCT treatment is descriptive, and the selected targets of CXCL16 and STAT1 seem arbitrary.

4. Figure 4C: MCT treatment increased the PD-1+ percentage of both CD4+ and CD8+ T cell populations; however, PD1+ T cell expression does not predict the response to ICB. The authors should consider TCF1+ T cells (Tpex), which may indicate responsiveness to ICB based on recent publications.

5. Fig. R2C4: The rationale for using anti-CXCL16 ab remains unclear and CXCL16 staining is difficult to interpret. To address the importance of myeloid cell CXCL16, the authors are encouraged to study mice with a Cre-specific CXCL16 genetic deletion. Single cell seq analysis, not real-time PCR, would be needed to show CXCL16 expression in individual immune cells in tumors.

6. Fig. R2C4 A: CXCL16+ cells were only 0.034%? How could such a small number of cells impact anti-tumor immunity?

7. PTP. Fig. R2C4 D: After magnetic sorting, CD45+ cells were only 0.82% and CXCL16 only 3.79% and only in CD11b+ cells? Where is the FMO control?

8. Fig. PTP. R2C4: The authors isolated CCR1+CD11b+ cells and cocultured them with syngeneic naïve CD8+ T cells. In the data shown, the positive readout is based on results obtained in cultures with beads, CCR1+CD11b+ cells and IL-2. No cultures are shown without beads or with only T cells with beads+IL-2. In the absence of such controls, these experiments cannot be interpreted.

9. PTP Fig. "R2C4-continued A": Missing from the experiment is a coculture of CXCL16+ cells enriched from TNBC tumors with naïve T cells without beads. Adding beads to the coculture does not inform how CXCL16+ cells stimulate naïve T cells, since in this syngeneic coculture system, DC cannot prime naïve T cells without Ag loading. Moreover, naïve T cells priming is known to occur in tumor draining lymph nodes rather than the tumor; therefore, using tumor infiltrating CXCL16+ cells to prime naïve T cells does not have a clear rationale.

10. Figure 4G-H: adoptive transfer of CD45.1+ cells: the original gating strategy for CD45.1+ cells, CD4+ and CD8+ T cells should be shown.

11. Figure 4M: Regarding the cluster 1 increase after inhibition of CXCL16, what is the relationship of this increase to anti-tumor immunity? Also, what is the relationship of STAT1 in cluster1 with CXCL16 secretion in cluster 0?

12. Figure 6H-I: Why was IL-2 added to the DC-T cell coculture?

13. Figure 6: Missing critical data: The authors state in the abstract that "Inhibiting immature myeloid cell specific STAT1 signaling in MCT-primed breast cancer relieved T cell exhaustion and significantly enhanced the sensitivity of anti-PD-1 ICB treatment.". If so, there should be profound changes in the CD8 T cells infiltrating the tumors, yet no such data are shown. The authors are encouraged to perform an analysis of tumor infiltrating CD8+ T cells following treatment, including at the very least an analysis of frequency, surface phenotype and expression of IFN γ , TNF α , Granzyme B, and Perforin. Such data would add significantly to the significance of the findings.

Reviewer #3 (Remarks to the Author):

The authors addressed all my concerns.

There is still one technical issue that I would like them to clarify: in Supplementary Figure1, they use DAPI to define cell viability by flow cytometry and they select DAPI high cells as viable cells. However, DAPI penetrates the membrane of dead (or dying) cells. Could the authors clarify their strategy and/or method for this?

Point-to-Point Answers to Reviewer's Critiques

RE: NCOMMS-22-02938A

We would like to thank the reviewers for valuable and constructive comments. In this revision, we have carefully considered each additional point raised by reviewers and address concerns by new patient sample analysis and in vivo experiments with our best efforts. We have provided additional evidence from human sample analysis and new in vivo experiments. In response to reviewer's request, we also modified our data analyses and provide additional explanations to address concerns of reviewers. We believe this version of our manuscript has been further improved.

To comply Nature Communications formatting guidelines, we also shorten the manuscript title and main text when appropriate. The major changes of the manuscript text have been highlighted with blue fonts. Our Point-to-Point response to each Reviewer's comments are as follows:

Reviewer 1:

1. However, the one key part that still needs addressing at least some validation on human subjects.

Responses:

We fully understand the reviewer's perspective. In this revision, we have obtained limited number of human TNBC breast cancer tissue specimens (untreated or treated with neoadjuvant ddACT regimen, see patient clinical information in new Supplementary Table 8) from UT Southwestern Tissue Core and performed immunofluorescence of CD11b, p-STAT1(S727) and CXCL16. We quantified the percentage of tumor infiltrating CD11b+/CXCL16+ myeloid cells and tumor infiltrating CD11b+/p-STAT1(S727) myeloid cells. Tumor infiltrating CD11b+/CXCL16+ myeloid cells increased from ~2% in the naive untreated group to ~14% in the chemo-treated group (**PTP. Fig R1C1 A-B**, see next page) and CD11b+/pStat1+ myeloid cells increased from ~13% to 25 % (**PTP. Fig R1C1 C-D**). We remain cautious in interpreting human data. Because metronomic chemotherapy (MCT) is not a standard clinical practice, human breast cancer specimens we obtained are from patients treated with maximum tolerant chemotherapy (MTD) regimen of ddACT, not MCT. Despite the above limitations, we believe that both IF staining and single cell analysis from patient data suggested a similar trend of increasing pSTAT1 and CXCL16 signaling after chemotherapy, supporting the clinical relevance of our results. We have put above human data as the new Figure panel in Supplementary Fig. S3 I-J and Supplementary Fig. S4 B.

PTP Figure R1C1a. Human patient breast tumor immunostaining. (A-B) Representative images of Iba1, CXCL16, and DAPI stain and their quantification of human breast cancer with and without ddACT chemotherapy. (C-D) Representative images of Iba1, pSTAT1, and DAPI stain and their quantification of human breast cancer with and without ddACT chemotherapy. Data in boxplots represents quantification of random regions ($n > 3$) from four untreated and five treated patients.

“Whilst re-analysis of data from single cell study: <https://pubmed.ncbi.nlm.nih.gov/33958794/> can illustrate some of the changes due to PD1, it is not sufficient to show changes via STAT1 and/or CXCL16.”

Responses:

We have analyzed Bassez., et.al single cell dataset (A single-cell map of intratumoral changes during anti-PD1 treatment of patients with breast cancer. Nat Med. 2021 May;27(5):820-832. and <https://pubmed.ncbi.nlm.nih.gov/33958794/>) in our previous revision. Because this dataset contains anti-PD1 treated patient sample, we were able to discern certain effects due to PD-1 treatment. We agree with the reviewer, since the patients in this single cell dataset did not receive any STAT1 inhibitor or CXCL16 neutralizing antibody, it is impossible to directly infer any causal changes via STAT1 and/or CXCL16 based on current data available. Unfortunately, validation of causal role of STAT1 requires a new clinical trial with MCT neoadjuvant chemotherapy followed by STAT1 inhibitor and/or CXCL16 antibody, which is beyond the scope of current preclinical study.

Reviewer 2:

1. PTP R2C2: To investigate Ag-specific CD8⁺ T cell priming and proliferation induced by DCs, the authors used the EO771-Ova 329-339 expressing cell line, and isolated tumor CD11c⁺ DCs to incubate with these cells and T cells. However, this experiment cannot address CD8⁺ T cell proliferation, since Ova 329-339 is an MHCII-restricted peptide and OTII mice are CD4⁺ Ova-specific TCR transgenic mice. Therefore, the

revised figure does not make any sense. Figure 2H-I (PTP R2C2) appears to be mislabeled, and the data is misinterpreted. In Figure 2I, which includes CD8+ T cells, the original IFN γ intracellular cellular staining flow cytometry data should also be included.

Response:

In this revision, we reperfomed co-culture experiment to investigate Ag-specific CD8+ T cell priming, and proliferation induced by DCs. We performed co-culture experiments using a chicken ovalbumin (Ova 257-264)-OT-I system. We first established E0771.ova tumor cells that are stably expressing 257-264 peptide (SIINFEKL) using a Piggybac vector containing an ER-targeting SIINFEKL gene fragment. The membrane presentation of SIINFEKL peptide was validated by flow cytometer using MHC class I molecule Kb bound to SIINFEKL (PTP Fig.R2C1a-A). We also obtained OT-I mice (Jax # 003831, C57BL/6-Tg (TcraTcrb)1100Mjb /J). These OT-I transgenic mice express the mouse alpha-chain and beta-chain T cell receptor that pairs with the CD8 coreceptor and is specific for chicken ovalbumin 257-264 (Ova 257-264) in the context of I-Ab. We validated Ova 257-264 specific T cells cells (splenocytes from OT-I transgenic mice) by flow cytometer using BV421 Flex-T™ H-2K(b) OVA (SIINFEKL) Tetramer (BioLegend, 280051). >90% of CD3+ T cells bind to SIINFEKL Tetramer (PTP Fig.R2C1a-B).

PTP R2C1a Establishing OVA-OT-I system (A) Upper panel: Gating strategy to identify SIINFEKL+ E0771 tumor cells. Lower panel: Flow analysis of SIINFEKL expression E0771 cells. (B) Biaxial plots showing the gating strategy to identify the SIINFEKL tetramer positive viable unstimulated OT-I T cells isolated from OT-I transgenic mice.

Next, we injected E0771.ova cells into the mammary fat pad of B6 mice to form primary breast cancer, followed by four weeks of Vehicle (PBS)/MCT treatment as we did before. For the co-culture experiments, we isolated CD11c+ tumor infiltrating lymphocytes (TIL DCs) from E0771-Ova tumors. Four groups of DCs were isolated for subsequent T cell co-culture experiments:

- 1) Vehicle-treated TIL-DCs: CD11c+ DCs isolated from E0771-Ova tumor after Vehicle treatment as control for four weeks.
- 2) MCT-treated TIL-DCs: CD11c+ DCs isolated from E0771-Ova tumor after MCT-treatment for four weeks.
- 3) Vehicle-treated LN-DCs: CD11c+ DCs isolated from E0771-Ova tumor draining lymph node after Vehicle treatment as control for four weeks.
- 4) MCT-treated LN-DCs: CD11c+ DCs isolated from E0771-Ova tumor draining lymph nodes after MCT-treatment for four weeks.

Total CD3+ T cells from OT-I transgenic mice were co-cultured with either DCs for 72 hrs. We examined T cell expansion/proliferation using CFSE dye dilution assay after co-cultured with TIL-DCs or LN-DCs.

When naïve T cells from OT-I mice were stimulated with anti-CD3-CD28, 60.5% of CD8 T cells divided more than 3 times (PTP Fig. R2C1b-A-B). When co-cultured with MCT-treated TIL-DCs, 28.3% of naïve CD8 T cells proliferated more than 3 times compared to 7.6% in the Vehicle-treated TIL-DCs group (Fig. PTP R2C1b-B boxplot), indicating an increase of ova antigen-specific activation of T cells under the MCT treatment. We have included this piece of new data in our revised manuscript as new Fig. 1G-H. On the other hand, when OT-I T cells were co-cultured with lymph node derived DCs, the % of T cells that divided more than 3 times were similar in both the treatment groups with 19.8% in Vehicle-treated LN-DCs and 20.7% in MCT-treated LN-DCs (Fig. PTP R2C1b-C boxplot). Furthermore, we observed that MCT co-cultures DC co-culture led to more CD44+ T cells and ~15% CD44+ proliferating CD8 T cells showed a higher level of IFN-gamma, which is comparable with positive control (T cells stimulated with anti-CD3-CD28). Neither co-culture with Vehicle-treated TIL-DCs nor LN-DCs (Fig. PTP R2C1b-D) induced high IFN-gamma T cells.

PTP R2C1b OT1 T cell coculture with tumor infiltrated Cd11c+ dendritic cells. (A) Gating strategy to identify CD8 OT1 T cells and IFN- γ positive CD8 T cells. (B) Box plots show percentage of CD8 T cells in division phases 0 to 3+ in different treatment strategies upon co-culture with tumor-infiltrating DCs. (C) Box plot shows % of CD8 T cells in division phase 3 in different treatment strategies upon co-culture with lymph node derived DCs. (D) Box plots show CD8+IFN- γ + CD8 T cells in co-culture with DCs infiltrating the lymph node (left) and tumor (right). Data ($n > 3$) in box plots represent Mean \pm SEM, Diamonds Means, Black lines Medians, and T-test p-values.

2. Fig 1C-E: Since the authors use CD11c cells to address DC-T cell functional interaction, CD11c histological staining should be shown in Figure 1C-E.

Response:

We have attempted to perform IF staining using anti-mouse Cd11c antibodies on FFPE tumor tissues. However, the antibody specificity for Cd11c staining is not satisfactory on FFPE sections. Therefore, we did not include Cd11c staining in our revision. However, we believe that additional ova-culture data (as stated above) provided more informative results that support our conclusions.

3. CITE-seq analysis, The markers defining conventional DCs are not appropriate. To define cDC1s in mice, *XCR1* and *Clec9A* should be used, and for human cDC1s, *CD141*, *Clec9A* should be used. For mouse cDC2s, *SIRP1a*, *CD11b* should be used, while *CD1c* should be used for human cDC2s. *SiglecH* is a pDC marker in mice, and therefore cluster 14 in Figure 2B is not clearly defined. *Ly6C* is a monocyte maker, which cannot be used to define conventional dendritic cells.

Response:

We thank Reviewer 2 for this suggestion. We have corrected our Figure 2A and Fig.S2B panels according to the suggestion and the panels are shown in **PTP Fig. R2C3**. We have used *XCR1*, *SiglecH*, and *SIRP1a* for identification of DC subsets and removed *Ly6C* markers from our gating strategy.

PTP R2C3. CITE-seq analysis on DCs. (A) Gating strategy used to identify cDCs and TAMs using cell surface markers. (B) cDCs (colored dots) projected on Seurat RNA UMAP. Rest of the other cell types are shown as gray dots.

In addition, the heterogeneous transcriptome status change after MCT treatment is descriptive, and the selected targets of CXCL16 and STAT1 seem arbitrary.

Response:

The nature of CITE-seq analysis is descriptive. Despite the descriptive nature, one important value of such a global view of transcriptome changes at single cell resolution is to suggest potentially important changes (hypothesis) in subsets of cells that might warrant further mechanistic study. The rationale behind selecting *CXCL16* and *STAT1* comes from our differential gene expression and Single-Cell Gene Regulatory Network Analysis of our CITE-Seq data.

- 1) **CXCL16:** Upon differential gene expression analysis on dendritic cell clusters (identified using Seurat RNA UMAP clustering and gene expression on CITE-Seq data) with and without MCT treatment, we observed a significant upregulation of *CXCL16* in MCT treated group (**PTP.R2C3 - A**). We therefore hypothesized that *CXCL16* changes after MCT treatment might influence tumor microenvironment immune dynamics. If *CXCL16* is functionally important for TME, neutralization of *CXCL16* might modulate tumor microenvironment and improve therapeutic efficacy upon combining with ICB treatment. Hence we explored *CXCL16* as one of our targets.
- 2) **STAT1:** Through Single-Cell Gene Regulatory Network Analysis on our CITE-Seq data, we observed that the specific myeloid cell clusters were regulated by the *STAT1* gene (**PTP.R2C3 - B**). Upon further performing *STAT1* targetome analysis we observed that PD-L1 (*Cd274*) was

regulated by STAT1 gene, further strengthened by in vitro STAT1 shRNA Knock down assay. Hence, we selected STAT1 as one of our targets for functional study.

PTP. R2C3 Target Selection Rationale. (A) Violin plot shows the expression of CXCL16 between Control and MCT treated dendritic cells. (B) Scatterplot shows the Cluster 0 Regulon normalised enrichment scores. (C) Scatter plot shows the genes regulated in the Cluster 0 Stat1 regulon.

4. Figure 4C: MCT treatment increased the PD-1+ percentage of both CD4+ and CD8+ T cell populations; however, PD1+ T cell expression does not predict the response to ICB. The authors should consider TCF1+ T cells (Tpex), which may indicate responsiveness to ICB based on recent publications.

Response:

We thank Reviewer 2 for pointing out Tpex cells. We agree that several other exhaustion markers, such as TCF1 (encoded by the TCF7 gene), Lag3, Tox, and CTLA4, should be considered along with PD1. We have analyzed tumor infiltrating T cells after Vehicle/MCT treatment using CITE-seq. We performed unbiased clustering of T cells using CITE-seq data (PTP. R2C4-A). There are three major clusters of T cells. We plotted a heatmap for top differentially expressed genes between different T cell clusters (PTP. R2C4-A). Cluster 0 T cells expressed CD8. Thus, we focused our analysis on this cluster. Next, we compared gene expression between control of MCT treatment in each T cell clusters (PTP. R2C4.B). PD1, Lag3 and CTLA4 were differentially increased upon MCT treatment in Cluster 0 T cells. We also plotted markers for precursors of exhausted T (Tpex) cells, including Tcf7, Sell(encodes CD62L), Myb (Utzschneider et al., 2020). While Tpex markers are only moderately expressed in Cluster 0 T cells, we observed an overall higher expression in MCT treated group, correlating with T cell exhaustion markers (PTP. R2C4-B). Above gene signatures analyses collectively suggested that MCT treatment reshapes a subcluster of tumor infiltrating CD8+ T cells towards an enrichment of Tpex phenotype. We have included these panels in our updated Figure.S5B.

PTP. R2C4 CITE-seq T cell analysis after MCT treatment. (A) Heatmap (left) shows identity of different Cd3e high T cells from CITE-Seq data. (B) Dotplot (right) split by treatment group to show percentage of cells expressing various progenitor and functional markers genes.

5. Fig. R2C4: The rationale for using anti-CXCL16 ab remains unclear and CXCL16 staining is difficult to interpret. To address the importance of myeloid cell CXCL16, the authors are encouraged to study mice with a Cre-specific CXCL16 genetic deletion. Single cell seq analysis, not real-time PCR, would be needed to show CXCL16 expression in individual immune cells in tumors.

Response:

We thank Reviewer 2 for the question regarding the rationale of the experiments related to CXCL16 . As we have stated above in our response to Comment 3, we performed Single-cell RNA seq (CITE-Seq) on immune cells isolated from the tumors of mice treated with either PBS or MCT for four weeks. We observed that the CXCL16 gene expression was upregulated in myeloid cells upon MCT treatment (see above **PTP. R2C3-A**). Thus, we hypothesized that CXCL16 is upregulated due to the exhausted/inflamed microenvironment, as shown in the article Abel et al. 2004 (Abel et al., 2004). We further hypothesized that neutralization CXCL16 in TME will decrease the tumor exhaustion and improve the tumor reduction upon combining with MCT. Hence, we used anti-CXCL16 neutralizing antibody in our study. We agree that the neutralizing antibody approach is not cell type specific. However, the priority of our study is to investigate the functional importance of CXCL16 in regulating TME composition and response to ICB. We have shown CXCL16 neutralization within the tumor impacts the immune cell infiltration, both myeloid and T cells (Figure 4). In our previous revision, we have also performed CITE-seq single cell analysis and observed an increase in the expression of CXCL16 in specific myeloid cell clusters upon MCT treatment (CITE-seq analysis, Supplementary Fig. S4A).

While Cre-specific CXCL16 genetic deletion is useful to study the cell-type specific role CXCL16, we feel this request is beyond the scope of the current manuscript. In this revision, we have changed the manuscript title to tone down the significance of “myeloid-specific CXCL16” from “Maximizing the Anti-tumor Potential of Immune Checkpoint Blockade through Modulation of Myeloid-specific CXCL16 and STAT1 Signaling in Triple-Negative Breast Cancer” to “Maximizing the Anti-tumor Potential of Immunotherapy through Targeting CXCL16 and STAT1 in Triple-Negative Breast Cancer”.

6. Fig. R2C4 A: CXCL16+ cells were only 0.034%? How could such a small number of cells impact anti-tumor immunity?

Response:

In R2C4, the percentage of CXCL16+ cells is 0.034% among **all cells** isolated from tumors and not just CD45+ cells. Literature also suggests that DCs are rare myeloid populations that can impact anti-tumor immunity, following are a few references.

Broz., et al. published in Cancer Cell, Dissecting the Tumor Myeloid Compartment Reveals Rare Activating Antigen Presenting Cells, Critical for T cell Immunity (Broz et al., 2014)

In this article, authors dissect the tumor myeloid compartment in different mouse models. They show that CD103+ DCs are rare antigen presenting DCs which have remarkably high antigen presentation ability and capability of activating CTL in the tumor. These populations were also found in human tumors.

Perez CR., et al. published in Nature Communications, Engineering dendritic cell vaccines to improve cancer immunotherapy (Perez and De Palma, 2019)

In this perspective, authors discuss the importance of DCs in anti-tumor immunity, their sparsity, and the significance of DC vaccines.

Salmon., et al. published in Cancer Cell, Expansion and Activation of CD103(+) Dendritic Cell Progenitors at the Tumor Site Enhances Tumor Responses to Therapeutic PD-L1 and BRAF Inhibition (Salmon et al., 2016)

In this article, the authors show how paucity of DCs negatively impacts immunotherapy and how this can be addressed.

Merad., et al. published in Annu Rev Immunol, The Dendritic Cell Lineage: Ontogeny and Function of Dendritic Cells and Their Subsets in the Steady State and the Inflamed Setting (Merad et al., 2013)

Here, authors discuss the ontogeny, lineage commitment, and percentage of DCs.

7. PTP. Fig. R2C4 D: After magnetic sorting, CD45+ cells were only 0.82% and CXCL16 only 3.79% and only in CD11b+ cells? Where is the FMO control?

Response:

We thank the Reviewer for this comment. Here, we performed an independent flow analysis of tumor-infiltrating (PTP R2C7) CD45+ CD11b+ with CXCL16 and without CXCL16 (FMO) to demonstrate the specificity of our approach.

8. Fig. PTP. R2C4: The authors isolated CCR1+CD11b+ cells and cocultured them with syngeneic naïve CD8+ T cells. In the data shown, the positive readout is based on results obtained in cultures with beads, CCR1+CD11b+ cells and IL-2. No cultures are shown without beads or with only T cells with beads+IL-2. In the absence of such controls, these experiments cannot be interpreted.

Response:

In this set of co-culture experiment, we aim to study the role of CCR1+ Cluster 0 myeloid cells in T cell activity. Based on the single cell analysis, we hypothesized that CCR1+ myeloid cells might have immune suppressive (MDSC-like) activity. Therefore, the co-culture experiment was designed to test suppressive

PTP R2C7. Histogram overlay for CXCL16. The histogram shows the expression of CXCL16 on CD45+ CD11b+ cells in FMO control and CXCL16 stained cells infiltrating the tumor.

activity of Ccr1+ myeloid cells. In the data shown for coculture of CCR1+CD11b+ cells with naive T cells, we do have only T cells with beads+IL-2 as our positive control (activated T cells, shown in gray boxes in the box plots - **PTP.Fig. R2C8.C**) and T cells without stain and co-culture as our negative control.

PTP R2C8. T cell suppression Assay. (A) Gating strategy to identify CD4 and CD8 T cells. (B) Gating strategy for sorting Ccr1 high and Ccr1 low myeloid cells and histograms show CFSE on T cells in coculture with the myeloid cells. (C) Boxplots show % of proliferated CD4 and CD8 T cells in different co-culture conditions.

9. PTP Fig. "R2C4-continued A": Missing from the experiment is a coculture of CXCL16+ cells enriched from TNBC tumors with naive T cells without beads. Adding beads to the coculture does not inform how CXCL16+ cells stimulate naive T cells, since in this syngeneic coculture system, DC cannot prime naive T cells without Ag loading. Moreover, naive T cells priming is known to occur in tumor draining lymph nodes rather than the tumor; therefore, using tumor infiltrating CXCL16+ cells to prime naive T cells does not have a clear rationale.

Response:

We thank Reviewer 2 for this comment. The rationale behind this experiment is to learn the impact of CXCL16+ cells on naive T cells. These naive T cells were isolated from the spleen of a non-tumor bearing mouse (of the same background of the tumor bearing mouse). We hypothesize that the CXCL16+ cells might help naive T cells proliferation. We agree with the reviewer, although the co-culture with CXCL16+ cell somewhat instigated naive T cells and led to some degree of proliferation. The experiment itself is not antigen specific. It did not directly inform whether CXCL16+ cell has DC-like activity after MCT. Thus, to avoid confusion, we removed this panel from Supplementary Fig. 4.

10. Figure 4G-H: adoptive transfer of CD45.1+ cells: the original gating strategy for CD45.1+ cells, CD4+ and CD8+ T cells should be shown.

Response:

We have now shown our gating strategy as part of in revised Supplementary Fig. 4C-D.

11. Figure 4M: Regarding the cluster 1 increase after inhibition of CXCL16, what is the relationship of this increase to anti-tumor immunity? Also, what is the relationship of STAT1 in cluster1 with CXCL16 secretion in cluster 0?

Response:

We believe the Reviewer was referring to Cluster 0, instead of Cluster 1. There is no cluster 1 in the Fig. 4M. In our manuscript text “results” section we explained Fig. 4M in the following lines, “*Moreover, while the percentage of CD11b high PD-L1 high cells among CD45+ cells did not change, treatment of CXCL16 NAb led to an increase of Cluster 0 cells under MCT plus anti-PD-1 dual treatment (Fig. 4M, right), suggesting a shift in homeostasis of the trajectory towards Path 1 Cluster 0 (Cd274+, Cd14+, S1008/9+) compensatory to the inhibition of CXCL16 signaling.*”. We believe that CXCL16 is important for the infiltration of the tumor with CXCL16+ cells. However, when CXCL16 levels decrease within the TME, we observe an increase in Cluster 0 cells. Although there is no direct evidence suggesting the regulation between CXCL16 and STAT1, neutralization of CXCL16 from the TME led to an increase of Cluster 0 cells, which has a high STAT1 activity.

12. Figure 6H-I: Why was IL-2 added to the DC-T cell coculture?

Response:

Our T cell expansion/proliferation assay set-up is below: The CD11c+ cells were FACS sorted and co-cultured with T cells in IL-2 supplemented heat inactivated RPMI medium. IL-2 is a T cell growth promoting factor for all the groups of this experiment. We agree with the reviewer's prior point regarding the specific antigen loading in co-culture experiments. Due to the ambiguous (non-Ag-specific) nature of these results, we decided to remove the panel H-I from the revised Figure 6.

13. The authors state in the abstract that “Inhibiting immature myeloid cell specific STAT1 signaling in MCT-primed breast cancer relieved T cell exhaustion and significantly enhanced the sensitivity of anti-PD-1 ICB treatment.”. If so, there should be profound changes in the CD8 T cells infiltrating the tumors, yet no such data are shown. The authors are encouraged to perform an analysis of tumor infiltrating CD8+ T cells following treatment, including at the very least an analysis of frequency, surface phenotype and expression of IFN γ , TNF α , Granzyme B, and Perforin. Such data would add significantly to the significance of the findings.

Response:

We thank the reviewer for this question. We also agree with the reviewer’s point that this data will enhance the significance of the findings. We performed an independent Flow Cytometry analysis where we compared the CD4 and CD8 infiltration into the C3-1-TAg tumors after MCT+anti-PD1 or MCT+anti-PD1+Stat1i. We observed (**PTP. Fig. R2C13**) that there was an increase in T cell infiltration into the tumors. Given the significance of the data we decided to include this in the manuscript Fig. 6G.

Moreover, we have also modified our statement in abstract from “Inhibiting immature myeloid cell specific STAT1 signaling in MCT-primed breast cancer relieved T cell exhaustion and

significantly enhanced the sensitivity of anti-PD-1 ICB treatment.” to “Inhibiting STAT1 signaling in MCT-primed breast cancer significantly sensitized TNBC to ICB treatment.”

Reviewer 3

There is still one technical issue that I would like them to clarify: in Supplementary Figure1, they use DAPI to define cell viability by flow cytometry and they select DAPI high cells as viable cells. However, DAPI penetrates the membrane of dead (or dying) cells. Could the authors clarify their strategy and/or method for this?

Response:

We apologize to Reviewer 3 for the lack of clarity. Here, we try to explain and clarify our experiment method. Method: MMTV-PyMT tumors from mice treated with either PBS or MCT were digested and stained for DNA with Hoescht (Thermo Fisher, Catalog # 62249) and immune cells with CD45 antibody. One million cells were stained in the Cell staining buffer (from Biolegend) at a concentration of 2 μ M in 100 μ L of cell staining buffer for 20 min before washing and proceeding for flow cytometry. Since Hoechst, not DAPI, can be used to stain and track live cells, we gated Hoescht high cells as “Viable cells”. Of these viable cells, we gated for CD45 high immune cells.

In this revision, we have removed the panels from the Supplementary Figure 1 as we think they do not add any significance to our main conclusion.

References Cited:

- Abel, S., Hundhausen, C., Mentlein, R., Schulte, A., Berkhout, T.A., Broadway, N., Hartmann, D., Sedlacek, R., Dietrich, S., Muetze, B., Schuster, B., Kallen, K.-J., Saftig, P., Rose-John, S., Ludwig, A., 2004. The transmembrane CXC-chemokine ligand 16 is induced by IFN-gamma and TNF-alpha and shed by the activity of the disintegrin-like metalloproteinase ADAM10. *J. Immunol. Baltim. Md* 1950 172, 6362–6372. <https://doi.org/10.4049/jimmunol.172.10.6362>
- Broz, M.L., Binnewies, M., Boldajipour, B., Nelson, A.E., Pollack, J.L., Erle, D.J., Barczak, A., Rosenblum, M.D., Daud, A., Barber, D.L., Amigorena, S., Van’t Veer, L.J., Sperling, A.I., Wolf, D.M., Krummel, M.F., 2014. Dissecting the tumor myeloid compartment reveals rare activating antigen-presenting cells critical for T cell immunity. *Cancer Cell* 26, 638–652. <https://doi.org/10.1016/j.ccell.2014.09.007>

PTP. Fig. R2C13. T cell infiltration into the tumors. (A) Box plots quantify the infiltration of tumors with CD4 (left) and CD8 (right) in MCT+anti-PD1 and MCT+ant-PD1+Stat1i treatment groups.

- Merad, M., Sathe, P., Helft, J., Miller, J., Mortha, A., 2013. The dendritic cell lineage: ontogeny and function of dendritic cells and their subsets in the steady state and the inflamed setting. *Annu. Rev. Immunol.* 31, 563–604. <https://doi.org/10.1146/annurev-immunol-020711-074950>
- Perez, C.R., De Palma, M., 2019. Engineering dendritic cell vaccines to improve cancer immunotherapy. *Nat. Commun.* 10, 5408. <https://doi.org/10.1038/s41467-019-13368-y>
- Salmon, H., Idoyaga, J., Rahman, A., Leboeuf, M., Remark, R., Jordan, S., Casanova-Acebes, M., Khudoynazarova, M., Agudo, J., Tung, N., Chakarov, S., Rivera, C., Hogstad, B., Bosenberg, M., Hashimoto, D., Gnjatic, S., Bhardwaj, N., Palucka, A.K., Brown, B.D., Brody, J., Ginhoux, F., Merad, M., 2016. Expansion and Activation of CD103(+) Dendritic Cell Progenitors at the Tumor Site Enhances Tumor Responses to Therapeutic PD-L1 and BRAF Inhibition. *Immunity* 44, 924–938. <https://doi.org/10.1016/j.immuni.2016.03.012>
- Utzschneider, D.T., Gabriel, S.S., Chisanga, D., Gloury, R., Gubser, P.M., Vasanthakumar, A., Shi, W., Kallies, A., 2020. Early precursor T cells establish and propagate T cell exhaustion in chronic infection. *Nat. Immunol.* 21, 1256–1266. <https://doi.org/10.1038/s41590-020-0760-z>

REVIEWERS' COMMENTS

Reviewer #1 (Remarks to the Author):

The authors attempted to address many gaps present in this study. The manuscript reads much better than previous version and the authors attempted to validate some of the findings on patients' TNBC samples, albeit small cohort of 10 patients. However the following needs to be addressed before the manuscript can be considered for publication:

1) In lines 963 to 965 please the code for the ethical approval used to conduct the experiments on TNBC patient samples

2) In the supplementary table two of the patients are over 75; patient 1 is 80 and patient 3 is 78. It will be good if an explanation is provided as to whether the TNBC is primary or secondary considering the age of the patients.

Minor comments:

Line 86: should be TIME instead of TIEM

Lines 106 and 107: use x instead of * for the numbers

Reviewer #2 (Remarks to the Author):

The authors have addressed most of my concerns with either new experimental data or adequate explanations, and I believe the work is of significance to the field and supports the authors' conclusions.

Reviewer #3 (Remarks to the Author):

The authors addressed my concerns.

Point-to-Point Answers to Reviewers Comments

RE: NCOMMS-22-02938A

We would like to thank the reviewers for valuable and constructive comments. In this revision, we have carefully considered each additional point raised by reviewers and addressed concerns. We have provided additional information regarding human study analysis and explanation regarding age of the patients to address concerns of reviewers. We believe this version of our manuscript has been further improved.

To comply Nature Communications formatting guidelines, we modified and edited the text when appropriate. The major changes of the manuscript text have been highlighted with track changes. Our Point-to-Point response to each Reviewer's comments are as follows:

Reviewer #1 (Remarks to the Author):

The authors attempted to address many gaps present in this study. The manuscript reads much better than previous version and the authors attempted to validate some of the findings on patients' TNBC samples, albeit small cohort of 10 patients. However the following needs to be addressed before the manuscript can be considered for publication:

1) *In lines 963 to 965 please the code for the ethical approval used to conduct the experiments on TNBC patient samples*

Answer: We thank the reviewer for this question. The study with TNBC clinical specimens in this publication was approved by the UT Southwestern Institutional Review Board (IRB# STU102010-051).

2) *In the supplementary table two of the patients are over 75; patient 1 is 80 and patient 3 is 78. It will be good if an explanation is provided as to whether the TNBC is primary or secondary considering the age of the patients.*

Answer: We thank the reviewer for this question. We have reconfirmed with UT Southwestern tissue bank. All TNBC tissues used in this publication were collected from the primary tumor site.

Minor comments:

Line 86: should be TIME instead of TIEM

Answer: We thank the reviewer for this question. We have corrected the typo. *Modulating the STAT1 signaling pathway in TIME myeloid cells fine-tunes the anti-cancer immunity of ~~TIEM~~ TIME to maximize the therapeutic efficacy of anti-PD1 for TNBC.*

Lines 106 and 107: use x instead of * for the numbers

Answer: We thank the reviewer for the question. We have replaced * with x. The count of CD45⁺ cells in peripheral blood decreased from ~~5*10³~~ 5×10^3 in Vehicle treated to ~~3*10³~~ 3×10^3 per μL blood upon MTD treatment while the count increased to ~~14*10³~~ 14×10^3 per μL of blood in MCT-treated mice (**Fig. S1c**).